# Reactant-dependent volcano trends in Pt-catalyzed cycloalkane dehydrogenation: orbital hybridization-guided design of active sites

Yongxiao Tuo[1,6], Jingying Qu[2,6], Huailu Sun[2], Qing Lu[2], Bin Wang [3,4] ✉, Hongwei Gai[1], Defu Yin[2], Xiang Feng [2] ✉ & De Chen [1,5] ✉

Catalyst design is often reaction-specific due to the lack of universal descriptors linking molecular structures to optimal active-site configurations. Here, we uncover a reactant-dependent volcano relationship in Pt-catalyzed dehydrogenation, governed by the Pt-Pt coordination number ($CN_{Pt-Pt}$). Using a tunable $Pt/MgAl_2O_4$ system, we identify optimal $CN_{Pt-Pt}$ values of ~2.5, ~4.7, and ~7.0 for cyclohexane, methylcyclohexane, and decalin, respectively. This activity trend is explained by an orbital hybridization-guided mechanism, where optimal activity emerges from a balance between C-H bond activation and product desorption, mediated by the interactions between Pt d-orbitals and the π* orbitals of dehydrogenated intermediates. We introduce the LUMO energy of aromatic products as a universal electronic descriptor that serves as a proxy for π* orbital energy. LUMO energy shows strong linear correlations with the d-band center of Pt at the optimal $CN_{Pt-Pt}$, enabling rational tuning of electronic interactions across diverse reactants. The optimized $Pt/MgAl_2O_4$ catalyst delivers 2-3 times higher activity with >100 h stability under industrial conditions at only 300 °C, offering a robust strategy for hydrogen release and active-site engineering in dehydrogenation catalysis.

Traditional catalytic research has relied heavily on extensive trial-and-error methods to discover high-performance catalysts[1–4]. However, recent advances in characterization techniques, theoretical modeling, and artificial intelligence have clarified structure-performance relationships, offering critical insights that support rational catalyst design[5–8]. Despite these advancements, the derived correlations are often specific to particular reactions and may not translate well to related systems[9–11]. To overcome this limitation, it is essential to study catalyst structure-performance relationships across a series of reactions while incorporating the reactant molecular structure as an additional dimension. This broader approach could uncover more universal correlations, thereby enhancing the generalizability of catalyst design strategies.

The development of efficient dehydrogenation catalysts for cycloalkanes, involving cyclohexane, methylcyclohexane, decalin, and related compounds, has become a critical research frontier as these

[1]State Key Laboratory of Heavy Oil Processing, College of New Energy, China University of Petroleum (East China), 66 West Changjiang Road, Qingdao, Shandong, China. [2]College of Chemistry and Chemical Engineering, China University of Petroleum (East China), 66 West Changjiang Road, Qingdao, Shandong, China. [3]Department of Chemical Engineering, Shaanxi Key Laboratory of Energy Chemical Process Intensification, Engineering Research Center of New Energy System Engineering and Equipment, Xi'an Jiaotong University, Xi'an, China. [4]Shaanxi HydroTransformer Energy Technologies Co., Ltd, Xi'an, China. [5]Department of Chemical Engineering, Norwegian University of Science and Technology, Trondheim, Norway. [6]These authors contributed equally: Yongxiao Tuo, Jingying Qu. ✉e-mail: binwang@xjtu.edu.cn; xiangfeng@upc.edu.cn; de.chen@ntnu.no

reactions serve as foundational processes in contemporary hydrocarbon valorization. These transformations not only generate valuable aromatic products such as benzene but also play an essential role in hydrogen storage systems when integrated with aromatic hydrogenation[12–15]. With hydrogen transportation posing significant challenges for the hydrogen economy, liquid organic hydrogen carriers, exemplified by the methylcyclohexane/toluene system, offer substantial potential to mitigate costs associated with large-scale hydrogen storage and distribution[16–18].

Pt-based catalysts have proven to be among the most effective for cycloalkane dehydrogenation[19–22]. However, the optimal active site structure of these catalysts varies significantly with the reactant. For example, a Pt catalyst anchored on curved graphene layers, with an average coordination number of approximately 2, is optimal for cyclohexane dehydrogenation[23]. In contrast, a higher coordination number of about 4.4 is more effective for the dehydrogenation of dodecahydro-N-ethylcarbazole[24]. Moreover, our previous research indicates that the optimal occupancy of the Pt 5 d state required for maximum catalytic activity differs between cyclohexane and decalin dehydrogenation[25]. Understanding these differences is crucial for guiding rational catalyst design. It requires an in-depth investigation into the mechanisms of cycloalkane dehydrogenation and the influence of molecular structure, ultimately establishing a framework that aligns cycloalkane structure with the design of optimal active sites.

The size of Pt species plays a critical role in cycloalkane dehydrogenation, as numerous studies have explored its influence across different reactions[26–28]. However, even within a single reaction, discrepancies in the reported optimal nanoparticle size frequently arise due to variations in catalyst preparation, support materials, and reaction conditions[29–32]. Moreover, the presence of highly dispersed Pt species, such as single atoms or sub-nanometer clusters, can unpredictably affect performance[33,34]. Each Pt-based catalytic system exhibits unique active site requirements, challenging the generalizability of traditional structure-performance relationships that assume uniform active sites. Therefore, a comprehensive and systematic evaluation of Pt species, ranging from single atoms to nanoparticles under consistent conditions across a series of cycloalkane dehydrogenation reactions, is essential for deriving reliable trends and deeper mechanistic insights. Such an approach will help uncover the intricate structure-sensitive relationships that underpin catalytic performance and ultimately advance the development of Pt-based catalysts.

In this study, we strategically engineered a tunable Pt/MgAl$_2$O$_4$ (Pt/MAO) system spanning single atoms, clusters, and nanoparticles, creating a coordination gradient (CN$_{Pt-Pt}$, defined as the Pt-Pt coordination number of catalytically active sites) to decode the catalytic fingerprint of dehydrogenation. Employing typical hydrogen carriers (cyclohexane, methylcyclohexane, and decalin) as probe molecules, we uncovered a dynamic volcano activity dependence on CN$_{Pt-Pt}$, with the peak reactivity shifting toward higher CN$_{Pt-Pt}$ as the reactant molecular size increased. Combining kinetic profiling and DFT simulations, we revealed this shift stems from a trade-off between strong adsorption of unsaturated species at lower CN$_{Pt-Pt}$ and weak reactant adsorption (C-H activation) at higher CN$_{Pt-Pt}$, resulting in an optimal CN$_{Pt-Pt}$ that balances these competing factors. Furthermore, electronic property analysis exposed a hidden driver: decreasing LUMO energy in dehydrogenation products enhances the interactions between Pt d-orbitals and the π* orbitals of dehydrogenated intermediates, necessitates higher CN$_{Pt-Pt}$ with lower d-band center to maintain optimal adsorption-desorption equilibrium. Leveraging this insight, we derived a linear correlation between LUMO energy and the optimal Pt d-band center, and validated it through the dehydrogenation of perhydro-benzyltoluene and perhydro-dibenzyltoluene. The optimized Pt/MAO catalyst achieved record-breaking performance, achieving 2–3 times higher activity than previously reported systems while maintaining excellent stability over 100 h at 300 °C with a WHSV of 4.7 h$^{-1}$.

## Results and discussion
### Synthesis and structural characterization of catalysts
A series of Pt/MgAl$_2$O$_4$ catalysts with tunable Pt dispersion were synthesized through precisely controlled Pt loading. The MgAl$_2$O$_4$ support was fabricated using an ethanol-modified solvothermal approach, followed by hydrogen treatment to create a defect-rich spinel surface containing abundant oxygen vacancies. These defects facilitated exceptional Pt-support interactions, acting as effective nucleation sites that enabled atomic-level Pt dispersion at low loadings. Nitrogen physisorption characterizations confirmed retention of stable pore structures and textural properties across all catalysts (Figure S1 and Table S1). ICP analysis verified the precise Pt loading for each sample (Table S2), with catalysts named accordingly (e.g., 0.02Pt/MAO, 0.05Pt/MAO). Aberration-corrected HAADF-STEM (Fig. 1a) visualized the structural evolution of Pt. At the lowest loading (0.02%), Pt was predominantly present as single atoms. Increasing the loading to 0.05% led to the formation of sub-nanometer clusters. At 0.15% loading, Pt primarily existed as clusters with a few larger particles also evident. At even higher loadings (0.375%, 1%, and 3%), Pt predominantly formed nanoparticles with average sizes of approximately 1.40, 1.70, and 2.45 nm, respectively, as corroborated by XRD (Figure S2) and CO chemisorption measurements (Table S2).

In situ CO-DRIFTS (Fig. 1b) was used to probe the atomic configuration of Pt sites. For 0.02Pt/MAO, a characteristic band at 2064 cm$^{-1}$ indicated CO linearly adsorbed on Pt single atom[27,35]. As loading increased, the vibrations of Pt-CO gradually red-shifted, with the emergence of bands at 2061 cm$^{-1}$ and 2057 cm$^{-1}$ corresponding to CO adsorption on Pt clusters and Pt nanoparticles[36], respectively. The deconvolution showed a combination of single atoms and clusters for 0.05Pt/MAO, while clusters dominated with some nanoparticles for 0.15Pt/MAO. At loadings above 0.375%, the band at 2057 cm$^{-1}$ confirmed the predominance of Pt nanoparticles. Additional bands at 1821 cm$^{-1}$ and 2180 cm$^{-1}$ corresponded to bridge adsorption on Pt and linear adsorption on PtO$_x$, respectively. Notably, the blue shift in the 2057 cm$^{-1}$ band with increasing Pt loading indicated the transformation from under-coordinated to well-coordinated Pt species[36,37], consistent with nanoparticle growth (Figure S3).

X-ray absorption near-edge structure (XANES) (Fig. 1c) and extended X-ray absorption fine structure (EXAFS) spectroscopy (Fig. 1d) provided detailed insights into the electronic and coordination structures of Pt species. In 0.15Pt/MAO, where limited Pt-support interactions occurred, the average electronic state of Pt remained comparable to Pt foil, indicating negligible electronic perturbation. As the Pt loading decreased, a growing fraction of Pt atoms engaged with the support, resulting in a systematic increase in electron deficiency. Notably, 0.02Pt/MAO, where Pt predominantly existed as isolated atoms, exhibited the most pronounced electron-deficient state. This evolution in Pt coordination was further mapped by EXAFS spectroscopy. As the Pt loading increased from 0.02Pt/MAO to 0.15Pt/MAO, an increase in Pt-Pt backscattering relative to Pt-O signals indicates the evolution from isolated single atoms to clusters and eventually to nanoparticles. Wavelet transform EXAFS (WT-EXAFS) offered a clear visualization of these transitions in k and R space (Fig. 1e). For 0.02Pt/MAO, only a Pt-O backscattering peak at 1.6 Å was observed, indicating the presence of isolated Pt atoms. In contrast, both 0.05Pt/MAO and 0.15Pt/MAO displayed a weak Pt-Pt backscattering signal at 2.6 Å, indicating the formation of Pt clusters. The intensity of peak became more pronounced in the 0.15Pt/MAO sample, aligning with a cluster-dominated structural configuration.

### Catalyst performance in cycloalkane dehydrogenation
The catalytic performance of the Pt/MAO catalyst series was evaluated for the dehydrogenation of cyclohexane, methylcyclohexane, and decalin (Figure S4). As shown in Fig. 2a, all three dehydrogenation reactions followed a volcano-shaped performance curve as a function

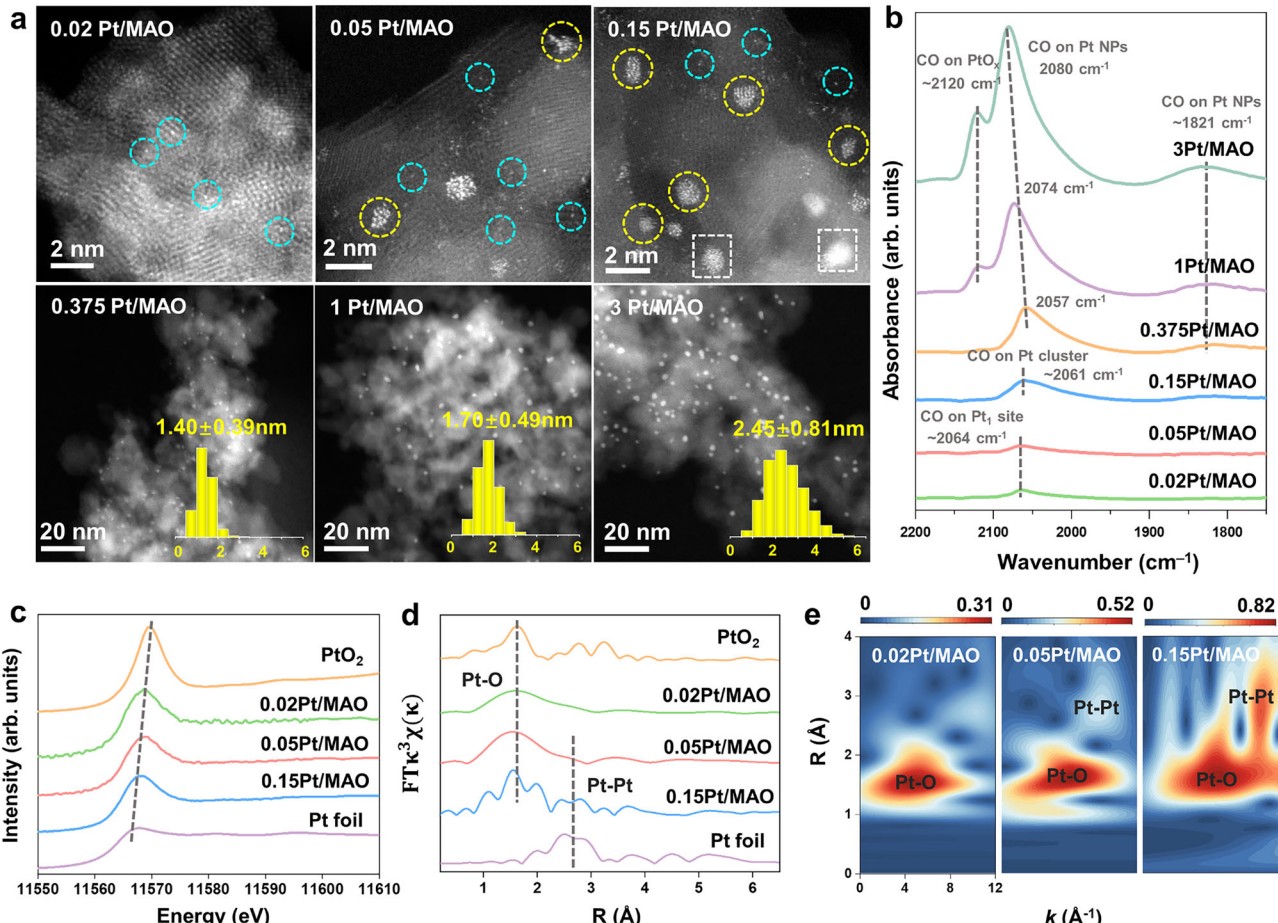

**Fig. 1 | Structural characterization of Pt/MAO catalysts. a** Atomic-resolution HAADF-STEM images of Pt/MAO catalysts, highlighting Pt single atoms (blue circles), clusters (yellow circles), and nanoparticles (white rectangles). **b** CO-DRIFTS spectra of Pt/MAO catalysts with varying Pt loadings. **c** Pt $L_{III}$-edge XANES spectra of 0.02Pt/MAO, 0.05Pt/MAO, and 0.15Pt/MAO, with bulk Pt foil and $PtO_2$ as references. **d** Pt $L_{III}$-edge FT-EXAFS spectra of 0.02Pt/MAO, 0.05Pt/MAO, and 0.15Pt/MAO, with bulk Pt foil and $PtO_2$ as references. **e** Wavelet transform of Pt $L_{III}$-edge EXAFS signals for 0.02Pt/MAO, 0.05Pt/MAO, and 0.15Pt/MAO.

of Pt species size at 280 °C. Notably, the Pt species size that yielded optimal performance increased with the molecular size of the cycloalkane, a trend that remained consistent across different reaction temperatures (270, 290, and 300 °C) (Figure S5). Comparative analysis with literature data (Fig. 2b, Table S3-5) demonstrates that Pt/MAO catalysts achieve hydrogen evolution rates 2-3 times higher than state-of-the-art catalysts under similar conditions.

The turnover frequency (TOF) for each catalyst in the three dehydrogenation reactions was systematically investigated to elucidate the dependence of the site activity on the $CN_{Pt-Pt}$. HAADF-STEM clearly showed size distribution varied from atomically dispersed Pt to clusters and nanoparticles with increase in Pt loading. For each sample there is also inhomogeneity of Pt active sites. To address this, we applied a least-squares kinetic fitting approach to deconvolute the contributions from different Pt species and extract kinetic data for specific Pt species (Fig. 2c) (see calculation details in Supporting Information)[38–40].

In the 0.375Pt/MAO, 1Pt/MAO, and 3Pt/MAO catalysts, Pt was predominantly present as nanoparticles, enabling site-specific TOF analysis using the truncated octahedral model, which is commonly used to represent Pt nanoparticle geometry[41,42] and consistent with our HAADF-STEM observations (Figure S6). For each nanoparticle size, the populations of atoms on (111), (110), surface, edge, and corner sites were explicitly enumerated (see calculation details in Supporting Information). Catalytic activity was then normalized to the number of each type of site (Figure S7). Only when normalized to edge atoms did

the TOF remain essentially constant across different particle sizes, suggesting that edge Pt atoms serve as the primary active sites for dehydrogenation on Pt nanoparticles. This observation aligns with prior findings, further reinforcing the critical role of under-coordinated edge sites in facilitating dehydrogenation reactions[43].

For 0.02Pt/MAO, where Pt atoms exist exclusively as single-atom sites, the TOF was calculated by normalizing the catalytic activity to the total number of Pt atoms. This approach enabled a direct evaluation of the intrinsic activity of isolated Pt single atoms in dehydrogenation reactions. The remaining unknowns—the proportions of Pt single atoms, clusters, and nanoparticles, together the TOF for Pt clusters in the 0.05Pt/MAO and 0.15Pt/MAO catalysts—were determined using the equations in Fig. 2c through the least-squares fitting method (see calculation details in Supporting Information). The experimentally measured rates across multiple temperatures and dehydrogenation reactions were used as fitting inputs, with unknown parameters optimized by minimizing residual error, following validated least-squares kinetic deconvolution approaches in the literature[38,44]. The TOF of cluster in each catalyst was assumed to be constant regardless of the cluster size.

The resulting proportions of Pt species and kinetic parameters for the 0.05Pt/MAO and 0.15Pt/MAO catalysts are detailed in Table S6. The quality of the fitting and parameter identifiability were validated through residual analysis and normalized profile likelihood curves (Figures S8-9). For the 0.05Pt/MAO catalyst, Pt single atoms and clusters account for about 63% and 37% of the total Pt content,

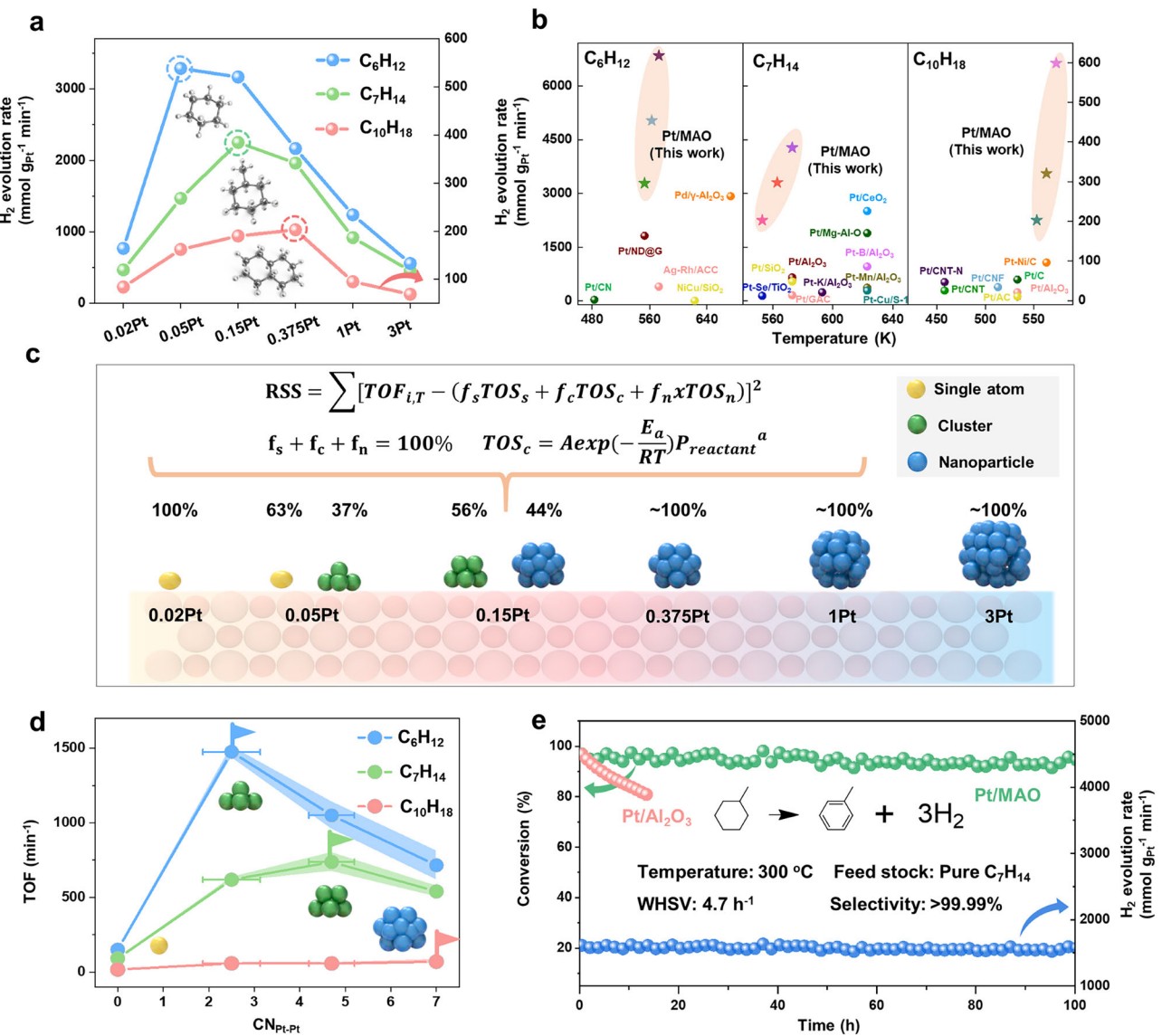

**Fig. 2 | Dehydrogenation performance of Pt/MAO catalysts. a** Hydrogen evolution rate of Pt/MAO catalysts for the dehydrogenation of cyclohexane, methylcyclohexane, and decalin at 280 °C. **b** Comparison of mass-specific activity of Pt/MAO catalysts with other advanced catalysts recently reported for the dehydrogenation of cyclohexane, methylcyclohexane, and decalin. **c** Schematic illustration of the dominant Pt structural motifs across different Pt loadings, as determined from global kinetic fitting. **d** Site-specific TOF as a function of $CN_{Pt\text{-}Pt}$ for different dehydrogenation reactions at 280 °C. Error bars represent 95% confidence intervals of the calculated $CN_{Pt\text{-}Pt}$, and shaded regions indicate 95% confidence intervals of the TOF values derived from uncertainty propagation. **e** Long-term stability of 0.15Pt/MAO for methylcyclohexane dehydrogenation at 300 °C.

respectively. In contrast, the 0.15Pt/MAO catalyst primarily comprises Pt clusters and nanoparticles, with proportions of 56% and 44%, respectively (Fig. 2c). These findings align with the HAADF-STEM results, confirming the successful decoupling of Pt species distributions. Subsequently, the TOF for Pt clusters in these catalysts was determined using kinetic data.

To achieve a statistically representative understanding of the Pt active site structures at atomic precision, the $CN_{Pt\text{-}Pt}$ for each Pt species was analyzed. For the 0.02Pt/MAO catalyst, where Pt exists primarily as single atoms, the $CN_{Pt\text{-}Pt}$ was determined to be 0. In contrast, for the 0.375Pt/MAO, 1Pt/MAO, and 3Pt/MAO catalysts, Pt was predominantly present as nanoparticles. We have shown that the edge Pt atoms on the nanoparticles are the main active sites, and therefore, the $CN_{Pt\text{-}Pt}$ of the active sites for these three catalysts is identical, with an edge Pt coordination number of 7 (Figure S10). For the 0.05Pt/MAO and 0.15Pt/MAO catalysts, based on quantitative EXAFS curve fitting (Figure S11, 12 and Table S7) and accounting for contributions from Pt single atoms and

nanoparticles[24], the $CN_{Pt\text{-}Pt}$ for the two types of clusters in the 0.05Pt/MAO and 0.15Pt/MAO catalysts was determined to be ~2.5 and ~4.7, respectively. Collectively, data fitting identified four representative active-site ensembles: atomically dispersed Pt ($CN_{Pt\text{-}Pt} = 0$), small clusters ($CN_{Pt\text{-}Pt} \approx 2.5$), medium clusters ($CN_{Pt\text{-}Pt} \approx 4.7$), and nanoparticles ($CN_{Pt\text{-}Pt}$ of edge Pt atom=7) (Fig. 2c). Intrinsic TOF and kinetic parameters were extracted for each ensemble, enabling the reliable use of $CN_{Pt\text{-}Pt}$ as a quantitative descriptor to systematically investigate the mechanistic and kinetic dependence of catalytic performance on Pt coordination.

Figure 2d summarizes the site-specific TOF as a function of the $CN_{Pt\text{-}Pt}$ for the dehydrogenation of cyclohexane, methylcyclohexane, and decalin, where the optimal $CN_{Pt\text{-}Pt}$ values for the three reactions were identified as ~2.5, ~4.7, and ~7.0, respectively. These findings are consistent with recent studies. For example, Ma et al. reported that fully exposed Pt ensembles with an average $CN_{Pt\text{-}Pt}$ of around 2 achieved optimal catalytic performance for cyclohexane dehydrogenation[23]. Our previous work demonstrated that in Pt

catalysts featuring particle sizes of 1.0–1.8 nm, edge-site Pt atoms with a $CN_{Pt-Pt}$ of 7 act as the predominant active centers for decalin dehydrogenation[43]. Collectively, these results highlight the importance of undercoordinated Pt sites, such as clusters or edge atoms within nanoparticles, as the preferred active sites for cycloalkane dehydrogenation. Furthermore, our results reveal a clear trend in which the optimal $CN_{Pt-Pt}$ increases with the molecular size of the cycloalkane, reflecting the influence of reactant molecular structure on the active site requirements.

To account for uncertainty, we propagated the 95% confidence intervals obtained from profile likelihood analysis (Figures S9) to the derived TOF and $CN_{Pt-Pt}$. The resulting error bars are shown in Fig. 2d. While the uncertainty in TOF is minimal, CN values exhibit broader variation. Importantly, the overall volcano-shaped trends and optimal CN ranges remain robust. To assess nanoparticle model sensitivity, we repeated the analysis using a cuboctahedron geometry. Although edge Pt remains the dominant active site, the lower edge fraction leads to higher apparent TOF of edge Pt and flatter volcano shapes, especially for methylcyclohexane dehydrogenation (Figure S13). The structural fractions also shift: for 0.15Pt/MAO, the fractions of Pt single atom, cluster and nanoparticle shift from 0/0.54/0.46 to 0.04/0.63/0.33, with the cluster $CN_{Pt-Pt}$ increasing from ~4.7 to ~5.2. Regardless of the specific model, both geometries represent idealized shapes, and real Pt nanoparticles may deviate from these forms, potentially exposing additional low-coordination or defect sites. Complementary techniques such as CO-DRIFTS[45], TPD[46], and EXAFS will be valuable in future studies to validate facet distributions and refine structural models.

To clarify the role of the support in the observed volcano-type relationship, Pt/Al$_2$O$_3$ catalysts were prepared with different loadings (0.02–1 wt%). HAADF-STEM revealed that Pt on Al$_2$O$_3$ dispersed as single atoms at 0.02Pt, but rapidly grew into clusters with nanoparticles at 0.05Pt, and aggregated predominantly into nanoparticles above 0.15Pt (Figure S14), in contrast to the higher dispersion maintained on MAO. Catalytically, Pt/Al$_2$O$_3$ exhibited significantly lower dehydrogenation activity than Pt/MAO across all particle sizes, underscoring the unique ability of MAO in stabilizing highly dispersed Pt and enhancing intrinsic activity via electronic modulation (Figure S15a). Nevertheless, Pt/Al$_2$O$_3$ catalysts still exhibited reactant-dependent volcano-type trends: decalin dehydrogenation peaked at 0.15Pt/Al$_2$O$_3$ (nanoparticle-dominated), methylcyclohexane at 0.05PtPt/Al$_2$O$_3$, closely resembling the 0.15Pt/MAO, while cyclohexane reached its optimum between 0.02 and 0.05Pt/Al$_2$O$_3$, aligning with the ~2.5 $CN_{Pt-Pt}$ optimum identified on MAO (Figure S15b). These results indicate that the reactant-dependent volcano relationship between $CN_{Pt-Pt}$ and activity is largely support-independent, but the ability of MAO to stabilize low-coordinated Pt clusters makes it uniquely suitable for systematically probing this relationship. Moreover, the pronounced differences in thermal stability, basicity, and metal-support interactions between MAO and Al$_2$O$_3$ may also influence Pt speciation and reactivity, and comparisons with additional supports in future work could help generalize these findings.

The long-term stability of the 0.15Pt/MAO catalyst in methylcyclohexane dehydrogenation was evaluated under industrially relevant conditions at 300 °C, notably lower than the conventional 350 °C benchmark (Fig. 2e). The catalyst sustains > 90% conversion and > 99.99% selectivity throughout a 100-h continuous operation with methane concentrations stably maintained at 30–40 ppm, well below the 100 ppm threshold specified for PEMFC applications[33]. Likewise, 1Pt/MAO exhibited similarly stable behavior (Figure S16). However, 0.05Pt/MAO showed gradual but reversible deactivation, as H$_2$ purging restored its initial activity (Figure S16), excluding irreversible sintering as the main cause. Instead, deactivation is mainly attributed to surface poisoning caused by strong adsorption of products and slight coke formation on low-coordinated Pt sites, which can be removed by hydrogen-assisted desorption or hydrogenation[47]. In-situ FTIR

confirmed this mechanism, showing gradual strengthen of toluene-related C = C stretching bands (1436 and 1608 cm$^{-1}$) during reaction and their disappearance upon H$_2$ treatment (Figure S17a-b). Additionally, CO-DRIFTS before and after stability testing showed a slight shift in the CO adsorption peak (Figure S17c), suggesting that partial migration of Pt single atoms to clusters or subsurface sites may also contribute to the observed deactivation.

In sharp contrast, 0.15Pt/Al$_2$O$_3$ suffered rapid deactivation, with conversion dropping from ~100% to ~80% within 15 h. The robust stability of Pt/MAO can be attributed to two key factors: oxygen vacancy-mediated strong metal-support interactions in the defect-engineered MAO, which effectively suppress Pt species sintering during prolonged reactions (Figure S18), and the excellent coke resistance, as evidenced by TG and Raman analysis of the spent catalyst revealing negligible carbon deposition after 100 h (Figure S19, 20). Importantly, the ability to operate at such low temperatures enables a paradigm shift in thermal management, replacing energy-intensive natural gas combustion heating with efficient heat-transfer oil circulation, thereby drastically increasing the energy efficiency in industrial settings[16]. The superior low-temperature activity coupled with exceptional operational stability highlights Pt/MAO as a transformative catalyst for sustainable cycloalkane dehydrogenation processes.

## Mechanistic exploration

We performed a kinetic study of cycloalkane dehydrogenation over Pt/MAO catalysts to systematically examine how Pt active sites and cycloalkane molecular structure govern dehydrogenation activity and elucidate the underlying mechanism. Figure S21, 22 show the Arrhenius and Eyring plots for cyclohexane, methylcyclohexane, and decalin dehydrogenation over various Pt/MAO catalysts. The results clearly indicated that catalytic activity does not scale with Pt loading, but instead aligning with the volcano-shaped dependence on $CN_{Pt-Pt}$ (Fig. 2d). Based on kinetic data and the contributions of different Pt species in various catalysts, the apparent activation energy ($E_{app}$) and entropy change of the transition state ($\Delta S^{\ddagger}_{app}$) for the three reactions were calculated and plotted as a function of $CN_{Pt-Pt}$ in Fig. 3a. The $E_{app}$ values for Pt nanoparticle sites during methylcyclohexane dehydrogenation (83.9 kJ·mol$^{-1}$) are in close agreement with previous reports for comparable catalysts (~ 80 kJ·mol$^{-1}$)[48]. Both $E_{app}$ and $\Delta S^{\ddagger}_{app}$ display a CN-dependent volcano-type trend, suggesting a mechanistic transition as $CN_{Pt-Pt}$ changes. Notably, the CN-dependence of $E_{app}$ is not a simple inverse reflection of the TOF volcano, as it reflects both the intrinsic barrier of the rate-determining step (RDS) and coverage-dependent adsorption enthalpies of reactants and products (Figure S23).

According to Langmuir-Hinshelwood-Hougen-Watson kinetics, $E_{app}$ can be described as:

$$E_{app} = E_{rds} + (1 - 2\theta_R)\Delta H_R - 2\theta_P \Delta H_P \qquad (1)$$

where $E_{rds}$ denotes the intrinsic activation energy of RDS, $\theta R$ and $\theta P$ represent the coverage of reactant (R) and intermediates/products (P), and $\Delta HR$ and $\Delta HP$ correspond to their respective adsorption enthalpies. At low $CN_{Pt-Pt}$, $E_{rds}$ is intrinsically small, but strong product adsorption ($\Delta HP$ large, $\theta P$ high) makes desorption the dominant energetic penalty, limiting active site turnover despite the low intrinsic barrier. As $CN_{Pt-Pt}$ increases, $E_{rds}$ increases moderately, while product adsorption weakens ($\Delta HP$ decreases, $\theta P$ lower). Here, $E_{app}$ reflects the sum of C-H activation and product desorption barriers, which can be higher than in the low-$CN_{Pt-Pt}$ case, but the optimal balance between the two processes yields the maximum TOF. At high-$CN_{Pt-Pt}$ region, product adsorption is weak ($\Delta HP$ negligible, $\theta P$ small), so $E_{app}$ is dominated by $E_{rds}$ and may decrease slightly. However, the reactant coverage and adsorption strength are too low to sustain high activity, leading to a drop in TOF.

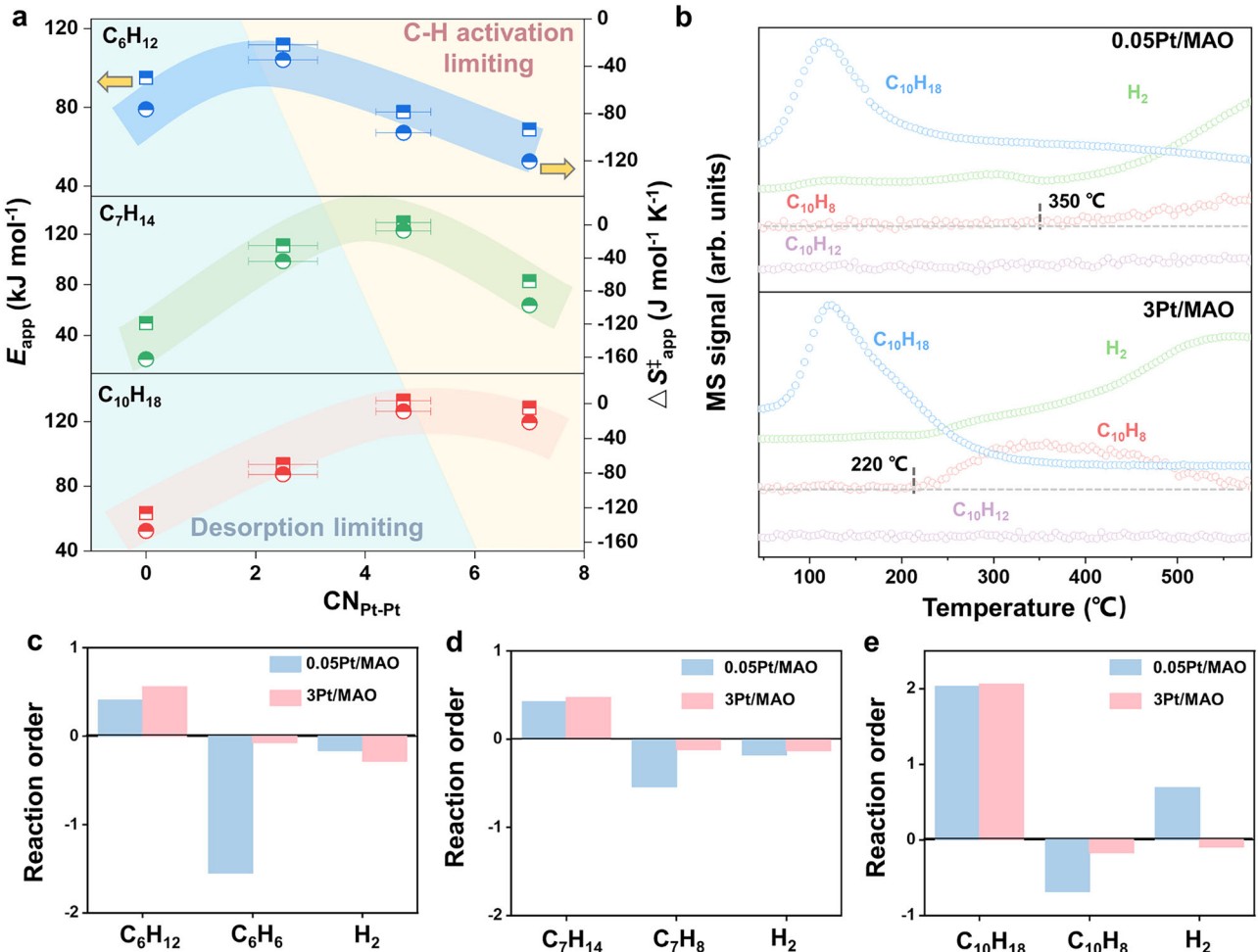

**Fig. 3 | Kinetic studies of cycloalkane dehydrogenation on Pt/MAO catalysts. a** Apparent activation energy (square) and entropy (circle) as a function of $CN_{Pt-Pt}$ for the dehydrogenation of cyclohexane, methylcyclohexane, and decalin. Error bars represent 95% confidence intervals of the calculated $CN_{Pt-Pt}$. Shaded regions indicate the general trends of these correlations. **b** TPSR profiles for decalin dehydrogenation on 0.05Pt/MAO and 3Pt/MAO catalysts. **c–e** Reaction orders of reactants, products and $H_2$ for the dehydrogenation of cyclohexane, methylcyclohexane, and decalin on 0.05Pt/MAO and 3Pt/MAO catalysts.

The $CN_{Pt-Pt}$ dependence of $\Delta S^{\ddagger}_{app}$ follows a similar trend to $E_{app}$, consistent with the enthalpy-entropy compensation effect[49]. Negative $\Delta S^{\ddagger}_{app}$ values at low $CN_{Pt-Pt}$ arise from that the transition state have fewer accessible degrees of freedom than the initial state on strongly bound surfaces. As $CN_{Pt-Pt}$ increases, adsorption weakens and $\Delta S^{\ddagger}_{app}$ becomes less negative. In the high-$CN_{Pt-Pt}$ regime, however, $\Delta S^{\ddagger}_{app}$ becomes more negative again, which may be attributed to multi-atom cooperative C-H activation at extended Pt ensembles, imposing greater configurational constraints on the transition state. Overall, these findings indicate that the volcano-shaped activity trend arises from the combined enthalpic and entropic contributions governed by coverage-dependent adsorption. The kinetics are thus best described as entropy-driven, with a mechanistic shift from desorption-limited at low $CN_{Pt-Pt}$ to C-H activation-controlled at higher $CN_{Pt-Pt}$. Notably, the negligible variation in $E_{app}$ among 0.375Pt/MAO, 1Pt/MAO, and 3Pt/MAO catalysts confirms their shared active-site configuration, which is a critical validation of Pt nanoparticle-dominated catalytic behavior (Figure S24).

To validate the mechanistic changes associated with $CN_{Pt-Pt}$, 0.05Pt/MAO and 3Pt/MAO were selected as representative catalysts for low- and high- coordinated Pt, respectively, and their reaction orders were measured (Figure S25-27). For all three dehydrogenation reactions, the negative reaction orders of the products indicate strong adsorption of unsaturated aromatic products, which inhibits the reaction (Fig. 3c-e). Within the LHHW framework, the apparent reaction orders of the reactant and product are expressed as:

$$n_R = 1 - 2\theta_R \quad (2)$$

$$n_P = -2\theta_P \quad (3)$$

Since reaction orders directly reflect surface coverages, more negative values correspond to stronger adsorption. Accordingly, the more negative product reaction order observed on 0.05Pt/MAO compared to 3Pt/MAO highlights the stronger adsorption on low-coordinated Pt and its greater inhibitory effect on activity[47]. To further validate the reaction order analysis, dehydrogenation rates were measured at high product concentrations (Figure S28), where the rate decrease caused by product inhibition was more pronounced for 0.05Pt/MAO. Furthermore, the higher reaction order of hydrogen for 0.05Pt/MAO compared to 3Pt/MAO indicates stronger adsorption of dehydrogenated intermediates on low-coordinated Pt. This behavior arises because low-coordinated Pt atoms stabilize dehydrogenation-derived intermediates more strongly, amplifying the thermodynamic driving force for reverse hydrogenation[47]. This mechanistic understanding aligns with our earlier stability tests, where 0.05Pt/MAO showed reversible deactivation due to strong product adsorption and was fully regenerated upon $H_2$ purging, which is not observed for 1Pt/MAO (Figure S16). Finally, the hydrogen reaction order shifts from

negative for cyclohexane dehydrogenation to positive for decalin dehydrogenation, revealing that dehydrogenated intermediates from decalin exhibit stronger adsorption than those from methylcyclohexane or cyclohexene.

Temperature-programmed surface reaction (TPSR) experiments using decalin dehydrogenation as a probe reaction were performed to investigate the reaction and desorption behaviors of reactants and products on 0.05Pt/MAO and 3Pt/MAO catalysts (Fig. 3b). Upon temperature ramping, a pronounced increase in the decalin signal was initially observed for both catalysts, corresponding to the desorption of pre-adsorbed decalin from the catalyst surfaces before dehydrogenation commenced. Subsequently, the $H_2$ signal exhibited a marked increase at approximately 240 °C, which coincided with the decline of the decalin signal, indicating comparable onset temperatures for the dehydrogenation reaction on both catalysts. Notably, the desorption profiles of naphthalene product revealed striking differences between the two catalysts. For 0.05Pt/MAO, naphthalene desorption occurred at substantially higher temperatures compared to 3Pt/MAO, suggesting stronger binding interactions between dehydrogenated intermediates and low-coordinated Pt active sites[43]. These experimental observations align with the earlier reaction order analysis, underscoring the critical role of product desorption kinetics in the catalytic performance over low-coordinated Pt sites.

DFT simulations were performed to investigate the three dehydrogenation reactions on Pt active sites with varying coordination numbers. $Pt_1$, $Pt_4$, and $Pt_{13}$ models were constructed on the $MgAl_2O_4$(111) surface to represent Pt species with coordination numbers of 0, ~2.5, and ~4.7, respectively, while a Pt(111) surface was used to model high-coordinated Pt (Figure S29). Upon optimization, strong Pt-MAO interactions induced partial flattening of $Pt_{13}$, reducing its $CN_{Pt-Pt}$ to 3.9. Despite this decrease, the $CN_{Pt-Pt}$ remains distinct from the other Pt models, enabling meaningful comparisons of catalytic properties across different $CN_{Pt-Pt}$ regimes. The possible dehydrogenation pathways for cyclohexane, methylcyclohexane, and decalin over $Pt_1$/MAO, $Pt_4$/MAO, $Pt_{13}$/MAO and Pt(111) were systematically studied (Figure S30-41). Cycloalkanes dehydrogenation involves adsorption on the surface via the σ-complexation, and activation of C-H bond via stepwise removal of hydrogen atoms to form partially dehydrogenated intermediates or polycyclic aromatic hydrocarbons, depending on reaction conditions.

For all three dehydrogenation reactions, the adsorption energies consistently follow the order: $Pt_1$/MAO <$Pt_4$/MAO <$Pt_{13}$/MAO <Pt(111) (Fig. 4a-c). The results clearly demonstrate that the adsorption strength of reaction intermediates gradually weakens as the $CN_{Pt-Pt}$ increases (Figure S42). For all three reactions, the step with the highest free energy in the energy profile was identified as the RDS, corresponding to either TS5 or TS6, depending on the specific reaction pathway[50]. This suggests that the transition state of the RDS is an unsaturated hydrocarbon species, analogous to the aromatic product. The lowest free energy, indicative of the highest reaction rate[51], was observed on $Pt_4$/MAO for cyclohexane dehydrogenation, whereas for methylcyclohexane, $Pt_4$/MAO and $Pt_{13}$/MAO exhibited similar activity. Interestingly, for decalin dehydrogenation, the highest activity was observed on Pt(111), aligning well with the experimental observation that the $CN_{Pt-Pt}$ corresponding to peak catalytic activity increases with the molecular size of the cycloalkane.

On low-coordinated Pt species, the strong adsorption of unsaturated intermediates not only reduces the availability of active sites for reactant adsorption but also inhibits subsequent C-H activation. Interestingly, these low-coordinated Pt active sites break the conventional scaling relationship between intermediate adsorption stability and activation energy. Typically, stronger adsorption facilitates C-H activation by weakening the C-H bond, thereby lowering the activation energy. However, DFT calculations reveal that while $Pt_1$/MAO ($CN_{Pt-Pt}$ = 0) exhibits high adsorption strength, it also presents an

unexpectedly high activation energy. For instance, in the dehydrogenation of cyclohexane over $Pt_4$/MAO, TS5 was identified as the RDS. Reaction configuration analysis (Fig. 4d) shows that the adsorbed $^*C_6H_8$ species are strongly bound to the Pt atom, restricting the activation of C-H bonds positioned further from the active site[52]. To facilitate subsequent C-H activation, $^*C_6H_8$ must undergo rotational movement, leading to a high activation energy despite its strong adsorption. This effect is corroborated by the near-zero $\Delta S^{\ddagger}$ values in Fig. 3a, indicating that the transition states involve significant rotational or translational entropy contributions[53,54]. The instability of the transition state for the dehydrogenated species on single-site Pt catalysts further contributes to this anomaly (Figure S43), increasing the activation energy and ultimately reducing catalytic efficiency. A similar phenomenon was observed for methylcyclohexane dehydrogenation on $Pt_{13}$/MAO (Fig. 4e). Nevertheless, this effect proves beneficial in inhibiting demethylation by either blocking methyl group access to Pt sites or requiring unfavorable molecular motions at the transition state (Figure S44), explaining the high selectivity and excellent anti-coking properties of 0.15Pt/MAO during long-term methylcyclohexane dehydrogenation. In contrast, highly coordinated Pt species such as Pt(111) exhibit weaker adsorption strength (Fig. 4f). While this reduced adsorption facilitates product desorption, the relatively high C-H activation energy at each step results in an overall increase in the reaction energy barrier. Consequently, Pt species with intermediate $CN_{Pt-Pt}$ achieve a balance between moderate adsorption strength and efficient C-H activation, leading to optimal catalytic performance.

It is crucial to emphasize that the $CN_{Pt-Pt}$ optimization for the dehydrogenation of different cycloalkanes is governed by electronic effects rather than geometric factors. Unlike the ensemble size hypothesis—which focuses on the geometric arrangement of active sites required to accommodate specific reaction intermediates—this system exhibits electronic structure-dominated behavior, primarily through modulation of Pt d-band characteristics. Figure 5a and Figure S45a shows that the Pt d-band center shifts toward the Fermi level as the $CN_{Pt-Pt}$ decreases, following the sequence: Pt(111) < $Pt_{13}$/MAO <$Pt_4$/MAO <$Pt_1$/MAO. Accompanying this trend is a corresponding increase in the fraction of unoccupied 5 d states above the Fermi level with decreasing $CN_{Pt-Pt}$. These observations are consistent with the Pt $L_{III}$-edge XANES spectra results (Fig. 1c), where the absorption edge moves to higher energy in the order 0.02Pt/MAO > 0.05Pt/MAO > 0.15Pt/MAO. Quantitative analysis of white-line intensities, under the assumption of a linear relationship with 5d-orbital vacancies, confirms this trend, with XANES-derived vacancy counts monotonically increasing as $CN_{Pt-Pt}$ decreases (Figure S45b).

From an orbital interaction perspective, the upshifted d-band center narrows the energy gap between Pt d-orbitals and adsorbate antibonding orbitals (e.g., π* for cycloalkenes). This enhances d-π* back-donation, increasing bonding-state occupancy and decreasing antibonding-state occupancy, thereby strengthening Pt-adsorbate interactions[55]. Such enhanced d-π* back-donation explains the stronger adsorption observed on low-coordinated Pt sites. Consistently, the calculated C-H activation barriers of partially dehydrogenated intermediates (e.g., $C_6H_{10}$, $C_7H_{12}$, and $C_{10}H_{16}$; Figure S42) generally follow the expected decreasing trend with stronger d-π* back-donation, with minor deviations attributed to steric and conformational effects. To examine possible side effects, we further calculated C-C cleavage barriers for partially dehydrogenated intermediates on Pt(111), $Pt_{13}$/MAO and $Pt_1$/MAO (Figure S46). While enhanced d-π* back-donation lowers the barriers on all sites, the reduction is most pronounced on $Pt_1$/MAO, where the C-C cleavage barriers become comparable to or even lower than those for C-H activation (Figure S47). This result suggests that Pt single atoms with extremely low coordination tend to promote side reactions such as C-C scission, which may lead to coke formation and catalyst deactivation.

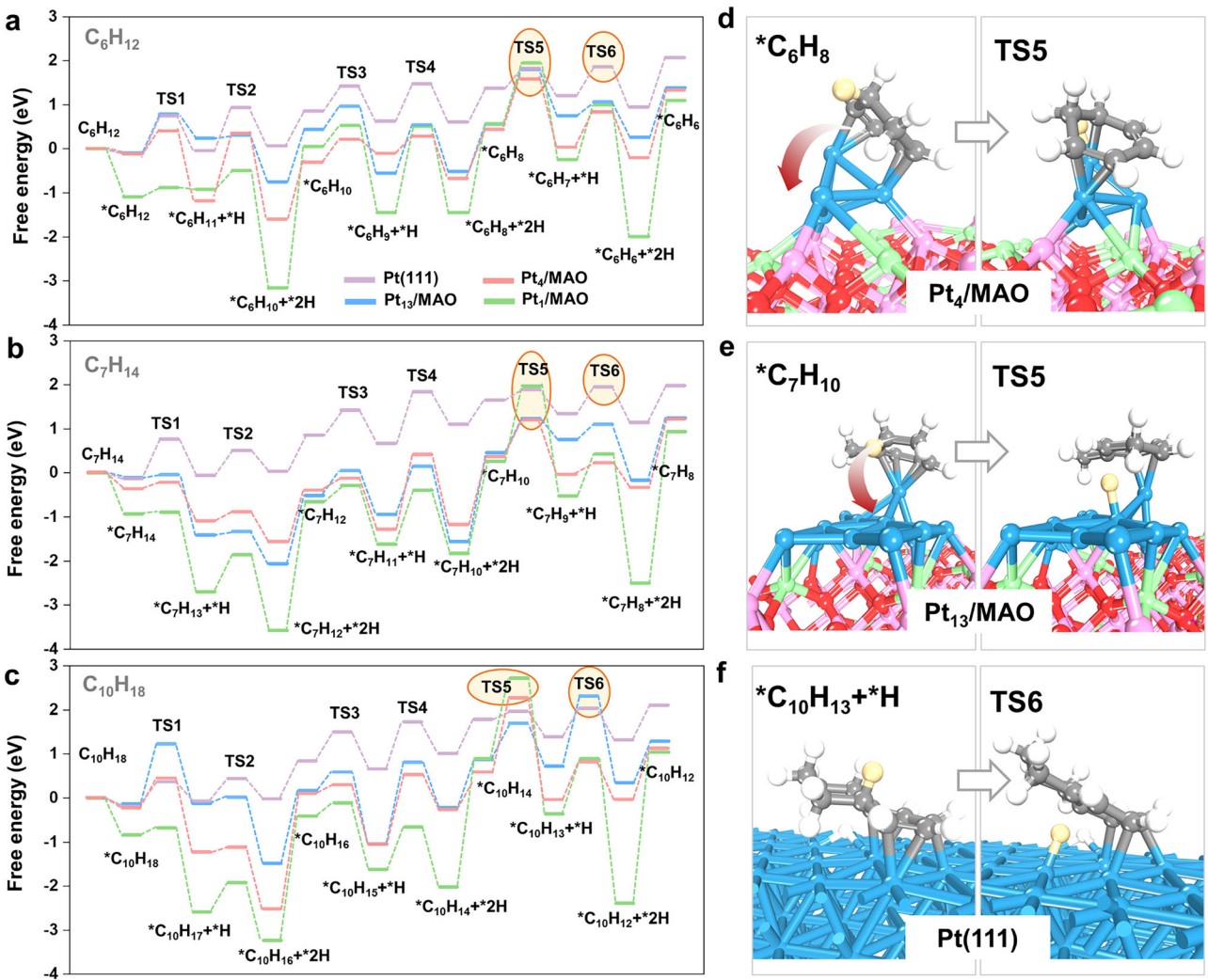

**Fig. 4 | DFT calculations of the Pt/MAO-catalyzed cycloalkane dehydrogenation process. a–c** Reaction energy diagrams for the dehydrogenation of cyclohexane, methylcyclohexane, and decalin, with RDS highlighted in circles. **d–f** Structural representations of the rate-determining transition states on the most active catalyst mode for the dehydrogenation of cyclohexane, methylcyclohexane, and decalin. Color code: Pt, blue; O, red; Al, pink; Mg, green; C, gray; H, white.

The reaction energy profiles suggest that the optimal active sites for cyclohexane, methylcyclohexane, and decalin dehydrogenation are $Pt_4$/MAO, $Pt_{13}$/MAO, and Pt(111), respectively. A comparison of the adsorption energies of dehydrogenated products shows that larger molecules exhibit stronger adsorption on the same Pt active sites (Figure S42). This trend indicates that dehydrogenation reactions involving intermediates with larger carbon numbers require higher-coordinated Pt species to achieve the optimal adsorption energy. In summary, based on mechanistic analysis, we elucidate the sensitivity of Pt active sites to cycloalkane structure in cycloalkane dehydrogenation (Fig. 5b). For different Pt species within the same reaction, reactant adsorption gradually weakens as Pt coordination increases. On low-coordinated Pt, the strong adsorption of dehydrogenated species significantly impacts the reaction, limiting active site availability and product desorption. Conversely, on high-coordinated Pt, weaker reactant adsorption shifts the limiting factor to C-H activation. The resulting volcano-shaped activity trend reflects that moderate Pt coordination provides balanced adsorption and C-H activation for optimal catalytic activity. Furthermore, for dehydrogenation of different cycloalkanes on the same Pt species, larger cycloalkane molecules strengthen adsorption, which shifts the optimal Pt coordination to higher values, ultimately revealing that distinct cycloalkanes require different $CN_{Pt-Pt}$ for optimal dehydrogenation.

To probe the molecular-level basis of these trends, we examined the molecular structures of their dehydrogenation products: benzene ($C_6H_6$), toluene ($C_7H_8$), and naphthalene ($C_{10}H_8$) (Figure S48). In toluene, the electron-donating methyl group elongates the adjacent C-H bonds, increasing electron density of neighboring carbons. In naphthalene, conjugation of the fused bi-ring system similarly elongates the four α-position C-H bonds near the ring junction, again leading to higher electron density at these positions. Higher carbon electron density stabilizes the π* antibonding orbitals by lowering their energy, as increased electron density screens the nuclear potential and reduces the effective energy of π-electrons[56]. This trend is reflected in the computed LUMO energies, which decrease progressively from benzene to toluene to naphthalene (Fig. 5c). Notably, partially dehydrogenated intermediates ($C_6H_8$, $C_7H_{10}$, and $C_{10}H_{14}$; $C_6H_{10}$, $C_7H_{12}$, and $C_{10}H_{16}$) follow the same trend (Figure S49), confirming the general applicability of LUMO energy as a reliable electronic descriptor. Bader charge analysis further supports this relationship, showing that molecules with higher carbon electron density exhibit lower LUMO energies (Figure S50).

For adsorption on Pt surfaces, lower LUMO energies (π* levels) reduce the energy gap between the π* orbitals and the Pt d-band, facilitating stronger d-π* back-donation and stabilizing Pt-adsorbate interactions[57]. In such cases, tuning the Pt d-band center downward

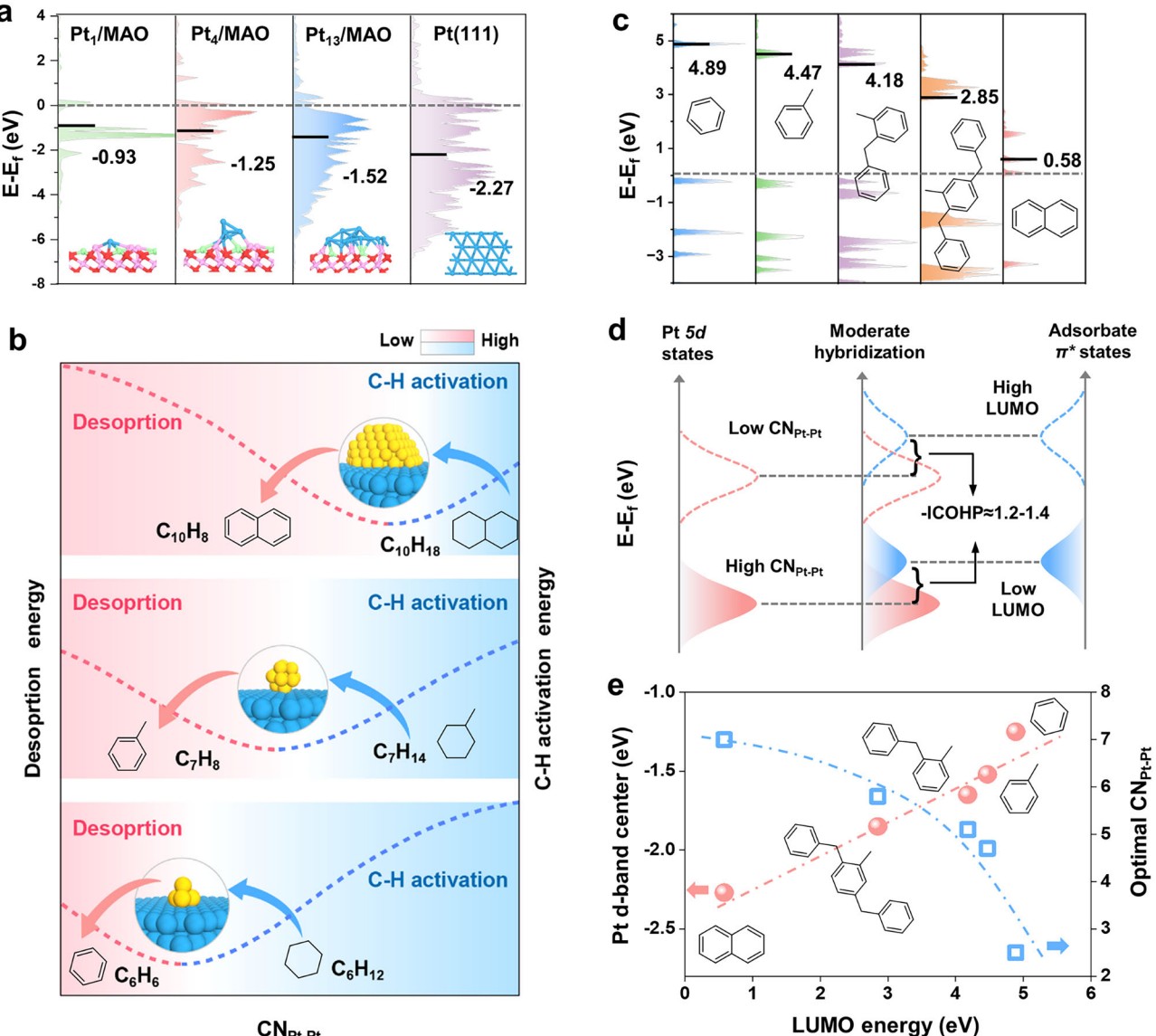

**Fig. 5 | Generalizing the volcano curves of Pt active sites for cycloalkane dehydrogenation. a** Projected density of states of Pt 5d-orbitals with band center positions. **b** Schematic representation illustrating the balance between C-H activation energy and product desorption energy for different cycloalkane dehydrogenation reactions. **c** Projected density of states and LUMO energy of aromatics. **d** Mechanistic illustration of $CN_{Pt-Pt}$ optimization guided by Pt-adsorbate orbital hybridization. **e** Correlation of Pt d-band center and optimal $CN_{Pt-Pt}$ with LUMO energy of aromatic product in cycloalkane dehydrogenation.

mitigates overly strong adsorption by restoring better energetic alignment with low-LUMO aromatics (Fig. 5d). Consequently, LUMO energy effectively captures the electronic requirements for optimal d-π* orbital alignment, enabling targeted tuning of the Pt d-band center to optimize orbital energy alignment with the aromatic adsorbate. In practice, distinct cycloalkane structures will demand specific $CN_{Pt-Pt}$ to achieve the desired adsorption strength and catalytic activity. These insights enable us to use LUMO energy as a molecular descriptor to locate the optimal $CN_{Pt-Pt}$ for catalytic performance, as depicted in Fig. 5e. Additionally, the correlations of LUMO energies for reactants and intermediates with optimal Pt d-band center were analyzed (Figure S51a). While reactants show similarly high LUMO levels, the dehydrogenated species, particularly the products exhibit significantly lower and more variable LUMO energies, indicating stronger π* orbital interactions as dehydrogenation proceeds. Although ΔLUMO values were also analyzed (Figure S51b), the trends are predominantly governed by product LUMOs. Thus, product LUMO correlates most strongly with the optimal Pt d-band center, highlighting the dominant

role of d-π* orbital interactions in governing adsorption strength and thus activity.

To further quantify this interaction, crystal orbital Hamilton populations (COHP) analysis of Pt-C bonds was performed for reactants adsorption, C-H activation, intermediates adsorption, and products adsorption (Figure S52). The results show that -ICOHP increases with decreasing Pt coordination, consistent with stronger covalent interactions from enhanced d-π* back-donation. Optimal catalytic performance across the different reactions is achieved when -ICOHP falls within an intermediate range, representing a balance between sufficient stabilization for C-H bond activation and facile product desorption. Moreover, the optimal-ICOHP range varies among the elementary steps, reflecting their distinct Pt-C bonding requirements. Specifically, the most effective Pt-C interactions for the C-H activation and product adsorption steps are found within -ICOHP ranges of around 1.4−2.1 and 1.2−1.4, respectively.

To validate this mechanism, we extended our investigation to two industrially important liquid organic hydrogen carriers: benzyltoluene

and dibenzyltoluene. The presence of benzyl substituents increases the electron density of aromatic rings relative to toluene, as reflected by the larger fraction of elongated C-H bonds, albeit less pronounced than in naphthalene (Figure S48). LUMO energy calculations yield average values of 4.24 eV for benzyltoluene isomers (2-, 3-, 4-) (Figure S53) and 2.85 eV for dibenzyltoluene isomers (2,3-, 2,4-, and 2,5-) (Figure S54), higher than in toluene but lower than in naphthalene. DFT calculations further showed that the adsorption energies of benzyltoluene isomers (-0.77 to -0.89 eV) and dibenzyltoluene isomers (-0.82 to −1.03 eV) on Pt(111) fall within a narrow range (Figure S55-56) and lie between those of toluene and naphthalene (Figure S57). Catalytic performance testing on Pt catalysts revealed that the optimal $CN_{Pt-Pt}$ for perhydro-benzyltoluene and perhydro-dibenzyltoluene dehydrogenation are approximately 5.1 and 5.8, respectively (Figure S58). As shown in Fig. 5e, these values fall precisely on the established correlation between optimal $CN_{Pt-Pt}$ and LUMO energy of aromatic product.

According to the Sabatier principle, optimal catalytic performance requires a moderate adsorption strength. As noted above, the π* antibonding character of transition states in cycloalkane dehydrogenation requires precise alignment with Pt d-band center. Through systematic quantification, we uncover a robust linear correlation between Pt d-orbital properties and LUMO energies of aromatics at the optimal $CN_{Pt-Pt}$ (Fig. 5e and Figure S59). These correlations mechanistically support the proposed framework in Fig. 5d, where maximum activity is achieved when the energy offset between aromatic π* orbitals and Pt d-orbitals is optimally tuned to yield moderate orbital hybridization. Importantly, these results establish LUMO energy as a universal descriptor for optimizing $CN_{Pt-Pt}$ in the dehydrogenation of various cycloalkanes. Nevertheless, structural factors such as ring number, substituents, and fusion topology may exert inherently complex influences, underscoring a significant direction for future refinement of the descriptor framework.

This study establishes a fundamental structure-activity relationship in Pt-catalyzed cycloalkane dehydrogenation, revealing that the optimal Pt-Pt coordination number ($CN_{Pt-Pt}$) of the active sites varies systematically with the molecular structure of the reactant. By engineering a coordination-tunable Pt/MgAl$_2$O$_4$ system and employing cyclohexane, methylcyclohexane, and decalin as model substrates, we identify volcano-type activity trends with optimal $CN_{Pt-Pt}$ values of ~2.5, ~4.7, and ~7.0, respectively. Mechanistic investigations demonstrate that this behavior arises from the balance between C-H activation and product desorption, both governed by orbital hybridization between Pt d-orbitals and aromatic π* orbitals. A key insight from this work is the discovery of LUMO energy of dehydrogenated aromatics, as a unifying molecular descriptor that directly correlates with the optimal $CN_{Pt-Pt}$. As LUMO energy decreases with increasing aromatic size and conjugation, metal-adsorbate interactions are strengthened, necessitating higher-coordinated Pt sites with lower d-band center to sustain efficient turnover. This principle not only explains the observed trends across multiple cycloalkanes, but also finds experimental validation in the dehydrogenation of perhydro-benzyltoluene and perhydro-dibenzyltoluene. By linking molecular electronic properties with coordination-specific catalytic performance, this work offers a predictive framework for rational catalyst design. The optimized Pt/MgAl$_2$O$_4$ catalyst demonstrates unprecedented low-temperature dehydrogenation activity and >100 h operational stability, setting a new benchmark for hydrogen storage and hydrocarbon conversion technologies. More broadly, the LUMO energy-guided design strategy opens a generalizable pathway toward reactant-specific engineering of catalytic active sites across diverse dehydrogenation systems.

## Methods
### Materials
H$_2$PtCl$_6$·6H$_2$O (AR grade, Pt≥37.5%), C$_2$H$_6$O (ethanol, AR grade), Mg(CH$_3$COO)$_2$·4H$_2$O (magnesium acetate tetrahydrate, AR grade), γ-

Al$_2$O$_3$ powder, and C$_{10}$H$_{18}$ (decalin, AR grade) were obtained from Sinopharm Chemical Reagent Co., Ltd., China. C$_9$H$_{21}$O$_3$Al (aluminum isopropoxide, AR grade) and C$_6$H$_{12}$ (cyclohexane, AR grade) were purchased from Shanghai Macklin Biochemical Co., Ltd. C$_7$H$_{14}$ (methylcyclohexane, ≥99%) was supplied by Shanghai Aladdin Biochemical Technology Co., Ltd. High-purity gases including N$_2$ (99.999%), Ar (99.999%), air, and H$_2$ (99.999%) were provided by Qingdao Xinke Gas Co., Ltd. Benzyltoluene (C$_{14}$H$_{14}$, 99%, mixture of isomers) and dibenzyltoluene (C$_{21}$H$_{20}$, 99%, mixture of isomers) was acquired from Adamas Reagent Co., Ltd. and subsequently hydrogenated in our laboratory to obtain perhydro-benzyltoluene (C$_{14}$H$_{26}$) and perhydro-dibenzyltoluene (C$_{21}$H$_{38}$). All aqueous solutions were prepared using deionized water (18.2 MΩ·cm) produced in our laboratory.

### Catalyst preparation
The MgAl$_2$O$_4$ support was synthesized using a hydrolysis method of aluminum isopropoxide with magnesium acetate tetrahydrate in ethanol. Briefly, 0.030 mol of aluminum isopropoxide and 0.015 mol of magnesium acetate tetrahydrate were dissolved in 50 mL of ethanol. The solution was transferred to an autoclave, where it was heated to 120 °C for 10 h, followed by an additional heating step at 160 °C for 10 h under continuous stirring at 500 rpm. After cooling to room temperature, the resulting precipitate was collected by centrifugation, thoroughly washed with deionized water and ethanol, dried at 80 °C for 12 h, and subsequently calcined at 700 °C for 5 h in a muffle furnace. The obtained sample was etched under a flow of 10 vol% H$_2$/Ar gas (50 mL min$^{-1}$) at 350 °C for 3 h.

Pt/MgAl$_2$O$_4$ and Pt/Al$_2$O$_3$ catalysts were prepared using the incipient wetness impregnation method. In a typical synthesis, an ethanol solution of H$_2$PtCl$_6$ was added dropwise in the appropriate amount to 1 g of MgAl$_2$O$_4$ support. The mixture was aged at room temperature for 12 h and subsequently dried at 80 °C overnight. The dried sample was reduced under a flow of 10 vol% H$_2$/Ar gas (50 mL min$^{-1}$) at 300 °C for 3 h. To obtain Pt species of varying sizes, different Pt loadings (0.02 wt.%, 0.05 wt.%, 0.15 wt.%, 0.375 wt.%, 1 wt.%, and 3 wt.%) were employed, and the resulting catalysts were designated as 0.02Pt/MAO, 0.05Pt/MAO, 0.15Pt/MAO, 0.375Pt/MAO, 1Pt/MAO, and 3Pt/MAO, respectively.

### Catalyst characterization
Inductively coupled plasma optical emission spectrometry (ICP-OES) was performed using an Agilent ICPOES-730 spectrometer to determine the elemental composition. High-resolution transmission electron microscopy (HRTEM) images were obtained using a JEM-2100 UHR microscope, while high-angle annular dark-field scanning transmission electron microscopy (HAADF-STEM) images were recorded with a Tecnai G2 F20 S-Twin instrument. Additionally, aberration-corrected HAADF-STEM (AC-HAADF-STEM) images were acquired using a JEOL ARM200F FEG microscope equipped with a probe corrector, operating at 80 kV. X-ray diffraction (XRD) patterns were collected on a Philips X'pert Pro MPD super diffractometer using Cu Kα radiation (λ = 1.5418 Å). Nitrogen adsorption-desorption isotherms were measured at 77 K with a Micromeritics ASAP 2020 analyzer to determine Brunauer-Emmett-Teller (BET) surface area and pore properties. Thermogravimetric (TG) analysis was carried out using a TG-DTA set-sys-evolution instrumentation of PerkinElmer Diamond. Raman spectra were collected using a LabRAM HR Evolution spectrometer with a 325 nm laser excitation source.

CO diffuse reflectance infrared Fourier transform spectroscopy (CO-DRIFTS) was conducted using a Nicolet IS20 FTIR spectrometer equipped with a mercury-cadmium telluride (MCT) detector. Prior to testing, the catalysts were pre-reduced at 300 °C for 3 h under a 5 vol% H$_2$/Ar gas flow. After reduction, the system was cooled to 25 °C under an Ar purge. A 5 vol% CO/Ar gas flow was then introduced into the reaction cell for CO adsorption, followed by switching the flow to pure

Ar (30 mL min⁻¹) to remove excess CO. The spectra were continuously recorded at 25 °C. In-situ FTIR measurements were conducted to monitor toluene adsorption behavior during methylcyclohexane dehydrogenation. The catalyst pre-reduced following the same procedure described above was purged with $N_2$ at 300 °C for 20 min to remove residual hydrogen species. A background spectrum was first collected before introducing methylcyclohexane into the $N_2$ carrier gas. Spectra were then recorded at 3 min intervals for 60 min. Subsequently, the reaction gas was switched to $H_2$ for purging over 60 min, during which spectra were continuously collected at 3 min intervals.

XAS measurements at the Pt $L_{III}$-edge were performed in transmission mode at the SSRF BL17B1 beamline using Si(111) channel-cut crystals as the monochromator. The storage ring operated at an energy of 2.5 GeV with an electron current below 200 mA. The X-ray absorption near-edge structure (XANES) and extended X-ray absorption fine structure (EXAFS) data were processed and analyzed using Athena software. For Wavelet Transform analysis, the $\chi(k)$ exported from Athena was imported into the Hama Fortran code. The parameters were listed as follow: R range, 0–6 Å, k range, 0–12.0 Å⁻¹; k weight, 2; and Morlet function with $\kappa = 10$, $\sigma = 1$ was used as the mother wavelet to provide the overall distribution.

Temperature-programmed surface reaction (TPSR) experiments were conducted in a quartz-bed reactor coupled with mass spectrometry (LC-D200M). 100 mg of catalyst was packed into an 8 mm diameter quartz reactor and activated under 10 vol% $H_2$/Ar gas flow at 300 °C for 3 h. The catalyst bed was then cooled to 45 °C in $N_2$ atmosphere and maintained under isothermal conditions. Decalin vapor was introduced using a $N_2$-bubbling saturator, with the adsorption stabilization determined by monitoring the m/z = 67 ($C_{10}H_{18}$) signal intensity. The temperature-programmed reaction was initiated upon signal stabilization, implementing a linear heating ramp of 10 °C min⁻¹ from 45 °C to 600 °C. The intensity variations of multiple characteristic signals were synchronously recorded: m/z = 2 ($H_2$), m/z = 67 ($C_{10}H_{18}$), m/z = 104 ($C_{10}H_{12}$), and m/z = 128 ($C_{10}H_8$).

## Catalytic reaction

The cycloalkane dehydrogenation reactions were conducted in a quartz-bed flow reactor. 100 mg of catalyst was loaded into an 8 mm diameter quartz tube and reduced in a 10 vol% $H_2$/Ar gas flow (50 mL min⁻¹) at 300 °C for 3 h. After reduction, the gas flow was switched to $N_2$ (30 mL min⁻¹). Cycloalkane reactants—cyclohexane (268.2 µL min⁻¹), methylcyclohexane (369.3 µL min⁻¹), decalin (191.8 µL min⁻¹), perhydro-benzyltoluene (60.42 µL min⁻¹) or perhydrodibenzyltoluene (285.6 µL min⁻¹)—were fed into the reactor via a peristaltic pump and preheated to 200 °C before entering the reaction zone. Dehydrogenation activity tests were performed at temperatures ranging from 270 °C to 300 °C. To ensure accurate kinetic measurements and turnover frequency (TOF) calculations, the reactant conversion was controlled below 20%. For long-term stability evaluations, pure methylcyclohexane (211 µL min⁻¹) was fed over 1.4 g of 0.05Pt/MAO, 2 g of 0.15Pt/MAO, 2 g of 1Pt/MAO, and 2 g of 0.15Pt/$Al_2O_3$ catalysts.

Gas-phase products were analyzed using an online gas chromatograph (Fuli GC9790II) equipped with both a flame ionization detector (FID) and a thermal conductivity detector (TCD). Liquid-phase products were analyzed using an Agilent 8890 gas chromatograph equipped with an FID and a PEG-20M capillary column. The conversion ($X_{reactant}$), selectivity ($S_{product}$), and hydrogen evolution rate ($r_{H_2}$) were calculated using the following equations:

$$X_{reactant}(\%) = \frac{F_{reactant,in} - F_{reactant,out}}{F_{reactant,in}} \times 100\% \quad (4)$$

$$S_{product}(\%) = \frac{F_{product,out}}{F_{reactant,in} - F_{reactant,out}} \times 100\% \quad (5)$$

$$r_{H_2} = \frac{F_{H_2,out}}{m_{Pt}} \quad (6)$$

where $F_{reactant}$, $F_{product}$ and $F_{H_2}$ represent the molar flow rates of the reactant, product and $H_2$, respectively, and $m_{Pt}$ is the Pt mass in the applied catalyst.

## Calculation of Pt species fractions and TOF in catalysts

The TOF for Pt single atoms, clusters, and nanoparticles in the catalysts was calculated based on their respective contributions to the overall catalytic activity. The method for determining the TOF of each species is outlined below:

For the 0.02Pt/MAO catalyst, all Pt atoms are assumed to exist as single-atom active sites. The TOF of Pt single atoms ($TOS_s$) is calculated as:

$$TOS_s = \frac{F_{H_2,0.02Pt/MAO}}{m_{Pt}} \quad (7)$$

where $F_{H_2,0.02Pt/MAO}$ is the molar flow rates of the $H_2$ for 0.02Pt/MAO, and $m_{Pt}$ denotes the Pt mass in the 0.02Pt/MAO catalyst.

For the 0.375Pt/MAO, 1Pt/MAO, and 3Pt/MAO catalysts, Pt is predominantly present as nanoparticles. The truncated octahedral model was applied to identify the dominant active sites. Given the measured particle diameter $d$, Eq. 8 yields the total number of Pt atoms per particle ($N_t$):

$$d = 1.105 \times d_{atom} \times N_t^{1/3} \quad (8)$$

where $d_{atom}$ is the diameter of a Pt atom (0.276 nm). Then the number of atoms along the particle edge (n) can be solved based on:

$$N_t = 16n^3 - 33n^2 + 24n - 6 \quad (9)$$

After which the numbers of site-specific atoms were calculated as follows:

$$N_s = 30n^2 - 60n + 32 \quad (10)$$

$$N_c = 24 \quad (11)$$

$$N_e = 36(n - 2) \quad (12)$$

$$N_{100} = 6(n - 2)^2 \quad (13)$$

$$N_{111} = 8(3n^2 - 9n + 7) \quad (14)$$

where $N_s$, $N_c$, $N_e$, $N_{100}$, and $N_{111}$ represent the numbers of surface, corner, edge, (100), and (111) Pt atoms, respectively. Using the mole number of Pt in each catalyst, the absolute numbers of Pt atoms at different sites were obtained. TOF was then calculated by normalizing catalytic activity to the number of each type of site. For direct comparison, TOF values were further normalized to the maximum TOF within each site category, as presented in Figure S7.

Only when normalized to edge atoms did the TOF remain essentially constant across different particle sizes, suggesting that edge Pt atoms serve as the primary active sites for cycloalkane

dehydrogenation on Pt nanoparticles. Consequently, the TOF of Pt nanoparticles ($TOS_n$) is identical across these catalysts and calculated as:

$$TOS_n = \frac{F_{H_2, Pt/MAO}}{m_{edgePt}} \tag{15}$$

where $F_{H_2, Pt/MAO}$ is the molar flow rates of the $H_2$ for 0.375Pt/MAO, 1Pt/MAO and 3Pt/MAO catalyst, and $m_{edgePt}$ is the mass of edge Pt atoms in these catalysts.

The TOF of Pt clusters ($TOS_c$) were derived using kinetic data from 0.05Pt/MAO and 0.15Pt/MAO catalysts. These catalysts contain mixtures of Pt single atoms, clusters, and nanoparticles, with unknown fractions and distinct cluster characteristics. To resolve the contributions of each species, a least-squares fitting approach was employed based on the following equations.

$$f_s + f_c + f_n = 100\% \tag{16}$$

where $f_s, f_c, f_n$ represent the mass fractions of Pt single atoms, clusters and nanoparticles in the 0.05Pt/MAO or 0.15Pt/MAO catalyst.

$$TOS_c = A\,exp\left(-\frac{E_a}{RT}\right)P_{reactant}{}^a \tag{17}$$

where $A$ denotes the pre-exponential factor, $P_{reactant}$ represents the partial pressures of the reactant (cyclohexane, methylcyclohexane, or decalin), $a$ is the reaction order with respect to the reactant. During kinetic measurements, the conversion was maintained <20% specifically to avoid complications from product inhibition effects. The reaction orders of reactant can be obtained from the reaction order results of 0.05Pt/MAO in Fig. 3c-e.

$$RSS = \sum [TOF_{i,T} - (f_s TOS_s + f_c TOS_c + f_n x TOS_n)]^2 \tag{18}$$

where $TOF_{i,T}$ is the experimentally measured TOF of 0.05Pt/MAO or 0.15Pt/MAO catalyst during the dehydrogenation of cyclohexane, methylcyclohexane, or decalin (denoted by $i$) at temperatures T (543, 553, 563 and 573 K). The parameter $x$ represents the mass fraction of edge Pt atoms within the Pt nanoparticles in the 0.05Pt/MAO or 0.15Pt/MAO catalysts. The model comprises 13 equations with 6 unknowns ($f_s, f_c, f_n, E_a, A$ and $x$), as listed below.

For 0.05Pt/MAO:

$$f_s + f_c = 100\%$$

$$f_s \times 545.39 + f_c \times e^{lnA - \frac{E_a}{R \times 543.15}} \times 0.7^{0.41} - 1874.85 = 0$$

$$f_s \times 767.69 + f_c \times e^{lnA - \frac{E_a}{R \times 553.15}} \times 0.7^{0.41} - 3284.58 = 0$$

$$f_s \times 1162.64 + f_c \times e^{lnA - \frac{E_a}{R \times 563.15}} \times 0.7^{0.41} - 5037.47 = 0$$

$$f_s \times 1616.21 + f_c \times e^{lnA - \frac{E_a}{R \times 573.15}} \times 0.7^{0.41} - 6848.69 = 0$$

$$f_s \times 385.25 + f_c \times e^{lnA - \frac{E_a}{R \times 543.15}} \times 0.7^{0.42} - 1093.49 = 0$$

$$f_s \times 467.71 + f_c \times e^{lnA - \frac{E_a}{R \times 553.15}} \times 0.7^{0.42} - 1469.04 = 0$$

$$f_s \times 583.75 + f_c \times e^{lnA - \frac{E_a}{R \times 563.15}} \times 0.7^{0.42} - 2655.11 = 0$$

$$f_s \times 699.10 + f_c \times e^{lnA - \frac{E_a}{R \times 573.15}} \times 0.7^{0.42} - 4275.61 = 0$$

$$f_s \times 59.40 + f_c \times e^{lnA - \frac{E_a}{R \times 543.15}} \times 0.5^{2.03} - 119.67 = 0$$

$$f_s \times 83.85 + f_c \times e^{lnA - \frac{E_a}{R \times 553.15}} \times 0.5^{2.03} - 162.03 = 0$$

$$f_s \times 102.46 + f_c \times e^{lnA - \frac{E_a}{R \times 563.15}} \times 0.5^{2.03} - 263.77 = 0$$

$$f_s \times 125.61 + f_c \times e^{lnA - \frac{E_a}{R \times 573.15}} \times 0.5^{2.03} - 339.33 = 0$$

For 0.15Pt/MAO:

$$f_s + f_c + f_n = 100\%$$

$$f_s \times 545.39 + f_c \times e^{lnA - \frac{E_a}{R \times 543.15}} \times 0.7^{0.41} + f_n \times x \times 2285.03 - 2263.38 = 0$$

$$f_s \times 767.69 + f_c \times e^{lnA - \frac{E_a}{R \times 553.15}} \times 0.7^{0.41} + f_n \times x \times 3170.67 - 3166.23 = 0$$

$$f_s \times 1162.64 + f_c \times e^{lnA - \frac{E_a}{R \times 563.15}} \times 0.7^{0.41} + f_n \times x \times 4224 - 4461.31 = 0$$

$$f_s \times 1616.21 + f_c \times e^{lnA - \frac{E_a}{R \times 573.15}} \times 0.7^{0.41} + f_n \times x \times 5492.56 - 5490.79 = 0$$

$$f_s \times 322.74 + f_c \times e^{lnA - \frac{E_a}{R \times 533.15}} \times 0.7^{0.42} + f_n \times x \times 1111.69 - 719.50 = 0$$

$$f_s \times 385.25 + f_c \times e^{lnA - \frac{E_a}{R \times 543.15}} \times 0.7^{0.42} + f_n \times x \times 1598.87 - 1245.60 = 0$$

$$f_s \times 467.71 + f_c \times e^{lnA - \frac{E_a}{R \times 553.15}} \times 0.7^{0.42} + f_n \times x \times 2582.31 - 2250.93 = 0$$

$$f_s \times 583.75 + f_c \times e^{lnA - \frac{E_a}{R \times 563.15}} \times 0.7^{0.42} + f_n \times x \times 3849.59 - 3307.76 = 0$$

$$f_s \times 59.40 + f_c \times e^{lnA - \frac{E_a}{R \times 543.15}} \times 0.5^{2.03} + f_n \times x \times 223.18 - 126.11 = 0$$

$$f_s \times 83.85 + f_c \times e^{lnA - \frac{E_a}{R \times 553.15}} \times 0.5^{2.03} + f_n \times x \times 394.10 - 189.87 = 0$$

$$f_s \times 102.46 + f_c \times e^{lnA - \frac{E_a}{R \times 563.15}} \times 0.5^{2.03} + f_n \times x \times 797.90 - 304.45 = 0$$

$$f_s \times 125.61 + f_c \times e^{lnA - \frac{E_a}{R \times 573.15}} \times 0.5^{2.03} + f_n \times x \times 1046.72 - 599.87 = 0$$

All equations were implemented in Python (PyCharm), and the parameters were obtained by performing least-squares fitting to minimize Eq. 18, yielding the optimized values of $f_s, f_c, f_n, E_a, A$ and $x$. This approach allows for the quantitative determination of the fractions of Pt species and their respective TOFs, providing insights into the contributions of Pt single atoms, clusters, and nanoparticles to the overall catalytic performance.

- 0.02Pt/MAO: predominantly atomically dispersed Pt ($CN_{Pt-Pt} \approx 0$).
- 0.05Pt/MAO: a mixture of atomically dispersed Pt (63%) and small clusters ($CN_{Pt-Pt} \approx 2.5$, 37%).
- 0.15Pt/MAO: coexistence of medium clusters ($CN_{Pt-Pt} \approx 4.7$, 56%) and Pt nanoparticles ($CN_{Pt-Pt} \approx 7$, 44%).

- 0.375Pt/MAO, 1Pt/MAO, and 3Pt/MAO catalysts: dominated by Pt nanoparticles ($CN_{Pt-Pt} \approx 7$).

To assess the identifiability of the fitted parameters, profile likelihood analysis was performed using the following definition:

$$PL(f) = \exp\left(-\frac{1}{2} \times \frac{RSS(f) - RSS_{min}}{\frac{RSS_{min}}{n-p}}\right) \tag{19}$$

where RSS(f) is the residual sum of squares when parameter $f_s, f_c$ or $f_n$ is fixed, $RSS_{min}$ is the minimum residual sum of squares obtained from the global optimization, n is the number of data, and p is the number of fitted parameters. This expression provides a normalized likelihood curve (maximum = 1), which enables us to estimate confidence intervals for each parameter by locating the range over which PL(f) > threshold (0.15 for 95% CI).

### Calculation of the activation energy and entropy change

The intrinsic rate constant ($k$) was calculated by TOF and reaction orders:

$$k = \frac{TOF}{P_{reactant}^a} \tag{20}$$

Based on the consistent reaction orders observed in Fig. 3c-e between low (0.05Pt/MAO) and high-coordinated Pt (3Pt/MAO), we approximated the orders for intermediate catalysts as follows: for 0.02-0.15Pt/MAO, we used values similar to 0.05Pt/MAO, while for 0.375-3Pt/MAO, we adopted values comparable to 3Pt/MAO. Notably, the measured reaction orders of reactant differ only slightly across these catalysts (0.41 to 0.56 for cyclohexane, 0.42 to 0.4 for methylcyclohexane, and 2.03 to 2.06 for decalin). The apparent activation energy ($Ea$) was calculated through the Arrhenius equation:

$$lnk = lnA - \frac{E_a}{RT} \tag{21}$$

Based on transition state theory, the rate constant is expressed as:

$$k = \frac{k_B T}{h} \exp\left[-\frac{\Delta H^{0*}}{RT}\right] \exp\left[\frac{\Delta S^{\ddagger}}{R}\right] \tag{22}$$

where $k_B$, $h$ and $\Delta H^{0*}$ were the Boltzmann constant, Planck constant and enthalpy of activation, respectively. Then the enthalpy change ($\Delta S^{\ddagger}$) can be obtained by the Eyring plots ($\ln \frac{k}{T}$ verus $\frac{1}{T}$) according to the equation

$$ln\frac{k}{T} = -\frac{\Delta H^{0*}}{RT} + ln\frac{k_B}{h} + \frac{\Delta S^{0*}}{R} \tag{23}$$

To evaluate the robustness of our analysis, we varied the reaction orders by ±50% and recalculated the activation entropy and volcano plots. The resulting changes in $\Delta S^{\ddagger}$ were minor and had negligible influence on the overall volcano shape or the optimal CN values (Figure S24), confirming that the fitted trends are insensitive to small deviations in reaction order.

### DFT calculations

Density Functional Theory (DFT) calculations were performed using the Vienna Ab initio Simulation Package (VASP 5.4.1). Structural optimizations and electronic energy calculations were performed using the projector augmented wave (PAW) method to describe the interaction between valence electrons and ionic cores, and the Perdew-Burke-Ernzerhof (PBE) functional for electron exchange-correlation energy. Prior to calculations, the cut-off kinetic energy and Brillouin

zone parameters were optimized and set to 400 eV and a $3 \times 3 \times 1$ k-point mesh, respectively. The convergence criteria for ionic relaxation were set to 0.03 eV Å$^{-1}$ for the force acting on each atom. Van der Waals effect was not corrected by the D3 method, for the adsorption energy calculated from D3 method was much deviated from the experimental values.

The $MgAl_2O_4$ surface model was constructed by cutting a $4 \times 2$ supercell along the (111) plane, exposing Mg and Al atoms to a 20 Å vacuum layer in the z-direction. The geometries of $Pt_1$, $Pt_4$, and $Pt_{13}$ deposited on the $MgAl_2O_4$(111) surface were optimized based on the minimal energy principle, resulting in the models $Pt_1$/MAO, $Pt_4$/MAO and $Pt_{13}$/MAO. The Pt(111) surface was modeled using a periodic $4 \times 4$ unit cells containing four atomic layers separated by a 12 Å vacuum layer. The adsorption behaviors of dehydrogenation intermediates on the catalyst surface models were simulated, and the adsorption energies were calculated using Eq. 24. Transition states for the dehydrogenation elementary steps were located using the dimer method, and activation energy barriers were calculated using Eq. 25. During the calculations, the uppermost two atomic layers were relaxed to simulate surface dynamics, while the remaining layers were fixed to represent the bulk catalyst.

$$E_{ads} = E_{adsorbate + Pt + support} - E_{adsorbate} - E_{Pt + support} \tag{24}$$

$$\Delta E_{reac} = E_{transitionstate} - E_{initialstate} \tag{25}$$

$$\Delta G = \Delta E_{ads} + \Delta E_{ZPE} - T\Delta S_{ads} \tag{26}$$

where $E_{adsorbate + Pt + support}$ is total energy of the system with adsorbate on the supported Pt catalyst, $E_{adsorbate}$ is energy of the adsorbate, $E_{Pt + support}$ is total energy of the supported Pt catalyst, $E_{transitionstate}$ is energy of the transition state, $E_{initialstate}$ is energy of the initial state, $\Delta E_{ZPE}$ is the zero point energy correction, and $T\Delta S_{ads}$ is the corresponding entropy correction (T is set to be 573 K).

## Data availability

The data related to the figures in the paper are provided as Excel files in Source data. Source data are provided with this paper. Data are available from the corresponding authors upon request. Source data are provided with this paper.

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

## Acknowledgements
This work was partially supported by the National Natural Science Foundation of China No. 22208374 (Y.T.) and 22578497 (Y.T.), the Excellent Youth Scientist Award Foundation of Shandong Province No. ZR2024YQ009 (Y.T.), the Distinguished Young Scholars of the National Natural Science Foundation of China No. 22322814 (X.F.), CNPC Innovation Found No. 2022DQ02-0607 (Y.T.), and the Fundamental Research Funds for the Central Universities No. 24CX07006A (Y.T.).

## Author contributions
Y.T. and J.Q. performed the catalyst preparation, characterization and catalytic tests. Y.T. performed the theoretical simulation and wrote the manuscript. H.S. and Q.L. performed the XANES and EXAFS characterization and analyzed the data. B.W. participated in the characterization, catalysis and discussion. H.G. and D.Y. provided help with discussion on the catalyst structure. X.F. and D.C. designed the study, analyzed the data and wrote the paper. All authors discussed the results and improved the paper.

## Funding
 Olavs Hospital - Trondheim University Hospital).

## Competing interests
The authors declare no competing interests.
