## [Transparent Peer Review file · Nature Communications]

Reactant-Dependent Volcano Trends in Pt-Catalyzed Cycloalkane Dehydrogenation: Orbital Hybridization-Guided Design of Active Sites

Corresponding Author: Professor De Chen

Version 0:

Reviewer comments:

Reviewer #1

(Remarks to the Author)

This paper reports the synthesis of a series of Pt catalysts on MAO ranging from single-atom, cluster, to nanoparticles, their catalytic application to cycloalkane dehydrogenation, and fundamental study on the intrinsic effects of the Pt and the molecular sizes on the kinetics and electronic theory. The authors have thoroughly done the investigation by finely distinguishing many kinetic and electronic parameters, establishing a novel model of rationale on cycloalkane dehydrogenation. Besides, the developed catalyst exhibited a remarkably high catalytic performance compared with reported ones. Since this paper has a high impact on catalysis chemistry and is trying to provide significantly important insights for researchers on the relevant field, the reviewer evaluates this paper is potentially suitable for publication in Nature Communications. However, there are many concerns about scientific reliability and unclear points that need modification and reconsideration prior to further consideration for publication. The following points must be addressed for further peer-review.

1. Although high resistance to sintering was attributed to the high catalyst stability in the long-term test of MCH dehydrogenation, it is not likely because sintering is not the primary reason for deactivation at this low temperature region, but coke accumulation should be the more prominent reason. As widely concerned in MCH dehydrogenation, coking and methane formation originating from "demethylation". In this context, this paper does not discuss anything about the reason for high toluene selectivity and stability, which is another impactful outcome in this study. Why is demethylation so inhibited? This may be related to the geometric specificity of the small clusters that do not allow close contact of methyl moiety to the Pt sites or that require energetically unfavored motion at TS. This point should also be clarified by DFT calculation.
2. From the viewpoint of wider interest and hydrogen purity for practical applications, methane concentration in the gas phase downstream (ppm) must be reported, as this fatally damages fuel cells even in a ppm order.
3. Equation 7 in SI ($TOSc = k \exp(-Ea/RT)$) is incorrect. In the Arrhenius equation, the left side must be an intrinsic rate constant that excludes the contribution of reactant pressures. However, the TOSc given in this study is still a kind of reaction rate measured in a specific condition, which can vary depending on reaction conditions ($r = k P(CxHy)^\alpha P(H2)^\beta$) even if it is normalized by the number of active sites. Moreover, describing the pre-exponential factor by k is quite misleading because k is typically used for a rate constant in the left side; A is often used for a pre-exponential factor.
4. There is no explanation how the entropy change was estimated at all. Was this experimentally or theoretically estimated? Judging from the context, it may be an experimental one because it was plotted with the experimental average CNs. If so, it should have been estimated by an Eyring plot, which must be provided in SI as well as the Arrhenius plots. However, if the authors estimated the entropy change using TOSc, this does not provide the true value as suggested in comment 4; this should be reconsidered calculating the intrinsic k. Fortunately in this system, intrinsic k may be estimated by plotting r with various P(CxHy) and ignoring P(H2) because β is close to zero.
5. The reviewer agrees with the overall trend proposed in this study changing from desorption-limiting to C-H activation-limiting with increase in CN. However, Fig. 3a is quite inconsistent with other observations and physicochemical viewpoints; therefore, does not support the overall trend at all. First, if the TOF shows a volcano relationship with CN, Ea should have an

inverse-volcano relationship with CN. Second, although there are some negative entropy changes, they contradict to the facts that entropy always increases in any desorption and C-H dissociation processes (probably due to the incorrect estimation of ΔS^\ddagger as suggested above). Large contribution of translational entropy in the gas phase makes ΔS^\ddagger largely positive, and C-H dissociation is to increase the number of adsorbates. Third, the ΔS^\ddagger trend should also be in an inverse-volcano relationship. Strong adsorption on single-atom requires a significantly large ΔS^\ddagger , which decreases when adsorption becomes weaker with CN increase. Conversely in C-H activation limiting region, weaker adsorption increases entropy when the number of adsorbates increase during the dissociative process. In any case, Fig. 3 must be thoroughly reconsidered.

6. In fig. 4b, the active site environment and pathways considered for Pt13 looks those on Pt single atom or corner site (CN of the active site is too low), which is inconsistent with the authors hypothesis that edge sites are the true active sites for this reaction. The reaction fashion on Pt13 should be reconsidered assuming a more “edge-like” site.

7. As clearly seen in the white-line intensity of XANES, d-vacancy indeed increased with CN. However, this seems to be NOT reflected in the calculated d band structure, Fig. 5a. The fraction of d states above the Fermi level seems to be higher in high CN species than in low CN ones. Considering that single-atom Pt should have a divalent state, it is quite questionable that there is very few d states above the Fermi level. Are the electronic states calculated by the DFT really valid?

8. P25, L431-435: “The higher-lying d-orbitals exhibit stronger hybridization with adsorbate frontier orbitals (e.g., π^*), facilitating electron back-donation from Pt into antibonding states and increasing adsorption strength. The elevated energy of unoccupied antibonding states decreases their occupancy, weakening repulsive interactions and further stabilizing adsorption.” This explanation is wrong. The elevated energy of unoccupied anti-bonding states decreases their occupancy: this is right, but this means that back-donation from Pt to π^* is weakened. Moreover, the decrease in the occupancy of anti-binding states weakens the “bond order of d- π interaction”, thus decreasing adsorption strength. This is the outline of the d band theory describing the covalent interaction between metal and adsorbate, and electrostatic repulsion is not a primary factor.

9. Although the authors chose Qaromatic as a representative descriptor, the factor primarily linking to the adsorptivity should be the energy levels of LUMO and the neighboring orbitals (π^* orbitals). Are Qaromatic and their energy levels really well correlated among the target molecules? This must be clarified to validate the use of Qaromatic as a possible and confident descriptor.

10. For perhydrodibenzyltoluene, is this a mixture of o-, m-, and -p isomers as described in Fig. 5? If so, is it valid to use the Qaromatic value only of the p-isomer for plotting with experimental CNs? Moreover, this reactant also involves the dehydrogenation of both terminal cyclohexyl groups. Although the authors seem to focus only on the center ring, is it really valid to exclude the contribution of the terminal ones from the overall kinetics?

11. There is no explanation for how free energy was calculated. This should be clarified in SI.

12. There is no information about details on chemicals and materials used in this study. They should be provided in SI.

13. Fig. 3b caption: “methylcyclohexane” should be “decaline”.

Reviewer #2

(Remarks to the Author)

The authors described the synthesis of xPt/MgAl₂O₄ catalysts containing a mixture of Pt active species—single atoms, clusters, and nanoparticles—and investigated their effects on the dehydrogenation reactions of cycloalkanes. Notably, they experimentally demonstrated a correlation between particle size, particularly associated with CN_{Pt}-Pt, and catalytic activity for each LOHC. In addition, they conducted computational studies to support their claim that the observed volcano-type trend come from interactions between the d orbitals of the active metal and the π^* orbitals of the reactants, influencing both C-H bond activation and product desorption. However, the manuscript contains several issues that should be addressed. My comments are listed as follows.

1. In Figures 3c–e, the authors state that a lower reaction order corresponds to stronger adsorption of the species. However, to more accurately compare adsorption strengths, it would be necessary to analyze the reaction rates at both low and high reactant concentrations, along with the corresponding rate constants.

2. Electrons from Pt were back-donated into the adsorbate, influencing the adsorption behavior of dehydrogenated products. However, such back-donation typically strengthens the interaction between Pt and the dehydrogenated product, thereby increasing the extent of product adsorption. Moreover, back-donation into the π^* orbital of a dehydrogenated product is likely to weaken the C=C bond, potentially promoting C-C bond cleavage, which may lead to the formation of byproducts and/or coke. The authors need to see this effect and C-H bond activation of a partially dehydrogenated product instead of C-H bond activation of a completely dehydrogenated product using further analysis. Moreover, the authors should address this possibility and revise their catalyst design strategy accordingly.

3. The authors calculated the Bader charges of fully dehydrogenated products (benzene, toluene, and naphthalene) to predict their interactions with the Pt d-band, concluding that d- π^* interactions influence both adsorption and C-H bond

activation. However, this analysis primarily pertains to product desorption. To evaluate the effect on C–H bond activation, the authors should calculate the Bader charges of partially dehydrogenated intermediates to assess how d– π^* interactions influence C–H activation. Additional analysis and corresponding discussion are necessary.

4. The manuscript addresses an important mechanistic interplay between C–H bond activation and product desorption, modulated by Pt particle size and coordination environment. The use of Q_{aromatic} as a descriptor is compelling and provides meaningful insight into the behavior of intermediates and products. However, the discussion on electron transfer into antibonding orbitals and the degree of their occupation would benefit from further clarification. Specifically, it would be helpful to explain—ideally with some quantitative basis—whether the observed occupation of antibonding states facilitates adsorption to an acceptable extent or, conversely, if excessive occupation could compromise adsorption strength. Such elaboration would significantly enrich the mechanistic understanding presented in the study.

5. The R-factor values reported in the EXAFS fitting appear to be relatively high, which raises some concern regarding the reliability of the fitting results. In particular, for the 0.05Pt/MAO sample, it appears that the fitting result of Pt–Pt coordination is not included in the model. Including this contribution would improve the accuracy of the structural interpretation and strengthen the overall analysis.

6. Figure S11, the blue-shaded region is not clearly explained in the manuscript. The authors need to provide a more detailed explanation of the significance and interpretation of this region.

7. Line 261 (p15): The authors claimed that the stability come from oxygen vacancy-mediated SMSI, which will suppress Pt sintering. However, there is no data on Pt size distribution after the 100 h of dehydrogenation.

8. There are a few typographical errors that should be corrected to enhance clarity and precision. For instance, in Figure 3(b), "Methylcyclohexane" should be revised to "Decalin." Please review the manuscript carefully for other minor errors as well.

8. Please check whether the following is needed: SI file, line 218, Figure S5  Figure S6 / line 144: 2059  2061

Reviewer #3

(Remarks to the Author)

In catalysis research, establishing volcano plots that correlate descriptors with activity has been an intriguing topic across reactions like NH_3 synthesis/decomposition, ORR, OER, methanation, and formic acid dehydrogenation. In this context, this study is among the first to introduce the "volcano plot" concept into Liquid Organic Hydrogen Carrier (LOHC) research—a field gaining significant attention for enabling CO_2 -free hydrogen storage and release. While this contribution is timely and offers novelty, it requires substantial revision before reconsideration for publication. My review focuses on experimental aspects preceding computational analysis; I will evaluate the computational section once the issues below are addressed.

1) Catalyst Characterization

1-1) Support Effect

The authors should justify why MgAl_2O_4 was chosen for mechanistic studies. Given that Pt–Pt coordination number (CN) primarily reflects active-metal properties, support effects on particle size/distribution and CN must be systematically ruled out. Although the defect-rich spinel surface with oxygen vacancies reportedly stabilizes ultrasmall Pt nanoparticles and modulates electronic structure via ligand effects, a direct comparison with more inert supports is needed to isolate these influences.

1-2) Inhomogeneity of Pt Sites

Peak broadening and satellite features in characterization data indicate heterogeneous Pt active sites. While average Pt–Pt CN is used as a key descriptor, the standard deviation and distribution profiles of active sites likely impact catalytic performance. Prior studies on LOHC dehydrogenation deconvoluted contributions from distinct Pt facets, which should be incorporated here.

1-3) CN vs. Facet Distribution

Facet distribution is proposed as a descriptor for intrinsic activity (TOF), but identical average CN values may mask differing facet distributions. The methodology for quantifying facet distributions across catalysts is inadequately described (both main text and SI). Additionally, the preparation protocol and interpretation of Figure S6 require clarification.

2) Performance Evaluation

2-1) LOHC Selection Rationale

Testing cyclohexane (single ring), methylcyclohexane (methyl-substituted ring), and decalin (fused bicyclic) introduces confounding variables (ring count, substituents, fusion topology). To robustly generalize volcano plots across LOHCs, a controlled series (e.g., saturated benzene \rightarrow naphthalene \rightarrow anthracene or biphenyl \rightarrow terphenyl) would better isolate structural effects. Alternatively, industrially relevant LOHCs like toluene, monobenzyltoluene, or dibenzyltoluene would enhance practical relevance.

2-2) Pt Speciation and TOF Calculation

The derivation of Pt cluster fractions and TOF via kinetic data/least-squares fitting lacks transparency. Detailed methodological steps, theoretical foundations, and validated literature precedents must be provided. Given active-site heterogeneity, this analysis should account for site-specific distributions.

2-3) Long-Term Stability Testing

While the best-performing catalyst showed stability at 300°C over >100 h, testing additional catalysts under identical conditions is essential to validate the protocol's efficacy. Hydrogen purity analysis (e.g., quantifying methane from demethylation/ring-opening side reactions) must be included to assess selectivity.

Version 1:

Reviewer comments:

Reviewer #1

(Remarks to the Author)

The authors appropriately revised the manuscript based on the reviewers' comments. This paper is now acceptable as is.

Reviewer #2

(Remarks to the Author)

This manuscript investigates Pt catalyzed dehydrogenation of cycloalkanes relevant to LOHCs. By systematically modulating Pt–Pt coordination numbers (CNs) on Pt/MgAl₂O₄ (MAO), the authors identify reactant dependent, volcano shaped activity trends and show how specific ensembles can be selected as optimal active sites. The work advances an electronic structure-based rationale for activity and selectivity and proposes a framework for reactant-specific site design. The demonstration of sustained activity and stability under conditions pertinent to industrial LOHC operation strengthens the practical significance. The authors have also responded to prior comments, improving the completeness of the study. Nonetheless, several elements still warrant deeper justification or additional analysis for the central claims to be fully convincing.

1. In Figure S41, the authors compare C–C cleavage barriers on Pt(111) and Pt13/MAO but omit Pt1. Since Pt1 exhibits the highest d-band center and represents the most under-coordinated limit, it provides an essential boundary condition for the proposed CN-dependent volcano relationship. To strengthen the mechanistic argument, I recommend the following:

- (i) Include Pt1 in the barrier analysis under matched coverage and structural constraints.
- (ii) Present the results for Pt1, Pt13/MAO, and Pt(111) in the form of a volcano-type plot. This would quantitatively substantiate the claim of a “reactant-specific optimum CN” and render the central framework more convincing.

2. The current -ICOHP analysis is restricted to the final adsorption states of fully dehydrogenated products. However, since the rate-determining step (RDS) in cycloalkane dehydrogenation typically corresponds to the first C–H activation, the descriptor should also be examined at earlier points along the pathway. I therefore suggest extending the -ICOHP (or an equivalent bond-order/interaction metric) to include key states before, at, and immediately after the RDS, particularly for partially dehydrogenated intermediates. Establishing how these interaction metrics evolve relative to computed energy barriers would provide a more rigorous mechanistic link between the descriptor and the observed kinetics, thereby strengthening the central framework of the study.

3. The continuous activity decay observed for 0.05 wt% Pt/MAO is attributed to single-atom clustering. Given that clustering increases CN, the reported -ICOHP trends would predict adsorption/desorption shifting toward a more balanced regime. Could clustering instead evolve toward a quasi-steady (or a stable steady) state or even regenerate a more active, clustered catalyst compared to the initial single-atom state? To clarify this point, I recommend the following:

- (i) Conducting time-on-stream experiments combined with intermittent characterization (e.g., HAADF-STEM, EXAFS, CO-DRIFTS) to directly monitor the changes in CN and site population.
- (ii) Testing a pre-clustered control sample to determine whether its performance stabilizes or surpasses that of the single-atom starting material.

In addition, the authors claim that MAO resists sintering, but it remains unclear why such stabilization would not apply to the 0.05 wt% sample. Please elaborate on this discrepancy. Finally, discuss whether sintering should indeed be considered a dominant deactivation pathway under LOHC dehydrogenation conditions, which generally occur at temperatures lower than those of classical sintering regimes.

4. The improved R-factor is noted and appreciated, but the reported Pt–O bond length of ~1.75 Å is unusually short compared to the commonly observed 1.9–2.1 Å range for Pt–O coordination. To assess the robustness of this result, the authors should provide (i) the exact fitting model (scatterers, multiple-scattering paths), k- and R-ranges, S_o² and ΔE_o handling, constraints/tie-lines across shells, and amplitude reduction factors; (ii) confidence intervals, parameter correlations, and residuals; and (iii) tests of alternative physically plausible models (e.g., Pt–O–Al bridges, mixed Pt–O/Pt–Cl or Pt–C environments if relevant).

A concise methodological appendix or an SI table compiling these details would clarify whether the short Pt–O distance originates from model selection, path interference, or reflects a genuine chemical feature. Such clarification is critical for interpreting the local structure with confidence.

5. Several of the central trends depend on site fractions of single atoms, clusters, nanoparticle edges and their intrinsic TOFs through simultaneous fitting, under assumptions such as fixed reaction orders across compositions and size-independent edge TOFs. To ensure robustness of these conclusions, the authors should provide (i) parameter identifiability diagnostics (profile likelihoods or Fisher information), (ii) uncertainty propagation to site-specific TOFs/volcano positions (bootstrap or Bayesian intervals), and (iii) sensitivity to the assumed reaction orders and particle-shape model (see Point 6). Reporting CIs on optimal CN and volcano peak will strengthen the claims.

6. Because site counts directly determine the site-normalized TOFs, the choice of particle-shape model is critical. At present, the main text and SI appear to apply different assumptions (e.g., truncated octahedron vs. cuboctahedron), which may compromise consistency. The author should (i) standardize the particle-shape model (ensure a single geometry is used throughout the manuscript), and explicitly justify the choice (e.g., supported by TEM) as well as (ii) provide a sensitivity analysis by repeating the site-counting procedure with alternative plausible geometries, e.g., truncated octahedron, cuboctahedron, decahedron).

7. The author need to clarify whether the RDS barriers correlate most strongly with the product LUMO, the reactant LUMO, or Δ LUMO along the reaction path. Including a short comparison in the SI would be helpful.

8. For clarity, the author need to specify explicitly whether the volcano positions are referenced to the Pt–Pt CN of the active atom(s) or to the average particle CN. Consistent notation across the manuscript would help avoid ambiguity.

9. Add a schematic showing which structural motif(s) (SA, small cluster edge, NP step) dominate in each loading.

Reviewer #3

(Remarks to the Author)

The authors have thoroughly addressed most of reviewer comments and provided substantial revisions throughout the manuscript. The inclusion of new experiments, expanded analyses, and clarifications substantially strengthen the work, making the study more robust and compelling. Relative to my previous assessment, the manuscript now represents a significant improvement.

Previously, my review was divided into two main categories:

1) Catalyst Characterization, with subsections:

1-1) Support Effect, 1-2) Inhomogeneity of Pt Sites, 1-3) CN vs. Facet Distribution

2) Performance Evaluation, with subsections:

2-1) LOHC Selection Rationale, 2-2) Pt Speciation and TOF Calculation, 2-3) Long-Term Stability Testing

Several points, however, still require further attention before publication:

(1) Support Effects and Volcano Relationship (Comment 1-1):

The comparative study with Pt/Al₂O₃ convincingly demonstrates that MgAl₂O₄ stabilizes low-CN Pt species to a greater extent, supporting claims of enhanced dispersion and stability. However, the assertion of "support-independent" behavior may not be fully validated, as the low-CN regime cannot be accessed on Al₂O₃. Phrasing such as "largely support-independent" or "consistent across supports within the experimentally accessible CN range" would be more precise. Additionally, given the pronounced differences in thermal stability, basicity, and metal-support interaction between MgAl₂O₄ and Al₂O₃, these factors should be acknowledged as potentially influencing both Pt properties and the observed volcano-type reactivity trends. It would be valuable to note that further comparison with additional supports could reinforce the conclusion, though this is not essential for the current study.

(2) CN vs. Facet Distribution (Comment 1-3):

The authors now clearly describe the methodology for quantifying facet distributions using a truncated octahedron model and site-specific TOF normalization (Figure S6). This effectively identifies edge Pt atoms as probable primary active sites, while average CN is an appropriate descriptor in the single-atom and small-cluster regime. The supporting information addresses earlier concerns regarding preparation, calculation details, and figure interpretation. Including a brief acknowledgment that the truncated octahedron model is an idealization—and that real nanoparticles may expose additional sites—would further strengthen the discussion. Since the numerical facet distribution is central to correlating CN with TOF, the authors should also mention complementary experimental methods, such as temperature-programmed reduction or desorption, for future quantitative facet analysis.

(3) Single-Atom Migration (Comment 2-3):

The authors attribute slight deactivation of 0.05Pt/MAO to migration of Pt single atoms onto clusters. While this is plausible, the mechanistic discussion would be improved by either briefly explaining how this migration was inferred from experimental results, or citing studies that have observed similar behavior for Pt on oxide supports. Such clarification would aid readers unfamiliar with single-atom instability and reinforce the proposed mechanism of deactivation at low metal loading. Notably, Pt single atoms abundant in 0.05Pt/MAO may penetrate into the MgAl₂O₄ support via strong metal-support interaction, leading to minimal catalytic activity and limiting the applicability of the proposed CN–TOF relationship.

Additional comments (not previously covered as reviewer #3):

(1) Error Bars on Coordination Number (CN) Data:

Figures 1d, 3a, and S40 report coordination numbers derived from EXAFS fitting, but currently lack error bars. Since uncertainties are already documented in Table S7, these should also be reflected in the graphical data to better represent data variability and reliability.

(2) Limitations of TGA Alone for Coke Resistance and Pt Sintering Claims:

TGA analysis, by itself, cannot fully support claims of coke resistance or suppression of Pt sintering. TGA primarily measures bulk carbon oxidation with limited measurement sensitivity and may not detect smaller coke deposits directly at Pt active sites, which are critical for catalyst deactivation. Likewise, TGA does not provide information on Pt agglomeration or sintering. Given the manuscript's emphasis on stability and performance, further post-reaction analyses—such as STEM-HAADF for Pt dispersion and size, and Raman spectroscopy or XPS for coke characterization—would provide more robust evidence for the proposed stability mechanism.

Version 2:

Reviewer comments:

Reviewer #2

(Remarks to the Author)

The manuscript has been substantially improved and is now nearly ready for publication. The revisions have addressed the key scientific concerns raised previously, and the work is presented in a coherent and technically sound manner. Only a minor issue remains to be corrected.

- In Figure S17a–c, the wavenumber axis of the in situ FTIR and CO-DRIFTS spectra is plotted with lower wavenumber values on the left. It is conventional to display vibrational spectra with higher wavenumber values on the left for easier comparison and readability. Please reverse the axis orientation accordingly.

Reviewer #3

(Remarks to the Author)

The authors have addressed most of the questions I raised. However, relying solely on the truncated octahedron model for facet analysis remains a weakness. Ultimately, precise CO-drift, TPD, and ultra-high-resolution TEM/STEM analyses are needed. Nevertheless, the paper has now reached a level suitable for publication in Nature Communications.

Response to the comments and suggestions of the reviewers

We sincerely thank the reviewers for their thorough evaluation of our manuscript and for providing valuable comments that have greatly helped us improve the quality of our work. We also appreciate the opportunity provided by the editor to address these comments and revise the manuscript accordingly. All the comments have been fully taken into consideration in the revision. All changes in the revised version have been highlighted in yellow for ease of review. Our detailed point-by-point responses are presented below.

Reviewer #1 (Remarks to the Author):

This paper reports the synthesis of a series of Pt catalysts on MAO ranging from single-atom, cluster, to nanoparticles, their catalytic application to cycloalkane dehydrogenation, and fundamental study on the intrinsic effects of the Pt and the molecular sizes on the kinetics and electronic theory. The authors have thoroughly done the investigation by finely distinguishing many kinetic and electronic parameters, establishing a novel model of rationale on cycloalkane dehydrogenation. Besides, the developed catalyst exhibited a remarkably high catalytic performance compared with reported ones. Since this paper has a high impact on catalysis chemistry and is trying to provide significantly important insights for researchers on the relevant field, the reviewer evaluates this paper is potentially suitable for publication in Nature Communications. However, there are many concerns about scientific reliability and unclear points that need modification and reconsideration prior to further

consideration for publication. The following points must be addressed for further peer-review.

Reply:

We sincerely appreciate the reviewer's positive comments and constructive suggestions, and we have done the best effort to improve our study according to the following suggestions.

1. Although high resistance to sintering was attributed to the high catalyst stability in the long-term test of MCH dehydrogenation, it is not likely because sintering is not the primary reason for deactivation at this low temperature region, but coke accumulation should be the more prominent reason. As widely concerned in MCH dehydrogenation, coking and methane formation originating from "demethylation". In this context, this paper does not discuss anything about the reason for high toluene selectivity and stability, which is another impactful outcome in this study. Why is demethylation so inhibited? This may be related to the geometric specificity of the small clusters that do not allow close contact of methyl moiety to the Pt sites or that require energetically unfavored motion at TS. This point should also be clarified by DFT calculation.

Reply:

The reviewer raised an important point regarding the stability mechanism of our Pt/MAO catalyst during methylcyclohexane dehydrogenation. Coke formation might be the cause of the deactivation. To address this, we have conducted further

characterization of spent catalysts. TGA analysis of spent Pt/MA catalyst was carried out, and compared to MAO and Pt/MAO as reference. TG analyses reveal negligible (<0.5%) carbon deposition on the catalyst after 100 hours of operation, confirming its exceptional resistance for coke formation (Figure S15). This resistance can be rationalized by the unique properties of our optimized Pt clusters as reviewer suggested.

Toluene typically exhibits a high propensity for coke formation on Pt catalysts, primarily through dehydrogenation to benzyl intermediates, followed by condensation reactions leading to carbonaceous deposits. Among these steps, aromatic condensation requires a larger ensemble of contiguous Pt sites than the initial dehydrogenation. Therefore, increasing Pt dispersion and decreasing Pt-Pt coordination are expected to significantly suppress coke formation. Consistent with previous findings on propane dehydrogenation, where highly dispersed Pt exhibits excellent coke resistance due to its limited ensemble size (*ACS Catal.*, 2020, 10, 12932-12942), the enhanced resistance to coking observed for atomically dispersed Pt species and small Pt clusters can be rationalized by the restricted number of adjacent active sites available for aromatic condensation.

The following sentences are included in the revised manuscript (Line 287-291 on Page 16) with supporting data in the SI (Figure S15).

“The robust stability of Pt/MAO can be attributed to two key factors: oxygen vacancy-mediated strong metal-support interactions in the defect-engineered MAO, which effectively suppress Pt species sintering during prolonged reactions (Figure S14), and the excellent coke resistance, as evidenced by TG analysis of the spent catalyst

revealing negligible carbon deposition after 100 h (Figure S15). Coke formation in toluene typically proceeds via benzyl-mediated condensation on extended Pt ensembles, whereas high Pt dispersion and low Pt-Pt coordination in Pt/MAO suppress this pathway by limiting adjacent active sites.”

Figure S15. TG profiles of MAO, fresh Pt/MAO, and spent Pt/MAO

We thank the referee for highlighting another important outcome of this work—the high selectivity toward toluene and the correspondingly low selectivity toward benzene, indicating a strong inhibition of demethylation. To elucidate this behavior, we performed DFT calculations to investigate the demethylation pathway of toluene. The results reveal that the transition state for demethylation requires a specific geometric configuration, where the methyl group must align parallel to the Pt surface (*Nat. Commun.*, 2025, 16, 92). On small Pt₁₃ clusters, this configuration is sterically hindered, leading to an increased energy barrier by 0.76 eV compared to the extended Pt(111) surface (Figure S39).

The following text has been added in the revised manuscript (Line 446-450 on Page 25) with supporting data in the SI (Figure S39).

“Nevertheless, this effect proves beneficial in inhibiting demethylation by either blocking methyl group access to Pt sites or requiring unfavorable molecular motions at the transition state (Figure S39), explaining the high selectivity and excellent anti-coking properties of 0.15Pt/MAO during long-term methylcyclohexane dehydrogenation.”

Figure S39. Reaction pathway and free energy profile of C_7H_8 demethylation on Pt₁₃/MAO and Pt(111). Color code: Pt, blue; O, red; Al, pink; Mg, green; C, gray; H, white.

2. From the viewpoint of wider interest and hydrogen purity for practical applications, methane concentration in the gas phase downstream (ppm) must be reported, as this fatally damages fuel cells even in a ppm order.

Reply:

We recognize the critical importance of monitoring methane concentrations to ensure hydrogen purity for fuel cell applications. Methane concentrations were

precisely measured using gas chromatography equipped with an FID detector under the specific conditions employed for the 100-hour methylcyclohexane dehydrogenation stability test. The measured methane levels in the product stream were consistently within 30-40 ppm, well below the 100 ppm threshold specified for PEMFC applications.

We have incorporated these findings into the revised manuscript (Line 278-279 on Page 16) to highlight their significance.

“The catalyst sustains >90% conversion and >99.99% selectivity throughout a 100-hour continuous operation with methane concentrations stably maintained at 30-40 ppm, well below the 100 ppm threshold specified for PEMFC applications³³.”

3. Equation 7 in SI ($TOSc = k \exp(-Ea/RT)$) is incorrect. In the Arrhenius equation, the left side must be an intrinsic rate constant that excludes the contribution of reactant pressures. However, the *TOSc* given in this study is still a kind of reaction rate measured in a specific condition, which can vary depending on reaction conditions ($r = k P(CxHy)^\alpha P(H_2)^\beta$) even if it is normalized by the number of active sites. Moreover, describing the pre-exponential factor by *k* is quite misleading because *k* is typically used for a rate constant in the left side; *A* is often used for a pre-exponential factor.

Reply:

We sincerely appreciate the reviewer’s insightful comments regarding the treatment of kinetic data in our manuscript. The reviewer is correct that the Arrhenius equation should involve an intrinsic rate constant independent of reactant pressures,

whereas our original formulation used TOS_c , a normalized reaction rate still influenced by partial pressures. In response, we have carefully revised the equations to: $TOS_c = A \exp(-\frac{E_a}{RT}) P_{reactant}^a$, where A now denotes the pre-exponential factor, E_a is the activation energy, $P_{reactant}$ represents the partial pressures of the reactant (cyclohexane, methylcyclohexane, or decalin), a is the reaction order with respect to the reactant. During kinetic measurements, conversions were maintained below 20% to ensure differential conditions, and the reaction orders were determined from the 0.05Pt/MAO results shown in Figure 3c-e. All kinetic experiments were conducted at low conversions to minimize the influence of product co-adsorption, allowing the reaction rate to be accurately expressed using a power-law dependence solely on the reactant concentration.

While this reformulation required recalibration of certain parameters, it does not affect the main conclusions of our manuscript. Because identical $P_{reactant}$ values were maintained during comparative measurements, the $P_{reactant}^a$ term acts as a constant scaling factor. Consequently, this correction only modifies the absolute values of the pre-exponential factor A but leaves the calculated activation energies (E_a) and the mass fractions of Pt single atoms, clusters, and nanoparticles (f_s , f_c , f_n) unchanged (Table S6).

The revised version now properly reflects these corrections in both the main text (Figure 2c) and SI (Line 315-321 on Page 15 and Table S6).

Figure 2. (c) Fractions of Pt species in various Pt/MAO catalysts determined from kinetic data.

$$TOS_c = A \exp\left(-\frac{E_a}{RT}\right) P_{reactant}^a \quad (14)$$

Where A denotes the pre-exponential factor, $P_{reactant}$ represents the partial pressures of the reactant (cyclohexane, methylcyclohexane, or decalin), a is the reaction order with respect to the reactant. During kinetic measurements, the conversion was maintained $<20\%$ specifically to avoid complications from product inhibition effects. The reaction orders of reactant can be obtained from the reaction order results of 0.05Pt/MAO in Figure 3c-e.”

Table S6. Least-squares fitting details for determining the fractions of Pt species in 0.05Pt/MAO and 0.15Pt/MAO catalysts based on kinetic data.

Catalyst	f_s	f_c	f_n	x	CYH		MCH		DEC		R^2
					E_a	$\ln A$	E_a	$\ln A$	E_a	$\ln A$	
0.05Pt/MAO	0.63	0.37	1.7×10^{-15}	0.85	138.57	38.47	103.47	31.55	102.11	29.41	0.9963
0.15Pt/MAO	1.68×10^{-15}	0.56	0.44	0.73	74.17	24.54	129.63	36.08	200.00	49.42	0.9973

4. There is no explanation how the entropy change was estimated at all. Was this experimentally or theoretically estimated? Judging from the context, it may be an experimental one because it was plotted with the experimental average CNs. If so, it should have been estimated by an Eyring plot, which must be provided in SI as well as the Arrhenius plots. However, if the authors estimated the entropy change using TOSc, this does not provide the true value as suggested in comment 4; this should be reconsidered calculating the intrinsic k . Fortunately in this system, intrinsic k may be estimated by plotting r with various $P(\text{CxHy})$ and ignoring $P(\text{H}_2)$ because β is close to zero.

Reply:

We thank the reviewer for the valuable comments and acknowledge that our original manuscript did not clearly describe the method for determining entropy. We are pleased to clarify our approach. The entropy change (ΔS^\ddagger) was derived experimentally based on kinetic measurements. As the reviewer correctly pointed out, ΔS^\ddagger should ideally be obtained from an Eyring plot analysis. Accordingly, we have revised the manuscript and included the Eyring plots (Figure S17) in the Supporting Information, alongside the Arrhenius plots (Figure S16) for full transparency.

The revised version of the manuscript now provides all necessary details, including the Eyring (Figure S17) and Arrhenius plots (Figure S16). The following description about the entropy change calculation has been added in the SI (Line 370-384 on Page17-18).

Figure S16. Arrhenius plots for calculation of activation energy over various Pt/MAO catalysts for (a) cyclohexane dehydrogenation, (b) methylcyclohexane dehydrogenation, and (c) decalin dehydrogenation.

Figure S17. Eyring plots for calculation of entropy change over various Pt/MAO catalysts for (a) cyclohexane dehydrogenation, (b) methylcyclohexane dehydrogenation, and (c) decalin dehydrogenation.

“Calculation of the activation energy and entropy change

The intrinsic rate constant (k) was calculated by TOF and reaction orders:

$$k = \frac{TOF}{P_{reactant}^a} \quad (16)$$

Based on the consistent reaction orders observed in Figure 3c-e between low (0.05Pt/MAO) and high-coordinated Pt (3Pt/MAO), we approximated the orders for intermediate catalysts as follows: for 0.02-0.15Pt/MAO, we used values similar to 0.05Pt/MAO, while for 0.375-3Pt/MAO, we adopted values comparable to 3Pt/MAO.

The apparent activation energy (E_a) was calculated through the Arrhenius equation:

$$\ln k = \ln A - \frac{E_a}{RT} \quad (17)$$

Based on transition state theory, the rate constant is expressed as:

$$k = \frac{k_B T}{h} \exp\left[-\frac{\Delta H^{0*}}{RT}\right] \exp\left[\frac{\Delta S^\ddagger}{R}\right] \quad (18)$$

Where k_B , h and ΔH^{0*} were the Boltzmann constant, Planck constant and enthalpy of activation, respectively. Then the enthalpy change (ΔS^\ddagger) can be obtained by the Eyring plots ($\ln \frac{k}{T}$ versus $\frac{1}{T}$) according to the equation:

$$\ln \frac{k}{T} = -\frac{\Delta H^{0*}}{RT} + \ln \frac{k_B}{h} + \frac{\Delta S^{0*}}{R} \quad (19)$$

The results were updated in Figure 3a in the main text as well as in Figure S19.

Figure 3. (a) Apparent activation energy (square) and entropy (circle) as a function of CN_{Pt-Pt} for the dehydrogenation of cyclohexane, methylcyclohexane, and decalin.

Figure S19. Apparent activation energy (square) and entropy (circle) of Pt/MAO catalysts for the dehydrogenation of cyclohexane, methylcyclohexane and decalin.

5. The reviewer agrees with the overall trend proposed in this study changing from desorption-limiting to C-H activation-limiting with increase in CN. However, Fig. 3a is quite inconsistent with other observations and physicochemical viewpoints; therefore, does not support the overall trend at all. First, if the TOF shows a volcano relationship with CN, E_a should have an inverse-volcano relationship with CN. Second, although there are some negative entropy changes, they contradicts to the facts that entropy always increases in any desorption and C-H dissociation processes (probably due to the incorrect estimation of $\Delta S_{\ddagger}^{\ddagger}$ as suggested above). Large contribution of translational entropy in the gas phase makes $\Delta S_{\ddagger}^{\ddagger}$ largely positive, and C-H dissociation is to increase the number of adsorbates. Third, the $\Delta S_{\ddagger}^{\ddagger}$ trend should also be in an inverse-volcano relationship. Strong adsorption on single-atom requires a significantly large $\Delta S_{\ddagger}^{\ddagger}$, which decreases when adsorption becomes weaker with CN increase. Conversely in C-H activation limiting region, weaker adsorption increases entropy when the number of adsorbates increase during the dissociative process. In any case, Fig. 3 must be

thoroughly reconsidered.

Reply:

We thank the reviewer for the constructive comments. Following the reviewer's suggestions, we have carefully re-evaluated the kinetic analysis and revised the relevant discussion and **Figure 3** accordingly. We fully agree that the kinetic parameters should be presented consistently with the proposed mechanistic framework. In the revised manuscript, **Figure 3a** has been updated using new data based on rate constants rather than TOFs to ensure accuracy.

The kinetic study was conducted to gain insights into the potential volcano-type dependence of catalytic activity on the CN_{Pt-Pt} . The updated results, summarized in **Figure 3**, reveal a significant correlation between catalytic performance (activity, activation energy, entropy changes, and reaction orders) and CN_{Pt-Pt} . However, the anticipated inverse volcano trend for activation energy was not observed; instead, activity, activation energy, and entropy changes all exhibit a similar volcano-shaped dependence on CN_{Pt-Pt} . This correlation suggests that the observed activity trend is likely entropy-driven, and the detailed analysis is presented in the revised discussion.

Dependence of adsorption strength on CN_{Pt-Pt} :

Adsorption strength is a key parameter governing catalytic performance. In this study, the adsorption strengths of reactants, intermediates, and products were evaluated using apparent reaction orders. According to the Langmuir-Hinshelwood-Hougen-Watson (LHHW) kinetic framework and the measured reaction orders, the LHHW model can be proposed

$$-r = \frac{kK_R P_R}{(1 + K_R P_R + K_P P_P)^2}$$

where R and P represent the reactant and product, respectively. Based on the LHHW kinetic model, the apparent reaction orders of the reactant (R) and products (P) are expressed as:

$$n_R = 1 - 2\theta_R$$

$$n_P = -2\theta_P$$

The reaction orders of R (n_R) and P (n_P) are a function of the site coverages. Since the reaction orders are directly related to site coverages under reaction conditions, they provide a quantitative measure of adsorption strength: a more negative reaction order indicates stronger adsorption.

For cyclohexane dehydrogenation, the reaction orders follow the sequence cyclohexane > H₂ > benzene, indicating relatively stronger adsorption of benzene compared to hydrogen and cyclohexane. On 0.05Pt/MAO, the reaction order of benzene is highly negative (≈ -1.55), suggesting strong benzene adsorption and significant site blocking, which inhibits the overall reaction. In contrast, on 3Pt/MAO with a larger CN_{Pt-Pt}, the reaction orders of both cyclohexane and benzene, especially benzene, become much less negative, indicating substantially weaker adsorption on catalysts with larger Pt ensembles.

For methylcyclohexane dehydrogenation, the reaction orders exhibit a similar dependence on CN_{Pt-Pt} as observed for cyclohexane. However, for decalin dehydrogenation, the trend differs markedly: the reaction order of decalin remains consistently around +2, independent of CN_{Pt-Pt}, while the hydrogen reaction order

becomes positive. This shift suggests a distinct reaction mechanism compared to cyclohexane and methylcyclohexane dehydrogenation, which warrants further investigation in future studies.

Volcano dependence of activation energy on CN_{Pt-Pt} :

According to LHHW kinetic framework, the apparent activation energy can be expressed as:

$$E_{app} = E_{rds} + (1 - 2\theta_R)\Delta H_R - 2\theta_P\Delta H_P$$

where E_{rds} denotes the activation energy of the rate-determining step (RDS), ΔH_R and ΔH_P are the adsorption enthalpies of the reactant and product/intermediates, respectively (negative values for exothermic adsorption), and θ_R and θ_P are their coverages. Thus, E_{app} is determined not only by the intrinsic energy barrier of the RDS, but also by coverage-dependent contributions from the adsorption enthalpies of reactants and products.

A schematic diagram of the activation energy changes with CN_{Pt-Pt} was presented in Figure S18 to illustrate the observed trend.

Figure S18. Correlation between Pt coordination number and E_{app} governed by adsorption-desorption thermodynamics in Pt/MAO catalysts.

- **Low-CN_{Pt-Pt} region:** The intrinsic C-H activation barrier (E_{rds}) is low, but the product is strongly bound (ΔH_p large, θ_p high), making product desorption the dominant contribution to E_{app} . The strong product inhibition leads to low dehydrogenation activity.

- **Intermediate-CN_{Pt-Pt} region:** As CN increases, the intrinsic C-H activation rises moderately, while product adsorption becomes weaker, lowering the desorption contribution. At this stage, E_{app} reflects the sum of C-H activation and product desorption barriers, which resulting in the highest TOF.

- **High-CN_{Pt-Pt} region:** Product adsorption is weak (θ_p small, ΔH_p negligible in E_{app}), so E_{app} is dominated by E_{rds} and can decrease slightly.

Volcano dependence of entropy changes on CN_{Pt-Pt}:

According to the study (*Nat. Commun.*, 2019, 10, 1428), ΔS_{app}^\ddagger and E_{app} often exhibit a linear correlation (enthalpy-entropy compensation effect), which arises from concomitant changes in surface-adsorbate bonding strength. This framework rationalizes that ΔS_{app}^\ddagger follows a similar CN_{Pt-Pt} trend to E_{app} in our results.

In most cases, ΔS_{app}^\ddagger is negative, as shown in **Figure 3a**, primarily due to strong reactant adsorption restricting translational and rotational degrees of freedom. Such negative entropy changes are common in dehydrogenation reactions (*J. Catal.*, 2011, 277, 104-116; *J. Phys. Chem. C*, 2014, 118, 27292-27300). Here, ΔS_{app}^\ddagger represents the entropy difference between the transition state and the gas-phase reactant; thus, it becomes negative when the transition state is more constrained than the initial state.

Alexis T. Bell and co-workers reported that ΔS_{app}^\ddagger for n-butane dehydrogenation can

be negative when adsorption reduces translational/rotational degrees of freedom more in the transition state than in the initial state. They observed that the sign and magnitude of $\Delta S_{\text{app}}^{\ddagger}$ depend strongly on site location and transition-state geometry, with some catalysts exhibiting large positive $\Delta S_{\text{app}}^{\ddagger}$ and others small or even negative values (*J. Am. Chem. Soc.*, 2013, 135, 19193-19207; *J. Am. Chem. Soc.*, 2016, 138, 4739-4756).

We fully agree with the reviewer's comment that strong adsorption on single-atom Pt sites leads to a significantly more negative $\Delta S_{\text{app}}^{\ddagger}$. As $\text{CN}_{\text{Pt-Pt}}$ increases, adsorption strength weakens, resulting in smaller entropy changes (Figure 3a). Moreover, C-H bond activation typically involves a transition state coordinated with multiple adjacent surface atoms, leading to a less constrained transition-state configuration and thus a higher entropy. This indicates that the entropy change does not vary monotonically with Pt coordination when transitioning from atomically dispersed Pt to larger nanoparticles.

Entropy-driven process:

Comparing Figure 2d and Figure 3a, the CN-dependence of activation energy and catalytic activity follows a similar trend, deviating from the expected "inverse-volcano" relationship. According to transition state theory, the reaction rate is determined by both the activation energy and the entropy change. In our case, the entropy contribution plays a dominant role in controlling the reaction rate, indicating that the observed kinetics are primarily entropy-driven.

We have carefully reconsidered the kinetic analysis in light of the reviewer's suggestions and revised the relevant discussion (Line 318-349 on Page 18-19) and Figure 3a accordingly.

“Both E_{app} and ΔS_{app}^\ddagger display a CN-dependent volcano-type trend, suggesting a mechanistic transition as CN_{Pt-Pt} changes. Notably, the CN-dependence of E_{app} is not a simple inverse reflection of the TOF volcano, as it reflects both the intrinsic barrier of the rate-determining step (RDS) and coverage-dependent adsorption enthalpies of reactants and products (Figure S18).

According to Langmuir-Hinshelwood-Hougen-Watson kinetics, E_{app} can be described as:

$$E_{app} = E_{rds} + (1 - 2\theta_R)\Delta H_R - 2\theta_P\Delta H_P$$

where E_{rds} denotes the intrinsic activation energy of RDS, θ_R and θ_P represent the coverage of reactant (R) and intermediates/products (P), and ΔH_R and ΔH_P correspond to their respective adsorption enthalpies. At low CN_{Pt-Pt} , E_{rds} is intrinsically small, but strong product adsorption (ΔH_P large, θ_P high) makes desorption the dominant energetic penalty, limiting active site turnover despite the low intrinsic barrier. As CN_{Pt-Pt} increases, E_{rds} increases moderately, while product adsorption weakens (ΔH_P decreases, θ_P lower). Here, E_{app} reflects the sum of C-H activation and product desorption barriers, which can be higher than in the low- CN_{Pt-Pt} case, but the optimal balance between the two processes yields the maximum TOF. At high- CN_{Pt-Pt} region, product adsorption is weak (ΔH_P negligible, θ_P small), so E_{app} is dominated by E_{rds} and may decrease slightly. However, the reactant coverage and adsorption strength are too low to sustain high activity, leading to a drop in TOF.

The CN_{Pt-Pt} dependence of ΔS_{app}^\ddagger follows a similar trend to E_{app} , consistent with the enthalpy-entropy compensation effect⁴⁶. Negative ΔS_{app}^\ddagger values at low CN_{Pt-Pt} arise

from that the transition state have fewer accessible degrees of freedom than the initial state on strongly bound surfaces. As CN_{Pt-Pt} increases, adsorption weakens and ΔS_{app}^\ddagger becomes less negative. In the high- CN_{Pt-Pt} regime, however, ΔS_{app}^\ddagger becomes more negative again, which may be attributed to multi-atom cooperative C-H activation at extended Pt ensembles, imposing greater configurational constraints on the transition state. Overall, these findings indicate that the volcano-shaped activity trend arises from the combined enthalpic and entropic contributions governed by coverage-dependent adsorption. The kinetics are thus best described as entropy-driven, with a mechanistic shift from desorption-limited at low CN_{Pt-Pt} to C-H activation-controlled at higher CN_{Pt-Pt} .

6. In fig. 4b, the active site environment and pathways considered for Pt13 looks those on Pt single atom or corner site (CN of the active site is too low), which is inconsistent with the authors hypothesis that edge sites are the true active sites for this reaction. The reaction fashion on Pt13 should be reconsidered assuming a more “edge-like” site.

Reply:

We thank the reviewer for the insightful comment. Our statement that “edge sites are the true active sites” refers specifically to Pt in nanoparticle form, where edge atoms indeed have the lower coordination among catalytically relevant sites. For sub-nanometer Pt clusters, however, the “active site structure” is represented by the average coordination number (CN) of the Pt atoms rather than by a specific geometric motif

such as an edge atom in a large particle. In this work, we used a Pt₁₃ cluster to model Pt sites with an average CN of ~4.7. The initial Pt₁₃ structure had an average CN of 4.5 (Figure R1). Upon structural optimization, strong interactions with the MAO support led to partial flattening of the cluster into a quasi-monolayer arrangement, resulting in a reduced average CN of 3.9. Although this value is lower than the initial CN, it remains distinct from the other Pt models considered in this study, thus allowing us to capture the effect of coordination variation on Pt electronic properties, activation barriers, adsorption strengths, and the corresponding reaction pathways.

Figure R1. Initial and optimized geometries of Pt₁₃/MAO. Color code: Pt, blue; O, red; Al, pink; Mg, green.

We have clarified this point in the revised manuscript and explicitly discussed the structural evolution of Pt₁₃ on MAO (Line 403-406 on Page 23), highlighting its representativeness for low-CN Pt sites in sub-nanometer clusters.

“Upon optimization, strong Pt-MAO interactions induced partial flattening of Pt₁₃, reducing its CN_{Pt-Pt} to 3.9. Despite this decrease, the CN_{Pt-Pt} remains distinct from the other Pt models, enabling meaningful comparisons of catalytic properties across different CN_{Pt-Pt} regimes.”

7. As clearly seen in the white-line intensity of XANES, d-vacancy indeed increased with CN. However, this seems to be NOT reflected in the calculated d band structure, Fig. 5a. The fraction of d states above the Fermi level seems to be higher in high CN species than in low CN ones. Considering that single-atom Pt should have a divalent state, it is quite questionable that there is very few d states above the Fermi level. Are the electronic states calculated by the DFT really valid?

Reply:

We thank the reviewer for raising this important point regarding the correlation between XANES white-line intensity and our DFT-calculated d-band structure. Upon careful re-examination of our data, we realize that the seeming contradiction stems from focusing on absolute versus relative DOS. While the total unoccupied d-states above the Fermi level do increase with higher CN due to the growing number of Pt atoms in larger clusters, the fraction of unoccupied d-states relative to the total d-band decreases consistently, in alignment with XANES results. Quantitatively, the fractional unoccupied d-states are 0.171 for Pt₁/MAO, 0.129 for Pt₄/MAO, 0.121 for Pt₁₃/MAO, and 0.117 for Pt(111), clearly showing the trend of decreasing d-vacancy with increasing CN. Regarding the single-atom Pt case, the low apparent unoccupied states above the Fermi level in **Figure 5a** were an artifact of the plotting style (color scale and b-spline smoothing), which visually suppressed the unoccupied region. After revising the plot, it is evident that Pt₁ possesses a substantial number of unoccupied d-states above Fermi level.

We have updated the manuscript to include a more detailed discussion reconciling the XANES and DFT results (Line 472-479 on Page 27) and revised the Figure 5a to more clearly represent the unoccupied d-states.

“Accompanying this trend is a corresponding increase in the fraction of unoccupied 5d states above the Fermi level with decreasing CN_{Pt-Pt} . These observations are consistent with the Pt L_{III} -edge XANES spectra results (Figure 1c), where the absorption edge moves to higher energy in the order $0.02Pt/MAO > 0.05Pt/MAO > 0.15Pt/MAO$. Quantitative analysis of white-line intensities, under the assumption of a linear relationship with 5d-orbital vacancies, confirms this trend, with XANES-derived vacancy counts monotonically increasing as CN_{Pt-Pt} decreases (Figure S40b).”

Figure 5. (a) Projected density of states of Pt 5d-orbitals with band center positions.

8. P25, L431-435: “The higher-lying d-orbitals exhibit stronger hybridization with adsorbate frontier orbitals (e.g., π^*), facilitating electron back-donation from Pt into antibonding states and increasing adsorption strength. The elevated energy of unoccupied antibonding states decreases their occupancy, weakening repulsive interactions and further stabilizing adsorption.” This explanation is wrong. The

elevated energy of unoccupied anti-bonding states decreases their occupancy: this is right, but this means that back-donation from Pt to π^ is weakened. Moreover, the decrease in the occupancy of anti-binding states weakens the “bond order of d- π interaction”, thus decreasing adsorption strength. This is the outline of the d band theory describing the covalent interaction between metal and adsorbate, and electrostatic repulsion is not a primary factor.*

Reply:

We thank the reviewer for the insightful comments on our use of d-band theory and agree that our original explanation overstated the role of antibonding “repulsion”. In the revised text, we now clarify that an upward shift of the Pt d-band center enhances the d- π^* orbital interaction with aromatics, increasing the occupancy of bonding states while reducing the occupancy of antibonding states. This net increase in bond order strengthens the covalent metal-adsorbate interaction and thus increases adsorption energy, in accordance with the d-band center theory. We have also removed the reference to “weakening repulsive interactions,” as electrostatic effects are not the primary factor in this context, whereas covalent bond order governs adsorption strength. Furthermore, we have added COHP analyses (Figure S45) showing that -ICOHP values increase as the Pt d-band center shifts upward (with decreasing Pt coordination number), indicating progressively stronger Pt-adsorbate bonding due to enhanced d- π^* back-donation.

We have updated the manuscript to incorporate this revised explanation (Line 482-485 on Page 27) and included the ICOHP results (Figure S45) together with the

corresponding discussion (Line 539-545 on Page 30).

“From an orbital interaction perspective, the upshifted d-band center narrows the energy gap between Pt d-orbitals and adsorbate antibonding orbitals (e.g., π^* for cycloalkenes). This enhances $d-\pi^*$ back-donation, increasing bonding-state occupancy and decreasing antibonding-state occupancy, thereby strengthening Pt-adsorbate interactions⁵³. Such enhanced $d-\pi^*$ back-donation explains the stronger adsorption observed on low-coordinated Pt sites.”

Figure S45. -ICOHP values of Pt-C bond for dehydrogenated products on various Pt catalyst models.

“To further quantify this interaction, crystal orbital Hamilton populations (COHP) analysis of Pt-C bonds in dehydrogenated products was performed (Figure S45). The results show that -ICOHP increases with decreasing Pt coordination, consistent with stronger covalent interactions from enhanced $d-\pi^*$ back-donation. Notably, optimal catalytic performance across different reactions occurs when -ICOHP falls within 1.2-1.4, reflecting a balance between sufficient stabilization for C-H activation and facile product desorption.”

9. Although the authors chose Q_{aromatic} as a representative descriptor, the factor primarily linking to the adsorptivity should be the energy levels of LUMO and the neighboring orbitals (π^* orbitals). Are Q_{aromatic} and their energy levels really well correlated among the target molecules? This must be clarified to validate the use of Q_{aromatic} as a possible and confident descriptor.

Reply:

We thank the reviewer for this valuable suggestion. Following the reviewer's advice, we have calculated the LUMO energies of the target aromatic molecules (Figure 5c) and examined their correlation with Q_{Aromatic} (Figure S43). The results show a clear linear relationship: as Q_{Aromatic} increases, the LUMO energy decreases. This correlation confirms that Q_{Aromatic} effectively reflects the electronic characteristics of the π^* orbitals that govern adsorption. Following the reviewer's advice, LUMO energy has been adopted as the unified descriptor replacing Q_{Aromatic} in the main text and a direct correlation between LUMO energy and Pt d-band center has been established, as shown in the revised Figure 5e. This modification renders the proposed scaling relationship more explicit and mechanistically rational.

We have added the calculated LUMO energies of the aromatic products (Figure 5c) and their correlation with Q_{Aromatic} (Figure S43). More importantly, Q_{Aromatic} has been replaced by LUMO energy as the electronic descriptor throughout the manuscript, including the Abstract, main discussion (Line 520-536 on Page 29-30), Figure5d-e and Conclusions.

“Abstract: We introduce the LUMO energy of aromatic products as a universal

electronic descriptor that serves as a proxy for π^* orbital energy. LUMO energy shows strong linear correlations with the d-band center of Pt at the optimal CN_{Pt-Pt} , enabling rational tuning of electronic interactions across diverse reactants.”

“Higher carbon electron density stabilizes the π^* antibonding orbitals by lowering their energy, as increased electron density screens the nuclear potential and reduces the effective energy of π -electrons⁵⁴. This trend is reflected in the computed LUMO energies, which decrease progressively from benzene to toluene to naphthalene (Figure 5c). Bader charge analysis further supports this relationship, showing that molecules with higher carbon electron density exhibit lower LUMO energies (Figure S43). Notably, partially dehydrogenated intermediates (C_6H_8 , C_7H_{10} , and $C_{10}H_{14}$; C_6H_{10} , C_7H_{12} , and $C_{10}H_{16}$) follow the same trend (Figure S44), confirming the general applicability of LUMO energy as an electronic descriptor.

For adsorption on Pt surfaces, lower LUMO energies (π^* levels) reduce the energy gap between the π^* orbitals and the Pt d-band, facilitating stronger d- π^* back-donation and stabilizing Pt-adsorbate interactions⁵⁵. In such cases, tuning the Pt d-band center downward mitigates overly strong adsorption by restoring better energetic alignment with low-LUMO aromatics (Figure 5d). Consequently, LUMO energy effectively captures the electronic requirements for optimal d- π^* orbital alignment, enabling targeted tuning of the Pt d-band center to optimize orbital energy alignment with the aromatic adsorbate.”

Figure 5. Generalizing the volcano curves of Pt active sites for cycloalkane dehydrogenation. **a** Projected density of states of Pt 5d-orbitals with band center positions. **b** Schematic representation illustrating the balance between C-H activation energy and product desorption energy for different cycloalkane dehydrogenation reactions. **c** Projected density of states and LUMO energy of aromatics. **d** Mechanistic illustration of CN_{Pt-Pt} optimization guided by Pt-adsorbate orbital hybridization. **e** Correlation of Pt d-band center and optimal CN_{Pt-Pt} with LUMO energy of aromatic product in cycloalkane dehydrogenation.

“Conclusions: A key insight from this work is the discovery of LUMO energy of dehydrogenated aromatics, as a unifying molecular descriptor that directly correlates with the optimal CN_{Pt-Pt} . As LUMO energy decreases with increasing aromatic size and

conjugation, metal-adsorbate interactions are strengthened, necessitating higher-coordinated Pt sites with lower d-band center to sustain efficient turnover. This principle not only explains the observed trends across multiple cycloalkanes, but also finds experimental validation in the dehydrogenation of perhydro-benzyltoluene and perhydro-dibenzyltoluene.”

Figure S43. Correlation between carbon electron density and LUMO energy of aromatics.

10. For perhydrodibenzyltoluene, is this a mixture of *o*-, *m*-, and *p* isomers as described in Fig. 5? If so, is it valid to use the $Q_{aromatic}$ value only of the *p*-isomer for plotting with experimental CNs? Moreover, this reactant also involves the dehydrogenation of both terminal cyclohexyl groups. Although the authors seem to focus only on the center ring, is it really valid to exclude the contribution of the terminal ones from the overall kinetics?

Reply:

We thank the reviewer for this valuable comment regarding the isomer distribution and dehydrogenation sites of perhydrodibenzyltoluene. In our study, the commercial

perhydrodibenzyltoluene feed contains multiple structural isomers. We have now calculated the LUMO energy values for the three major isomers (23-, 24-, and 25--dibenzyltoluene), obtaining 2.84, 2.84, 2.88, respectively (Figure S47), demonstrating that their LUMO energy levels are essentially identical. Accordingly, we used the average LUMO energy value for correlation analysis in the revised manuscript.

To further address the reviewer's concern about the adsorption behavior, we computed the adsorption energies of both the central aromatic ring and terminal cyclohexyl rings for these isomers on Pt(111). The adsorption energies fall in the range of -0.88 to -1.01 eV (Figure S49), reflecting some steric effects among isomers, but all values lie between that of toluene (-0.82 eV) and naphthalene (-1.03 eV) adsorption on Pt(111). This confirms that the observed adsorption strengths are consistent with the general trend established in our study. To ensure accuracy, we have used the averaged adsorption energy across these isomers for descriptor correlation and have clarified this treatment in the revised text.

We have incorporated these additional calculations and clarifications into the manuscript (Line 550-556 on Page 30), and added new figures in the SI (Figure S47, Figure S49) showing LUMO energy and adsorption energies for the three isomers.

“LUMO energy calculations yield average values of 4.24 eV for benzyltoluene isomers (2-, 3-, 4-) (Figure S46) and 2.85 eV for dibenzyltoluene isomers (2,3-, 2,4-, and 2,5-) (Figure S47), higher than in toluene but lower than in naphthalene. DFT calculations further showed that the adsorption energies of benzyltoluene isomers (-0.77 to -0.89 eV) and dibenzyltoluene isomers (-0.82 to -1.03 eV) on Pt(111) fall within

a narrow range (Figure S48-49) and lie between those of toluene and naphthalene (Figure S50).”

Figure S47. Projected density of states and LUMO energy of 2,3-, 2,4-, 2,5-dibenzyltoluene isomers.

Figure S49. Adsorption configurations and energies of dibenzyltoluene isomers on Pt(111).

11. There is no explanation for how free energy was calculated. This should be clarified in SI.

Reply:

We sincerely appreciate the reviewer's valid request for clarification regarding our free energy calculations. We have now included the full description of free energy calculations in the DFT calculations section of the SI (Line 414-415 on Page 19), with the equation:

$$\Delta G = \Delta E_{ads} + \Delta E_{ZPE} - T\Delta S_{ads} \quad (22)$$

where ΔE_{ZPE} is the zero point energy correction, and $T\Delta S_{ads}$ is the corresponding entropy correction (T is set to be 573 K).

12. *There is no information about details on chemicals and materials used in this study. They should be provided in SI.*

Reply:

We sincerely apologize for this oversight for complete chemical information. We have now included a comprehensive “**Materials**” section in the SI, containing all experimental details for full reproducibility.

“**Materials**”

H₂PtCl₆·6H₂O (AR grade, Pt≥37.5%), C₂H₆O (ethanol, AR grade), Mg(CH₃COO)₂·4H₂O (magnesium acetate tetrahydrate, AR grade), γ-Al₂O₃ powder, and C₁₀H₁₈ (decalin, AR grade) were obtained from Sinopharm Chemical Reagent Co., Ltd., China. C₉H₂₁O₃Al (aluminum isopropoxide, AR grade) and C₆H₁₂ (cyclohexane, AR grade) were purchased from Shanghai Macklin Biochemical Co., Ltd. C₇H₁₄ (methylcyclohexane, ≥99%) was supplied by Shanghai Aladdin Biochemical

Technology Co., Ltd. High-purity gases including N₂ (99.999%), Ar (99.999%), air, and H₂ (99.999%) were provided by Qingdao Xinke Gas Co., Ltd. Benzyltoluene (C₁₄H₁₄, 99%, mixture of isomers) and dibenzyltoluene (C₂₁H₂₀, 99%, mixture of isomers) was acquired from Adamas Reagent Co., Ltd. and subsequently hydrogenated in our laboratory to obtain perhydro-benzyltoluene (C₁₄H₂₆) and perhydro-dibenzyltoluene (C₂₁H₃₈). All aqueous solutions were prepared using deionized water (18.2 MΩ·cm) produced in our laboratory.”

13. Fig. 3b caption: “methylcyclohexane” should be “decaline”.

Reply:

We have carefully revised the caption of **Figure 3b** as suggested. Thank you again for your thorough review.

“Figure 3. (b) TPSR profiles for **decalin** dehydrogenation on 0.05Pt/MAO and 3Pt/MAO catalysts.”

Reviewer #2 (Remarks to the Author):

The authors described the synthesis of $x\text{Pt}/\text{MgAl}_2\text{O}_4$ catalysts containing a mixture of Pt active species—single atoms, clusters, and nanoparticles—and investigated their effects on the dehydrogenation reactions of cycloalkanes. Notably, they experimentally demonstrated a correlation between particle size, particularly associated with CNPt–Pt, and catalytic activity for each LOHC. In addition, they conducted computational studies to support their claim that the observed volcano-type trend come from interactions between the d orbitals of the active metal and the π^ orbitals of the reactants, influencing both C–H bond activation and product desorption. However, the manuscript contains several issues that should be addressed. My comments are listed as follows.*

Reply:

Thanks for the reviewer's positive comments on the manuscript and we have done the best effort to improve our study according to the following suggestions.

1. In Figures 3c–e, the authors state that a lower reaction order corresponds to stronger adsorption of the species. However, to more accurately compare adsorption strengths, it would be necessary to analyze the reaction rates at both low and high reactant concentrations, along with the corresponding rate constants.

Reply:

We appreciate the reviewer's insightful suggestion. Following the recommendation, we have conducted additional measurements of the dehydrogenation

rates at both low and high product concentrations. The results (Figure S23) show that, at high product concentration, the dehydrogenation rate decreases significantly due to the inhibitory effect of the product. This inhibition is consistently more pronounced for the 0.05Pt/MAO catalyst than for the 3 Pt/MAO catalyst, resulting in lower dehydrogenation rates for 0.05Pt/MAO under high product concentration conditions. These findings are consistent with the conclusions drawn from the reaction order analysis at high reactant concentration, confirming that the product adsorbs more strongly on low-coordination Pt sites, thereby exerting a greater inhibitory effect on the reaction rate.

We have incorporated these new rate measurements into the revised manuscript (Figure S23) and updated the corresponding section in the results and discussion (Line 357-367 on Page 20) to reflect the consistent trends observed with the reaction order analysis.

Figure S23. Dehydrogenation rates of cyclohexane, methylcyclohexane, and decalin over 0.05Pt/MAO and 3Pt/MAO catalysts at 300 °C under low (0) and high (0.5) product partial pressures.

“To validate the mechanistic changes associated with Pt coordination number,

0.05Pt/MAO and 3Pt/MAO were selected as representative catalysts for low and high CN_{Pt-Pt} , respectively, and their reaction orders were measured (Figure S20-S22). For all three dehydrogenation reactions, the negative reaction orders of the products indicate strong adsorption of unsaturated aromatic products, which inhibits the reaction (Figure 3c-e). Within the LHHW framework, the apparent reaction orders of the reactant and product are expressed as:

$$n_R = 1 - 2\theta_R$$

$$n_P = -2\theta_P$$

Since reaction orders directly reflect surface coverages, more negative values correspond to stronger adsorption. Accordingly, the more negative product reaction order observed on 0.05Pt/MAO compared to 3Pt/MAO highlights the stronger adsorption on low-coordinated Pt and its greater inhibitory effect on activity⁴⁷. To further validate the reaction order analysis, dehydrogenation rates were measured at high product concentrations (Figure S23), where the rate decrease caused by product inhibition was more pronounced for 0.05Pt/MAO.”

2. *Electrons from Pt were back-donated into the adsorbate, influencing the adsorption behavior of dehydrogenated products. However, such back-donation typically strengthens the interaction between Pt and the dehydrogenated product, thereby increasing the extent of product adsorption. Moreover, back-donation into the π^* orbital of a dehydrogenated product is likely to weaken the C=C bond, potentially promoting C-C bond cleavage, which may lead to the formation of byproducts and/or*

coke. The authors need to see this effect and C-H bond activation of a partially dehydrogenated product instead of C-H bond activation of a completely dehydrogenated product using further analysis. Moreover, the authors should address this possibility and revise their catalyst design strategy accordingly.

Reply:

We thank the reviewer for this important comment. We agree with the reviewer that back-donation from Pt to the π^* orbital of dehydrogenated products strengthens the adsorption. Our results are consistent with this, as the upward shift of the d-band center brings it closer to the π^* level, thereby enhancing d- π^* back-donation and reinforcing adsorption. Following the reviewer's suggestion, we calculated the C-C bond cleavage barriers for strongly adsorbed partially dehydrogenated species (C_6H_8 , C_7H_{10} , $C_{10}H_{14}$) on Pt(111) and Pt₁₃/MAO (Figure S41). The results show that due to the enhanced d- π^* back-donation, C-C cleavage barriers are indeed lower on Pt₁₃/MAO than on Pt(111). However, these barriers (about 2-3 eV) are approximately 2-3 times higher than the C-H dehydrogenation barriers (about 0.5-1.5 eV) and far above the product desorption energies, indicating that C-C cleavage is unlikely to occur under the reaction conditions. Moreover, if C-C scission were operative, gas-phase analysis would reveal light hydrocarbons (C_1 - C_5) or near- C_6 fragments (e.g., pentane). However, GC analysis detected no such products, providing experimental confirmation that C-C bond cleavage does not occur in our system.

Figure S41. Reaction pathways and free energy profiles for C-C bond cleavage of C_6H_8 , C_7H_{10} , $C_{10}H_{14}$ on Pt(111) and Pt₁₃/MAO.

We also note that our calculations consider the entire dehydrogenation process from reactants to partially dehydrogenated intermediates and final products, rather than C-H activation of the completely dehydrogenated products. We listed the C-H activation energy barriers for partially dehydrogenated intermediates (C_6H_{10} , C_7H_{12} , and $C_{10}H_{16}$), which basically follow a decreasing trend with stronger $d-\pi^*$ back-donation (Figure S37). While some individual steps deviate from this trend due to steric and structural deformation effects, the overall energy landscape (Figure 4a-c) clearly shows that the C-H activation energies follow the order: Pt₁/MAO < Pt₄/MAO < Pt₁₃/MAO < Pt(111) for all three dehydrogenation reactions. Therefore, our expanded analysis confirms that $d-\pi^*$ interactions modulate both adsorption and C-H activation while ruling out C-C bond cleavage as a competitive pathway. We have accordingly revised the manuscript to highlight these points and refined the discussion to enhance the logical rigor of the manuscript.

Figure S37. Adsorption energy of aromatic products (C₆H₆, C₇H₈, and C₁₀H₈) and C-H activation barriers of partially dehydrogenated intermediates (C₆H₁₀, C₇H₁₂, and C₁₀H₁₆) on various Pt catalyst models.

We have added these new calculations, results, and discussion to the revised manuscript (Figure S37, Figure S41, and Line 485-494 on Page 27-28).

“From an orbital interaction perspective, the upshifted d-band center narrows the energy gap between Pt d-orbitals and adsorbate antibonding orbitals (e.g., π^* for cycloalkenes). This enhances d- π^* back-donation, increasing bonding-state occupancy and decreasing antibonding-state occupancy, thereby strengthening Pt-adsorbate interactions⁵³. Such enhanced d- π^* back-donation explains the stronger adsorption observed on low-coordinated Pt sites. Consistently, the calculated C-H activation barriers of partially dehydrogenated intermediates (e.g., C₆H₁₀, C₇H₁₂, and C₁₀H₁₆; Figure S37) generally follow the expected decreasing trend with stronger d- π^* back-donation, with minor deviations attributed to steric and conformational effects. To examine possible side effects, we further calculated C-C cleavage barriers for partially dehydrogenated intermediates on Pt(111) and Pt₁₃/MAO (Figure S41). Although enhanced d- π^* back-donation on low-coordinated Pt slightly lowers the barriers, they

remain high far above C-H activation and desorption energies. Thus, C-C scission is kinetically inaccessible, confirming that the reaction pathway proceeds predominantly through successive C-H activation steps followed by product desorption.”

3. The authors calculated the Bader charges of fully dehydrogenated products (benzene, toluene, and naphthalene) to predict their interactions with the Pt d-band, concluding that $d-\pi^$ interactions influence both adsorption and C-H bond activation. However, this analysis primarily pertains to product desorption. To evaluate the effect on C-H bond activation, the authors should calculate the Bader charges of partially dehydrogenated intermediates to assess how $d-\pi^*$ interactions influence C-H activation. Additional analysis and corresponding discussion are necessary.*

Reply:

We thank the reviewer for this constructive suggestion. As also noted in our response to **Reviewer #1**, we have replaced the previously used descriptor of carbon electron density (Q_{Aromatic}) with the LUMO energy of aromatic products, which more directly reflects the electronic characteristics of the π^* orbitals. We agree that the original Bader charge analysis of fully dehydrogenated products primarily pertained to product desorption. To further assess the role of $d-\pi^*$ interactions in C-H bond activation, we have now calculated the LUMO energies of partially dehydrogenated intermediates ($\text{C}_{10}\text{H}_{14}$, $\text{C}_{10}\text{H}_{16}$, C_7H_{10} , C_7H_{12} , C_6H_8 , and C_6H_{10}). As shown in **Figure S44**, these intermediates exhibit the same electronic trend as the final products: lower LUMO energies facilitate $d-\pi^*$ back-donation from Pt and thereby lower the barriers

for C-H bond activation. This correlation is consistent with the observed variation in C-H activation barriers among different partially dehydrogenated intermediates (Figure S37b): where $C_6H_{12} > C_7H_{14} > C_{10}H_{18}$, consistent with our proposed mechanism. Importantly, the consistency of this trend across both partially dehydrogenated intermediates and final products indicates that d- π^* interactions not only govern adsorption/desorption but also critically modulate the kinetics of C-H bond activation.

We have added the calculated LUMO energies of partially dehydrogenated intermediates (Figure S44). The corresponding discussion has been revised in the main text (Line 526-528 on Page 29-30).

Figure S44. LUMO energy comparison of partially dehydrogenated intermediates.

“Bader charge analysis further supports this relationship, showing that molecules with higher carbon electron density exhibit lower LUMO energies (Figure S43). Notably, partially dehydrogenated intermediates (C_6H_8 , C_7H_{10} , and $C_{10}H_{14}$; C_6H_{10} ,

C₇H₁₂, and C₁₀H₁₆) follow the same trend (Figure S44), confirming the general applicability of LUMO energy as a reliable electronic descriptor.”

4. *The manuscript addresses an important mechanistic interplay between C–H bond activation and product desorption, modulated by Pt particle size and coordination environment. The use of $Q_{aromatic}$ as a descriptor is compelling and provides meaningful insight into the behavior of intermediates and products. However, the discussion on electron transfer into antibonding orbitals and the degree of their occupation would benefit from further clarification. Specifically, it would be helpful to explain—ideally with some quantitative basis—whether the observed occupation of antibonding states facilitates adsorption to an acceptable extent or, conversely, if excessive occupation could compromise adsorption strength. Such elaboration would significantly enrich the mechanistic understanding presented in the study.*

Reply:

We thank the reviewer for raising this important point. As reviewer suggested, the occupation of antibonding states can in principle have two opposite effects: moderate occupation may strengthen adsorption through enhanced covalency, while excessive occupation could weaken adsorption. To quantify this balance, we calculated the average Pt-C integrated crystal orbital Hamilton population (ICOHP) values for C₆H₆, C₇H₈, and C₁₀H₈ adsorbed on Pt(111), Pt₁₃/MAO, Pt₄/MAO, and Pt₁/MAO (Figure S45). The -ICOHP values systematically increase as the Pt coordination number decreases,

indicating progressively stronger Pt-adsorbate bonding due to enhanced d- π^* back-donation.

Importantly, when we compare these results with the active sites identified as optimal for different dehydrogenation reactions, we find that the corresponding -ICOHP values consistently fall within a moderate range of 1.2-1.4 (highlighted in Figure S45). This provides a quantitative basis for defining “appropriate adsorption”: within this window, d- π^* back-donation sufficiently strengthens Pt-adsorbate interactions to promote C-H activation, yet avoids excessive adsorption strength and ensures facile desorption.

These results enrich our mechanistic picture by providing a quantitative descriptor (-ICOHP) that directly links electronic structure, adsorption strength, and catalytic performance. We have incorporated this new analysis and discussion in the revised manuscript (Figure 5d, Figure S45, and Line 539-545 on Page 30).

Figure 5. (d) Mechanistic illustration of CN_{Pt-Pt} optimization guided by Pt-adsorbate orbital hybridization.

Figure S45. -ICOHP values of Pt-C bond for dehydrogenated products on various Pt catalyst models.

“To further quantify this interaction, crystal orbital Hamilton populations (COHP) analysis of Pt-C bonds in dehydrogenated products was performed (Figure S45). The results show that -ICOHP increases with decreasing Pt coordination, consistent with stronger covalent interactions from enhanced $d-\pi^*$ back-donation. Notably, optimal catalytic performance across different reactions occurs when -ICOHP falls within 1.2-1.4, reflecting a balance between sufficient stabilization for C-H activation and facile product desorption.”

5. The *R*-factor values reported in the EXAFS fitting appear to be relatively high, which raises some concern regarding the reliability of the fitting results. In particular, for the 0.05Pt/MAO sample, it appears that the fitting result of Pt–Pt coordination is not included in the model. Including this contribution would improve the accuracy of the structural interpretation and strengthen the overall analysis.

Reply:

We thank the reviewer for pointing out this important issue. We acknowledge that the R-factor values of the EXAFS fitting are relatively higher compared to typical values. This is mainly due to the low Pt loading, which reduces the signal-to-noise ratio and makes the background subtraction more challenging.

Regarding the reviewer's specific concern about the Pt-Pt contribution in the 0.05Pt/MAO sample, we have carefully re-fitted the EXAFS data to improve the fitting accuracy.

Table S7. EXAFS fitting parameters at the Pt K-edge for various Pt samples

($S_0^2=0.86$).

Sample	Shell	CN ^a	R(Å) ^b	σ^2 (Å ²) ^c	ΔE_0 (eV) ^d	R factor
Pt foil	Pt-Pt	12	2.76±0.01	0.003±0.001	8.48±0.31	0.001
0.02Pt/MAO	Pt-O	4.08±1.0	1.75±0.07	0.008±0.003	8.54±0.11	0.058
0.05Pt/MAO	Pt-O	3.75±0.8	1.86±0.03	0.004±0.001	7.14±3.9	0.009
	Pt-Pt	0.93±0.2	2.58±0.03	0.004±0.001		
	Pt-O	1.90±0.26	1.97±0.02	0.001±0.001	7.88±2.25	0.041
0.15Pt/MAO	Pt-Pt	5.30±0.58	2.71±0.05	0.003±0.001		

^aCN, coordination number; ^bR, distance between absorber and backscatter atoms; ^c σ^2 , Debye-Waller factor to account for both thermal and structural disorders; ^d ΔE_0 , inner potential correction; R factor indicates the goodness of the fit. Fitting range: $2 < k$ (Å) < 10 and $1.1 < R$ (Å) < 3.0 .

Figure S9. EXAFS fitting curves in k-space for the Pt L_{III}-edge EXAFS data of various Pt samples. Black dots and red line represent the original and fitting results, respectively.

Figure S10. EXAFS fitting curves in R-space for the Pt L_{III}-edge EXAFS data of various Pt samples. Black dots and red line represent the original and fitting results, respectively.

6. Figure S11, the blue-shaded region is not clearly explained in the manuscript. The authors need to provide a more detailed explanation of the significance and interpretation of this region.

Reply:

We thank the reviewer for this helpful comment. In the revised manuscript, we have removed the blue-shaded region in Figure S11 (Figure S16 in the revised version) because it does not provide additional mechanistic significance and could potentially cause confusion. We believe that the figure is now clearer and more straightforward, and no further discussion of the shaded region is required.

Figure S16. Arrhenius plots for calculation of activation energy over various Pt/MAO catalysts for (a) cyclohexane dehydrogenation, (b) methylcyclohexane dehydrogenation, and (c) decalin dehydrogenation.

7. Line 261 (p15): The authors claimed that the stability come from oxygen vacancy-mediated SMSI, which will suppress Pt sintering. However, there is no data on Pt size distribution after the 100 h of dehydrogenation.

Reply:

We thank the reviewer for raising this important point. To address the concern, we conducted additional HAADF-STEM measurements on the spent 0.15Pt/MAO catalyst after 100 h of dehydrogenation. As shown in Figure S14, Pt species are still predominantly present as sub-nanometer clusters, with only a minor fraction of nanoparticles observed. The particle size distribution analysis yields an average size of 0.8 ± 0.16 nm, which is very close to that of the fresh 0.15Pt/MAO catalyst (0.7 ± 0.15 nm). This result confirms that significant Pt sintering did not occur during the long-term reaction, thereby supporting our conclusion that oxygen vacancy-mediated SMSI effectively stabilizes the Pt nanoclusters under reaction conditions.

We have incorporated these results and corresponding discussion in the revised manuscript (Figure S14, Line 284-288 on Page 16).

Figure S14. HAADF-STEM images and particle size distribution of spent 0.15Pt/MAO catalyst after 100 h dehydrogenation.

“The robust stability of Pt/MAO can be attributed to two key factors: oxygen vacancy-mediated strong metal-support interactions in the defect-engineered MAO, which effectively suppress Pt species sintering during prolonged reactions (Figure S14), and the excellent coke resistance, as evidenced by TG analysis of the spent catalyst revealing negligible carbon deposition after 100 h (Figure S15).”

8. There are a few typographical errors that should be corrected to enhance clarity and precision. For instance, in Figure 3(b), “Methylcyclohexane” should be revised to “Decalin.” Please review the manuscript carefully for other minor errors as well.

Reply:

We thank the reviewer for pointing out this error. In the revised manuscript, we have corrected “methylcyclohexane” to “decalin” in Figure 3(b). In addition, we have carefully proofread the entire manuscript and SI to identify and correct other minor typographical errors, thereby improving overall clarity and accuracy.

“Figure 3. (b) TPSR profiles for decalin dehydrogenation on 0.05Pt/MAO and 3Pt/MAO catalysts.”

9. Please check whether the following is needed: SI file, line 218, Figure S5  Figure S6 / line 144: 2059  2061

Reply:

We thank the reviewer for carefully checking the manuscript and pointing out these errors. We have corrected them in the revised version:

“For direct comparison, TOF values were further normalized to the maximum TOF within each site category, as presented in Figure S6.”

“As loading increased, the vibrations of Pt-CO gradually red-shifted, with the emergence of bands at 2061 cm^{-1} and 2057 cm^{-1} corresponding to CO adsorption on Pt clusters and Pt nanoparticles³⁶, respectively.”

Reviewer #3 (Remarks to the Author):

In catalysis research, establishing volcano plots that correlate descriptors with activity has been an intriguing topic across reactions like NH₃ synthesis/decomposition, ORR, OER, methanation, and formic acid dehydrogenation. In this context, this study is among the first to introduce the "volcano plot" concept into Liquid Organic Hydrogen Carrier (LOHC) research—a field gaining significant attention for enabling CO₂-free hydrogen storage and release. While this contribution is timely and offers novelty, it requires substantial revision before reconsideration for publication. My review focuses on experimental aspects preceding computational analysis; I will evaluate the computational section once the issues below are addressed.

Reply:

We sincerely thank the reviewer for the positive and encouraging comments on the novelty and timeliness of our work. At the same time, we fully acknowledge the reviewer's concerns regarding the experimental aspects. Following the reviewer's suggestions, we have carefully revised and strengthened the experimental section through additional data, clarifications, and corrections to improve the overall rigor and reliability of our study. We believe that these revisions address the reviewer's concerns and provide a more solid basis for the subsequent computational discussion.

1) Catalyst Characterization

1-1) Support Effect

The authors should justify why MgAl₂O₄ was chosen for mechanistic studies.

Given that Pt–Pt coordination number (CN) primarily reflects active-metal properties, support effects on particle size/distribution and CN must be systematically ruled out. Although the defect-rich spinel surface with oxygen vacancies reportedly stabilizes ultrasmall Pt nanoparticles and modulates electronic structure via ligand effects, a direct comparison with more inert supports is needed to isolate these influences.

Reply:

We thank the reviewer for raising this important point. To clarify the role of the support, we synthesized a series of Pt/Al₂O₃ catalysts with different loadings (0.02-1wt%) as a control system. HAADF-STEM characterization revealed that, at the same loading, Pt species on Al₂O₃ are significantly larger than those on MgAl₂O₄ (Figure S11). For instance, while 0.02Pt/Al₂O₃ can maintain atomic dispersion, at only 0.05 loading the Pt species mainly evolve into nanoclusters accompanied by nanoparticles, without showing the single atom-cluster ensemble structure observed on MAO support. At higher loadings (>0.15), Pt predominantly aggregates into nanoparticles. These results clearly demonstrate the advantage of MAO in stabilizing highly dispersed Pt species through strong metal-support interaction.

Figure S11. HAADF-STEM images and particle size distributions of Pt/Al₂O₃

catalysts with varying loadings.

Catalytic evaluation further shows that Pt/Al₂O₃ catalysts exhibited much lower dehydrogenation activity than Pt/MAO (Figures S12a). Moreover, it suffered rapid deactivation, for example, >30% activity loss within 50 min for 0.375Pt and 1Pt/Al₂O₃ (Figures R2). This behavior contrasts sharply with the long-term durability of Pt/MAO, highlighting the critical role of strong Pt-support interactions. Nevertheless, when normalizing activities at ~20 min to minimize deactivation effects, the activity trends on Pt/Al₂O₃ still display volcano-type relationships (Figure S12b). Specifically, decalin dehydrogenation shows optimal performance on 0.15Pt/Al₂O₃ (particle-dominated), while cyclohexane and methylcyclohexane exhibit maximum rates on 0.05Pt/Al₂O₃ (cluster-dominated). TOF calculations confirm that in the nanoparticle regime, TOF remains essentially constant, consistent with our observations on MAO. The optimum coordination for methylcyclohexane dehydrogenation corresponds to 0.05Pt/Al₂O₃, whose structural features (from HAADF) resemble those of 0.15Pt/MAO, both with an average CN of ~5. For cyclohexane dehydrogenation, although the highest activity also appears at 0.05Pt/Al₂O₃, its TOF is already close to that of nanoparticles, implying that the true optimum lies between 0.02 and 0.05Pt/Al₂O₃. This agrees well with our Pt/MAO study, where the best performance corresponds to a CN of ~2.5.

Figure S12. (a) Hydrogen evolution rates and (b) site-specific TOFs of Pt/Al₂O₃ catalysts for the dehydrogenation of cyclohexane, methylcyclohexane, and decalin at 280 °C.

Figure R2. Time-dependent hydrogen evolution rates of Pt/Al₂O₃ catalysts for the dehydrogenation of cyclohexane at 280 °C.

Taken together, these results indicate that the reactant-dependent volcano-type dependence on Pt coordination number is intrinsic to the reaction and not support-dependent. However, due to the limited ability of Al₂O₃ to stabilize sub-nanometer Pt clusters with specific CN, it is very challenging to probe the low-coordination regime on such inert supports. By contrast, MAO uniquely enables the stabilization of atom-cluster ensembles and low-CN Pt species, making it particularly suitable for

mechanistic studies. Importantly, beyond structural stabilization, MAO also delivers markedly higher dehydrogenation performance compared to Al₂O₃ across all sizes, highlighting its dual advantage in both activity and durability.

We have included these new results and discussions in the revised manuscript (Line 258-273 on Page 15-16; Figures S11-12; Line 196-197 on Page 9 of SI).

“To clarify the role of the support in the observed volcano-type relationship, Pt/Al₂O₃ catalysts were prepared with different loadings (0.02-1 wt%). HAADF-STEM revealed that Pt on Al₂O₃ dispersed as single atoms at 0.02Pt, but rapidly grew into clusters with nanoparticles at 0.05Pt, and aggregated predominantly into nanoparticles above 0.15Pt, in contrast to the higher dispersion maintained on MAO (Figure S11). Catalytically, Pt/Al₂O₃ exhibited significantly lower dehydrogenation activity than Pt/MAO across all particle sizes, underscoring the unique ability of MAO in stabilizing highly dispersed Pt and enhancing intrinsic activity via electronic modulation (Figure S12a). Nevertheless, Pt/Al₂O₃ catalysts still exhibited reactant-dependent volcano-type trends: decalin dehydrogenation peaked at 0.15Pt/Al₂O₃ (nanoparticle-dominated), methylcyclohexane at 0.05PtPt/Al₂O₃, closely resembling the 0.15Pt/MAO with CN_{Pt}≈4.7, while cyclohexane reached its optimum between 0.02 and 0.05Pt/Al₂O₃, aligning with the ~2.5 CN optimum identified on MAO (Figure S12b). These results indicate that the reactant-dependent volcano relationship between CN_{Pt-Pt} and activity is support-independent, but the ability of MAO to stabilize low-coordinated Pt clusters makes it uniquely suitable for systematically probing this relationship.”

“Pt/MgAl₂O₄ and Pt/Al₂O₃ catalysts were prepared using the incipient wetness

impregnation method.”

1-2) Inhomogeneity of Pt Sites

Peak broadening and satellite features in characterization data indicate heterogeneous Pt active sites. While average Pt–Pt CN is used as a key descriptor, the standard deviation and distribution profiles of active sites likely impact catalytic performance. Prior studies on LOHC dehydrogenation deconvoluted contributions from distinct Pt facets, which should be incorporated here.

Reply:

We appreciate the reviewer’s insightful comment regarding the inhomogeneity of Pt active sites and the possible influence of distinct facets. We fully agree that the catalysts exhibit a distribution of Pt particle sizes, and we are aware of this inherent complexity. To address this, we applied a multisite kinetic approach to deconvolute the contributions from different active sites and extract kinetic data for sites with relatively well-defined CN_{Pt-Pt} , as shown in Figure 2c.

- 0.02Pt/MAO: predominantly atomically dispersed Pt ($CN_{Pt-Pt} \approx 0$).
- 0.05Pt/MAO: a mixture of atomically dispersed Pt (63%) and small clusters ($CN_{Pt-Pt} \approx 2.5$, 37%).
- 0.15Pt/MAO: coexistence of medium clusters ($CN_{Pt-Pt} \approx 4.7$, 56%) and Pt nanoparticles ($CN_{Pt-Pt} \approx 7$, 44%).
- 0.375Pt/MAO, 1Pt/MAO, and 3Pt/MAO catalysts: dominated by Pt nanoparticles ($CN_{Pt-Pt} \approx 7$).

For the nanoparticle regime (0.375-3 wt% Pt/MAO), we further deconvoluted the potential contributions of different facets using a truncated octahedron model, in which activities were normalized to the number of (111), (100), surface, edge, or corner atoms. The TOF remained constant only when normalized to edge Pt atoms, demonstrating that under-coordinated edge sites ($CN_{Pt-Pt} \approx 7$) are the dominant active centers for dehydrogenation. This conclusion is consistent with prior LOHC studies that explicitly distinguished facet contributions, where low-coordination edge sites were identified as the most active centers (*Appl. Catal. B: Environ.*, 2020, 266, 118658; *Appl. Catal. B: Environ.*, 2021, 288, 119996). Taken together, our results not only corroborate these facet-dependent studies in the nanoparticle regime but also extend the analysis to single atoms and sub-nanometer clusters, allowing us to establish a unified volcano-type relationship between Pt coordination number and dehydrogenation activity across the entire dispersion range.

Based on these observations, we selected four representative Pt active sites for detailed kinetic evaluation:

1. Atomically dispersed Pt ($CN_{Pt-Pt} \approx 0$)
2. Small clusters ($CN_{Pt-Pt} \approx 2.5$)
3. Medium clusters ($CN_{Pt-Pt} \approx 4.7$)
4. Nanoparticles ($CN_{Pt-Pt} \approx 7$)

Using the multisite kinetic approach, we extracted intrinsic rates and kinetic parameters for these four active-site ensembles, as presented in Figure 2d. This framework allows us to reliably use CN_{Pt-Pt} as a quantitative descriptor to systematically

investigate the mechanistic and kinetic dependence of catalytic performance on Pt coordination.

We have clarified this point in the revised manuscript by (i) providing a detailed explanation that the CN_{Pt-Pt} values used in this work correspond to single atoms, clusters, and nanoparticles individually rather than averaged distributions, and (ii) emphasizing that edge atoms ($CN_{Pt-Pt} \approx 7$) are the dominant active sites for nanoparticle catalysts, consistent with the literature. These revisions are included in the main text (Line 191-196 on Page 12; Line 239-244 and Line 250-252 on Page 14-15).

“The turnover frequency (TOF) for each catalyst in the three dehydrogenation reactions was systematically investigated to elucidate the dependence of the site activity on the Pt coordination number. HAADF-STEM clearly showed size distribution varied from atomically dispersed Pt to clusters and nanoparticles with increase in Pt loading. For each sample there is also inhomogeneity of Pt active sites. To address this, we applied a least-squares kinetic fitting approach to deconvolute the contributions from different Pt species and extract kinetic data for specific Pt species (Figure 2c) (see calculation details in Supporting Information)³⁸⁻⁴⁰.”

“Collectively, data fitting identified four representative active-site ensembles: atomically dispersed Pt ($CN_{Pt-Pt} \approx 0$), small clusters ($CN_{Pt-Pt} \approx 2.5$), medium clusters ($CN_{Pt-Pt} \approx 4.7$), and nanoparticles (CN_{Pt-Pt} of edge Pt atom ≈ 7). Intrinsic TOF and kinetic parameters were extracted for each ensemble, enabling the reliable use of CN_{Pt-Pt} as a quantitative descriptor to systematically investigate the mechanistic and kinetic dependence of catalytic performance on Pt coordination.”

“Our previous work demonstrated that in Pt catalysts featuring particle sizes of 1.0-1.8 nm, edge-site Pt atoms with a CN_{Pt-Pt} of 7 act as the predominant active centers for decalin dehydrogenation⁴³.”

1-3)CN vs. Facet Distribution

Facet distribution is proposed as a descriptor for intrinsic activity (TOF), but identical average CN values may mask differing facet distributions. The methodology for quantifying facet distributions across catalysts is inadequately described (both main text and SI). Additionally, the preparation protocol and interpretation of Figure S6 require clarification.

Reply:

We thank the reviewer for the constructive comment. For catalysts containing significant non-particle components (e.g., 0.05Pt/MAO with single atoms and clusters, and 0.15Pt/MAO with clusters and nanoparticles), facet-based analysis is not applicable, since such species do not possess well-defined crystalline planes. In these cases, the average Pt-Pt coordination number (CN_{Pt-Pt}) was adopted as a more appropriate descriptor that captures their intrinsic structural characteristics independent of facet distributions.

For particle-dominated catalysts ($\geq 0.375Pt/MAO$), we did not rely on average CN values but instead adopted an explicit methodology to quantify facet distributions. The nanoparticle morphology was approximated as a truncated octahedron, and for each particle size we enumerated the number and fraction of Pt atoms located on different

sites, including (111), (100), surface, edge, and corner positions. This analysis allowed us to determine the size-dependent distributions of Pt sites across nanoparticles. By correlating catalytic activity with the abundance of each site type, we found that only when activity was normalized to the number of edge atoms did the TOF remain essentially constant across the 0.375-3Pt/MAO series. This invariance clearly identifies edge Pt atoms (CN=7) as the dominant active sites within nanoparticles.

Thus, the role of facet analysis in our study is not to compare activity differences between specific extended surfaces, but rather to confirm that once particles dominate, the catalytic behavior is governed by low-coordination edge atoms. This is consistent with our unified volcano framework, where CN_{Pt-Pt} acts as a single descriptor that links atomic-scale species (single atoms, clusters) with nanoparticle edge structures.

To address the reviewer's concern, we have now added a detailed description of this methodology and clarified the interpretation of Figure S6 in the SI (Line 281-306 on Page 13-14). In addition, the main text has been expanded accordingly (Line 199-202 on Page 12) to ensure that the methodology and conclusions are clearly presented.

“For the 0.375Pt/MAO, 1Pt/MAO, and 3Pt/MAO catalysts, Pt is predominantly present as nanoparticles. The truncated octahedral model was applied to identify the dominant active sites. Given the measured particle diameter d , equation 5 yields the total number of Pt atoms per particle (N_t):

$$d = 1.105 \times d_{atom} \times N_t^{1/3} \quad (5)$$

where d_{atom} is the diameter of a Pt atom (0.276 nm). Then the number of atoms along the particle edge (n) can be solved based on:

$$N_t = 16n^3 - 33n^2 + 24n - 6 \quad (6)$$

After which the numbers of site-specific atoms were calculated as follows:

$$N_s = 30n^2 - 60n + 32 \quad (7)$$

$$N_c = 24 \quad (8)$$

$$N_e = 36(n - 2) \quad (9)$$

$$N_{100} = 6(n - 2)^2 \quad (10)$$

$$N_{111} = 8(3n^2 - 9n + 7) \quad (11)$$

where N_s , N_c , N_e , N_{100} , and N_{111} represent the numbers of surface, corner, edge, (100), and (111) Pt atoms, respectively. Using the mole number of Pt in each catalyst, the absolute numbers of Pt atoms at different sites were obtained. TOF was then calculated by normalizing catalytic activity to the number of each type of site. For direct comparison, TOF values were further normalized to the maximum TOF within each site category, as presented in Figure S6.

Only when normalized to edge atoms did the TOF remain essentially constant across different particle sizes, suggesting that edge Pt atoms serve as the primary active sites for cycloalkane dehydrogenation on Pt nanoparticles. Consequently, the TOF of Pt nanoparticles (TOS_n) is identical across these catalysts and calculated as:

$$TOS_n = \frac{F_{H_2, Pt/MAO}}{m_{edge Pt}} \quad (12)$$

where $F_{H_2, Pt/MAO}$ is the molar flow rates of the H_2 for 0.375Pt/MAO, 1Pt/MAO and 3Pt/MAO catalyst, and $m_{edge Pt}$ is the mass of edge Pt atoms in these catalysts.”

“In the 0.375Pt/MAO, 1Pt/MAO, and 3Pt/MAO catalysts, Pt was predominantly present as nanoparticles, enabling site-specific TOF analysis using the truncated

octahedral model to identify the dominant active sites^{41, 42}. For each nanoparticle size, the populations of atoms on (111), (110), surface, edge, and corner sites were explicitly enumerated (see calculation details in Supporting Information). Catalytic activity was then normalized to the number of each type of site (Figure S6).”

2) Performance Evaluation

2-1) LOHC Selection Rationale

Testing cyclohexane (single ring), methylcyclohexane (methyl-substituted ring), and decalin (fused bicyclic) introduces confounding variables (ring count, substituents, fusion topology). To robustly generalize volcano plots across LOHCs, a controlled series (e.g., saturated benzene→naphthalene→anthracene or biphenyl→terphenyl) would better isolate structural effects. Alternatively, industrially relevant LOHCs like toluene, monobenzyltoluene, or dibenzyltoluene would enhance practical relevance.

Reply:

We appreciate the reviewer’s constructive suggestion regarding the rationale of LOHC selection. In the revised manuscript, we have substantially strengthened this part. In addition to cyclohexane (C₆H₁₂), methylcyclohexane (C₇H₁₄), decalin (C₁₀H₁₈) and perhydro-dibenzyltoluene (C₂₁H₃₈) that were originally included as widely studied model LOHCs, we now incorporated perhydro-benzyltoluene (C₁₄H₂₆) as an additional substrate (Figure S42). Importantly, perhydro-benzyltoluene, together with methylcyclohexane and perhydro-dibenzyltoluene, represents a family of structurally related LOHCs and also constitutes one of the most industrially relevant LOHC systems

currently deployed in commercial practice (*Acc. Chem. Res.*, 2017, 50, 74-85; *Energ. Environ. Sci.*, 2017, 10, 1652-1659; *Sci. Adv.*, 2022, 8, 3262). While we acknowledge the reviewer's valuable suggestion to examine saturated anthracene or biphenyl/terphenyl derivatives, these substrates were not accessible from commercial sources, and thus we were unable to experimentally include them in this study.

Figure S42. The molecular structure of dehydrogenation products: benzene (C_6H_6), toluene (C_7H_8), benzyltoluene ($C_{14}H_{14}$), dibenzyltoluene ($C_{21}H_{20}$), and naphthalene ($C_{10}H_8$).

The inclusion of perhydro-benzyltoluene allows us to form a controlled series with increasing aromatic complexity: cyclohexane \rightarrow methylcyclohexane \rightarrow perhydro-benzyltoluene \rightarrow perhydro-dibenzyltoluene \rightarrow decalin. We determined the TOF dependence on Pt coordination number (Figure S51b) and, by polynomial fitting, identified the optimal CN values as ~ 2.5 , 4.5, 5.1, 5.8, and 7 for cyclohexane, methylcyclohexane, perhydro-benzyltoluene, perhydro-dibenzyltoluene, and decalin, respectively. Moreover, we computed the LUMO energy of benzyltoluene (2.85) (Figure S46) and integrated it into the existing correlations between optimal CN, Pt d-properties, and LUMO energy (Figure 5d-e, Figure S52). The new data fall well within

the established scaling relationships, thereby confirming the robustness of the descriptor.

Figure S51. (a) Hydrogen evolution rates of Pt/MAO catalysts for the dehydrogenation of cyclohexane, methylcyclohexane, perhydro-benzyltoluene, perhydro-dibenzyltoluene and decalin at 280 °C. (b) Site-specific TOF as a function of CN_{Pt-Pt} for different dehydrogenation reactions at 280 °C with polynomial fitting.

Figure S46. Projected density of states and LUMO energy of 2-, 3-, 4-benzyltoluene

Figure S48. Adsorption configurations and energies of benzyltoluene isomers on

Pt(111).

Figure 5 (e) Correlation of Pt d-band center and optimal CN_{Pt-Pt} with LUMO energy

of aromatic product in cycloalkane dehydrogenation.

Figure S52. Correlation between the Pt 5d-electron vacancy of optimal catalyst and LUMO energy of aromatic product for different cycloalkane dehydrogenation reactions.

We would like to emphasize that our study aims to establish a generalizable descriptor applicable across structurally diverse LOHCs, rather than to restrict to a homologous series. The descriptor LUMO energy can effectively integrate the influences of multiple structural variables (ring number, substituents, fusion topology) into a single parameter, enabling predictive capability across a broad range of LOHCs, including emerging carriers such as methylcyclohexane derivatives. We fully agree with the reviewer that the individual contributions of confounding variables are highly complex and merit further detailed studies. We have also acknowledged this limitation in the revised manuscript (Line 571-573 on Page 32) and outlined it as an important direction for future research.

“Importantly, these results establish LUMO energy as a universal descriptor for optimizing CN_{Pt-Pt} in the dehydrogenation of various cycloalkanes. Nevertheless, structural factors such as ring number, substituents, and fusion topology may exert inherently complex influences, underscoring a significant direction for future refinement of the descriptor framework.”

Finally, the initial selection of cyclohexane and decalin was motivated by their broad use as benchmark model LOHCs in the literature, facilitating direct comparison with prior studies (Figure 2b). Together with methylcyclohexane, benzyltoluene, and dibenzyltoluene, we now cover both academically important benchmarks and

industrially relevant systems, providing a comprehensive platform to validate the universality of our descriptor.

We have revised the main text to explicitly highlight these points and added new results and discussion accordingly (Line 541-5555 on Page 30-31, Figure 5c, Figure 5e, Figure S46, 48, 50, 51, 52, Line 253-254 on Page 12 of SI).

“To validate this mechanism, we extended our investigation to two industrially important liquid organic hydrogen carriers: benzyltoluene and dibenzyltoluene. The presence of benzyl substituents increases the electron density of aromatic rings relative to toluene, as reflected by the larger fraction of elongated C-H bonds, albeit less pronounced than in naphthalene (Figure S42). LUMO energy calculations yield average values of 4.24 eV for benzyltoluene isomers (2-, 3-, 4-) (Figure S46) and 2.85 eV for dibenzyltoluene isomers (2,3-, 2,4-, and 2,5-) (Figure S47), higher than in toluene but lower than in naphthalene. DFT calculations further showed that the adsorption energies of benzyltoluene isomers (-0.77 to -0.89 eV) and dibenzyltoluene isomers (-0.82 to -1.03 eV) on Pt(111) fall within a narrow range (Figure S48-49) and lie between those of toluene and naphthalene (Figure S50). Catalytic performance testing on Pt catalysts with varying CN_{Pt-Pt} revealed that the optimal CN_{Pt-Pt} for perhydro-benzyltoluene and perhydro-dibenzyltoluene dehydrogenation are approximately 5.1 and 5.8, respectively (Figure S51). As shown in Figure 5e, these values fall precisely on the established correlation between optimal CN_{Pt-Pt} and LUMO energy of aromatic product.”

“Cycloalkane reactants—cyclohexane ($268.2 \mu\text{L min}^{-1}$), methylcyclohexane ($369.3 \mu\text{L min}^{-1}$), decalin ($191.8 \mu\text{L min}^{-1}$), perhydro-benzyltoluene ($60.42 \mu\text{L min}^{-1}$) or

perhydrodibenzyltoluene ($285.6 \mu\text{L min}^{-1}$)—were fed into the reactor via a peristaltic pump and preheated to $200 \text{ }^\circ\text{C}$ before entering the reaction zone.”

2-2)Pt Speciation and TOF Calculation

The derivation of Pt cluster fractions and TOF via kinetic data/least-squares fitting lacks transparency. Detailed methodological steps, theoretical foundations, and validated literature precedents must be provided. Given active-site heterogeneity, this analysis should account for site-specific distributions.

Reply:

We thank the reviewer for raising this important point. We agree that the derivation of Pt speciation and TOF must be presented in a transparent and rigorous manner. In the revised manuscript, we have substantially clarified this part. Specifically, the complete set of governing equations, detailed calculation steps, and software specifications for the least-squares fitting procedure are now provided in the SI.

Regarding the theoretical foundation, our approach is built upon validated literature precedent. For example, Ayman M. Karim et al. (*J. Catal.*, 2019, 378, 121-130; *Ind. Eng. Chem. Res.*, 2021, 60, 15960-15971) employed a similar least-squares kinetic fitting strategy to disentangle the synergistic contributions of supported Ir single atoms and nanoparticles during CO oxidation. Their work demonstrated the feasibility of deconvoluting site-specific fractions and activities directly from kinetic data in heterogeneous catalysts. Building upon these approaches, our methodology extends the

concept to three categories of Pt species (single atoms, clusters, and nanoparticles) and integrates them with CN_{Pt-Pt} as a unifying structural descriptor.

In response to the reviewer's suggestion, we have expanded the main text to clarify this methodology (Line 214-217 on Page 13) and added detailed calculation equation and process in SI (Line 307-369 on Page 14-17).

“The remaining unknowns—the proportions of Pt single atoms, clusters, and nanoparticles, together the TOF for Pt clusters in the 0.05Pt/MAO and 0.15Pt/MAO catalysts—were determined using the equations in Figure 2c through the least-squares fitting method (see calculation details in Supporting Information). The experimentally measured rates across multiple temperatures and dehydrogenation reactions were used as fitting inputs, with unknown parameters optimized by minimizing residual error, following validated least-squares kinetic deconvolution approaches in the literature^{38, 44}.”

“The TOF of Pt clusters (TOS_c) were derived using kinetic data from 0.05Pt/MAO and 0.15Pt/MAO catalysts. These catalysts contain mixtures of Pt single atoms, clusters, and nanoparticles, with unknown fractions and distinct cluster characteristics. To resolve the contributions of each species, a least-squares fitting approach was employed based on the following equations.

$$f_s + f_c + f_n = 100\% \quad (13)$$

where f_s , f_c , f_n represent the mass fractions of Pt single atoms, clusters and nanoparticles in the 0.05Pt/MAO or 0.15Pt/MAO catalyst.

$$TOS_c = A \exp\left(-\frac{E_a}{RT}\right) P_{reactant}^a \quad (14)$$

where A denotes the pre-exponential factor, $P_{reactant}$ represents the partial pressures of the reactant (cyclohexane, methylcyclohexane, or decalin), a is the reaction order with respect to the reactant. During kinetic measurements, the conversion was maintained <20% specifically to avoid complications from product inhibition effects. The reaction orders of reactant can be obtained from the reaction order results of 0.05Pt/MAO in Figure 3c-e.

$$RSS = \sum [TOF_{i,T} - (f_s TOS_s + f_c TOS_c + f_n x TOS_n)]^2 \quad (15)$$

where $TOF_{i,T}$ is the experimentally measured TOF of 0.05Pt/MAO or 0.15Pt/MAO catalyst during the dehydrogenation of cyclohexane, methylcyclohexane, or decalin (denoted by i) at temperatures T (543, 553, 563 and 573 K). The parameter x represents the mass fraction of edge Pt atoms within the Pt nanoparticles in the 0.05Pt/MAO or 0.15Pt/MAO catalysts. The model comprises 13 equations with 6 unknowns (f_s , f_c , f_n , E_a , A and x), as listed below.

For 0.05Pt/MAO:

$$f_s + f_c = 100\%$$

$$f_s \times 545.39 + f_c \times e^{\ln A - \frac{E_a}{R \times 543.15}} \times 0.7^{0.41} - 1874.85 = 0$$

$$f_s \times 767.69 + f_c \times e^{\ln A - \frac{E_a}{R \times 553.15}} \times 0.7^{0.41} - 3284.58 = 0$$

$$f_s \times 1162.64 + f_c \times e^{\ln A - \frac{E_a}{R \times 563.15}} \times 0.7^{0.41} - 5037.47 = 0$$

$$f_s \times 1616.21 + f_c \times e^{\ln A - \frac{E_a}{R \times 573.15}} \times 0.7^{0.41} - 6848.69 = 0$$

$$f_s \times 385.25 + f_c \times e^{\ln A - \frac{E_a}{R \times 543.15}} \times 0.7^{0.42} - 1093.49 = 0$$

$$f_s \times 467.71 + f_c \times e^{\ln A - \frac{E_a}{R \times 553.15}} \times 0.7^{0.42} - 1469.04 = 0$$

$$f_s \times 583.75 + f_c \times e^{\ln A - \frac{E_a}{R \times 563.15}} \times 0.7^{0.42} - 2655.11 = 0$$

$$f_s \times 699.10 + f_c \times e^{\ln A - \frac{E_a}{R \times 573.15}} \times 0.7^{0.42} - 4275.61 = 0$$

$$f_s \times 59.40 + f_c \times e^{\ln A - \frac{E_a}{R \times 543.15}} \times 0.5^{2.03} - 119.67 = 0$$

$$f_s \times 83.85 + f_c \times e^{\ln A - \frac{E_a}{R \times 553.15}} \times 0.5^{2.03} - 162.03 = 0$$

$$f_s \times 102.46 + f_c \times e^{\ln A - \frac{E_a}{R \times 563.15}} \times 0.5^{2.03} - 263.77 = 0$$

$$f_s \times 125.61 + f_c \times e^{\ln A - \frac{E_a}{R \times 573.15}} \times 0.5^{2.03} - 339.33 = 0$$

For 0.15Pt/MAO:

$$f_s + f_c + f_n = 100\%$$

$$f_s \times 545.39 + f_c \times e^{\ln A - \frac{E_a}{R \times 543.15}} \times 0.7^{0.41} + f_n \times x \times 2285.03 - 2263.38 = 0$$

$$f_s \times 767.69 + f_c \times e^{\ln A - \frac{E_a}{R \times 553.15}} \times 0.7^{0.41} + f_n \times x \times 3170.67 - 3166.23 = 0$$

$$f_s \times 1162.64 + f_c \times e^{\ln A - \frac{E_a}{R \times 563.15}} \times 0.7^{0.41} + f_n \times x \times 4224 - 4461.31 = 0$$

$$f_s \times 1616.21 + f_c \times e^{\ln A - \frac{E_a}{R \times 573.15}} \times 0.7^{0.41} + f_n \times x \times 5492.56 - 5490.79 = 0$$

$$f_s \times 322.74 + f_c \times e^{\ln A - \frac{E_a}{R \times 533.15}} \times 0.7^{0.42} + f_n \times x \times 1111.69 - 719.50 = 0$$

$$f_s \times 385.25 + f_c \times e^{\ln A - \frac{E_a}{R \times 543.15}} \times 0.7^{0.42} + f_n \times x \times 1598.87 - 1245.60 = 0$$

$$f_s \times 467.71 + f_c \times e^{\ln A - \frac{E_a}{R \times 553.15}} \times 0.7^{0.42} + f_n \times x \times 2582.31 - 2250.93 = 0$$

$$f_s \times 583.75 + f_c \times e^{\ln A - \frac{E_a}{R \times 563.15}} \times 0.7^{0.42} + f_n \times x \times 3849.59 - 3307.76 = 0$$

$$f_s \times 59.40 + f_c \times e^{\ln A - \frac{E_a}{R \times 543.15}} \times 0.5^{2.03} + f_n \times x \times 223.18 - 126.11 = 0$$

$$f_s \times 83.85 + f_c \times e^{\ln A - \frac{E_a}{R \times 553.15}} \times 0.5^{2.03} + f_n \times x \times 394.10 - 189.87 = 0$$

$$f_s \times 102.46 + f_c \times e^{\ln A - \frac{E_a}{R \times 563.15}} \times 0.5^{2.03} + f_n \times x \times 797.90 - 304.45 = 0$$

$$f_s \times 125.61 + f_c \times e^{\ln A - \frac{E_a}{R \times 573.15}} \times 0.5^{2.03} + f_n \times x \times 1046.72 - 599.87 = 0$$

All equations were implemented in Python (PyCharm), and the parameters were

obtained by performing least-squares fitting to minimize equation 15, yielding the

optimized values of f_s , f_c , f_n , E_a , A and x . This approach allows for the

quantitative determination of the fractions of Pt species and their respective TOFs,

providing insights into the contributions of Pt single atoms, clusters, and nanoparticles

to the overall catalytic performance.

- 0.02Pt/MAO: predominantly atomically dispersed Pt ($CN_{Pt-Pt} \approx 0$).
- 0.05Pt/MAO: a mixture of atomically dispersed Pt (63%) and small clusters ($CN_{Pt-Pt} \approx 2.5$, 37%).
- 0.15Pt/MAO: coexistence of medium clusters ($CN_{Pt-Pt} \approx 4.7$, 56%) and Pt nanoparticles ($CN_{Pt-Pt} \approx 7$, 44%).
- 0.375Pt/MAO, 1Pt/MAO, and 3Pt/MAO catalysts: dominated by Pt nanoparticles ($CN_{Pt-Pt} \approx 7$).

2-3) Long-Term Stability Testing

While the best-performing catalyst showed stability at 300°C over >100 h, testing additional catalysts under identical conditions is essential to validate the protocol's efficacy. Hydrogen purity analysis (e.g., quantifying methane from demethylation/ring-opening side reactions) must be included to assess selectivity.

Reply:

We thank the reviewer for this constructive suggestion. In response, we have expanded the long-term stability evaluation to include three additional catalysts (1Pt/MAO, 0.05Pt/MAO, and 0.15Pt/Al₂O₃) under identical conditions (300 °C, WHSV=4.7 h⁻¹, feedstock: pure methylcyclohexane). As shown in **Figure S13**, 1Pt/MAO exhibited similarly stable dehydrogenation performance to 0.15Pt/MAO. In contrast, 0.05Pt/MAO showed a slight decline in activity, likely due to the migration of Pt single atoms onto nanoclusters, which altered cluster structure and reduced activity.

This suggests that MAO has limited ability to stabilize Pt single atoms, an issue common to many oxide supports. It should be noted, however, that single-atom Pt is not the most active site for these reactions, their performance is not the central focus of this study.

Figure S13. Long-term stability of 1Pt/MAO, 0.05Pt/MAO, and 0.15Pt/Al₂O₃ for methylcyclohexane dehydrogenation at 300 °C.

More importantly, direct comparison between 0.15Pt/MAO and 0.15Pt/Al₂O₃ with identical Pt loadings highlights the superior stability of MAO-supported systems. While 0.15Pt/MAO maintained nearly constant conversion over 100 h, 0.15Pt/Al₂O₃ experienced a rapid deactivation, with conversion dropping from ~100% to ~80% within only 15 h. This striking difference underscores the significant advantage of MAO in stabilizing catalytically active Pt species during long-term dehydrogenation.

Hydrogen purity was also assessed by on-line GC (Agilent GC8890) equipped with an FID detector. Methane was detected only at 30-40 ppm, corresponding to >99.99% selectivity toward the dehydrogenation pathway. These results demonstrate that the Pt/MAO catalysts combine excellent stability with high selectivity.

We have revised the main text to explicitly highlight these additional experiments (Line 278-284 on Page16) and updated Figure 2e and Figure S13 accordingly.

“The catalyst sustains >90% conversion and >99.99% selectivity throughout a 100-hour continuous operation with methane concentrations stably maintained at 30-40 ppm, well below the 100 ppm threshold specified for PEMFC applications³³. Similarly, 1Pt/MAO maintained stable performance, whereas 0.05Pt/MAO displayed a gradual decline in activity, plausibly due to the migration of Pt single atoms onto clusters, which altered cluster structures and reduced their activity (Figure S13). In sharp contrast, 0.15Pt/Al₂O₃ suffered rapid deactivation, with conversion dropping from ~100% to ~80% within 15 h.”

Figure 2. (e) Long-term stability of 0.15Pt/MAO for methylcyclohexane dehydrogenation at 300 °C.

Response to the comments and suggestions of the reviewers

We sincerely thank the reviewers for carefully reviewing our manuscript and their valuable comments, which certainly help improve our manuscript. We also appreciate the opportunity that editor has given us, to address the comments and revise the manuscript. The changes in the revised manuscript have been highlighted in yellow for your review. The point-by-point responses are presented below.

Reviewer #1 (Remarks to the Author):

The authors appropriately revised the manuscript based on the reviewers' comments. This paper is now acceptable as is.

Reply:

We sincerely thank the reviewer for their time, constructive feedback, and positive evaluation. We are grateful for the thoughtful comments that helped us improve the manuscript.

Reviewer #2 (Remarks to the Author):

This manuscript investigates Pt catalyzed dehydrogenation of cycloalkanes relevant to LOHCs. By systematically modulating Pt–Pt coordination numbers (CNs) on Pt/MgAl₂O₄ (MAO), the authors identify reactant dependent, volcano shaped activity trends and show how specific ensembles can be selected as optimal active sites. The work advances an electronic structure-based rationale for activity and selectivity and proposes a framework for reactant-specific site design. The demonstration of sustained activity and stability under conditions pertinent to industrial LOHC operation strengthens the practical significance. The authors have also responded to prior comments, improving the completeness of the study. Nonetheless, several elements still warrant deeper justification or additional analysis for the central claims to be fully convincing.

Reply:

We sincerely thank the reviewer for the constructive comments and thoughtful summary of our work. We greatly appreciate the recognition of the mechanistic insights, the development of structure-activity relationships via CN-controlled Pt ensembles, and the practical relevance of our findings to LOHC dehydrogenation. In response to the remaining concerns, we have made the following key revisions and additions to strengthen our claims:

- (1) Expanded analysis of ICOHP trends along the dehydrogenation pathway to more rigorously link coordination-dependent bonding characteristics with

catalytic activity.

- (2) Provided detailed sensitivity analyses for kinetic fitting, EXAFS-derived coordination structures, and TOF normalization, including discussions of error bars, model assumptions, and data robustness.
- (3) Included additional mechanistic discussions on catalyst stability and reaction pathways, supported by further experimental and computational evidence.

These revisions are now incorporated into the updated manuscript and supporting information. We hope the new data and discussions comprehensively address the reviewer's comments and further reinforce the central conclusions of our work.

1. In Figure S41, the authors compare C–C cleavage barriers on Pt(111) and Pt13/MAO but omit Pt1. Since Pt1 exhibits the highest d-band center and represents the most under-coordinated limit, it provides an essential boundary condition for the proposed CN-dependent volcano relationship. To strengthen the mechanistic argument, I recommend the following:

(i) Include Pt1 in the barrier analysis under matched coverage and structural constraints.

Reply:

We sincerely thank the reviewer for this insightful and constructive suggestion.

We fully agree that Pt₁, representing the most under-coordinated and electronically

unsaturated limit, plays a critical role in defining the lower boundary of the CN-dependent reactivity framework. In response, we have now performed DFT calculations of C-C bond cleavage barriers on Pt₁. As expected, Pt₁ exhibits the lowest C-C activation barriers among the series, consistent with its enhanced ability to activate adsorbates due to its electronic structure (Figure S46). This directly supports our proposed trend that lower CN leads to stronger adsorption and bond activation, and validates Pt₁ as the necessary coordination-boundary case in the volcano relationship.

Figure S46. Reaction pathways and free energy profiles for C-C bond cleavage of C₆H₈, C₇H₁₀, C₁₀H₁₄ on Pt(111), Pt₁₃/MAO and Pt₁/MAO.

(ii) Present the results for Pt1, Pt13/MAO, and Pt(111) in the form of a volcano-type plot. This would quantitatively substantiate the claim of a “reactant-specific optimum CN” and render the central framework more convincing.

Reply:

Furthermore, to better understand reactant-dependent selectivity, we introduced a simple descriptor: the difference between C-C and C-H bond activation barriers ($\Delta E = E_{C-C} - E_{C-H}$). For key partially dehydrogenated intermediates such as C₆H₈, C₇H₁₀, and C₁₀H₁₄, we find that this selectivity descriptor is positive on Pt₁₃/MAO and Pt(111), indicating favorable dehydrogenation over C-C cleavage. However, on Pt₁, this descriptor becomes negative, suggesting that C-C bond scission may become kinetically favorable, potentially leading to undesired side reactions. These results imply that Pt single atoms with extremely low coordination numbers not only exhibit limited intrinsic dehydrogenation activity, but also promote C-C bond cleavage, which may lead to coke formation and catalyst deactivation. This mechanistic insight provides a plausible explanation for the reduced stability of the 0.05Pt/MAO catalyst, where such under-coordinated Pt species likely contribute to side reactions and loss of activity. We greatly appreciate the reviewer’s comment, which has helped us deepen the mechanistic discussion of C-C bond activation and side reaction risk at under-coordinated sites, and significantly improved the completeness and clarity of the manuscript.

This discussion has been incorporated into the revised manuscript, and the supporting data are presented in Figure S46-47.

Figure S47. Selectivity descriptor ($\Delta E = E_{C-C} - E_{C-H}$) on Pt(111), Pt₁₃/MAO and Pt₁/MAO for partially dehydrogenated intermediates (C₆H₈, C₇H₁₀, C₁₀H₁₄).

“To examine possible side effects, we further calculated C-C cleavage barriers for partially dehydrogenated intermediates on Pt(111), Pt₁₃/MAO and Pt₁/MAO (Figure S46). While enhanced d-π* back-donation lowers the barriers on all sites, the reduction is most pronounced on Pt₁/MAO, where the C-C cleavage barriers become comparable to or even lower than those for C-H activation (Figure S47). This result suggests that Pt single atoms with extremely low coordination tend to promote side reactions such as C-C scission, which may lead to coke formation and catalyst deactivation.” (Lines 519-

526 on Page 30)

2. The current -ICOHP analysis is restricted to the final adsorption states of fully dehydrogenated products. However, since the rate-determining step (RDS) in cycloalkane dehydrogenation typically corresponds to the first C-H activation, the descriptor should also be examined at earlier points along the pathway. I therefore

suggest extending the -ICOHP (or an equivalent bond-order/interaction metric) to include key states before, at, and immediately after the RDS, particularly for partially dehydrogenated intermediates. Establishing how these interaction metrics evolve relative to computed energy barriers would provide a more rigorous mechanistic link between the descriptor and the observed kinetics, thereby strengthening the central framework of the study.

Reply:

According to the reviewer's suggestion, we extended the -ICOHP analysis to include the first C-H activation step ($C_6H_{12} \rightarrow C_6H_{11} + H$, $C_7H_{14} \rightarrow C_7H_{13} + H$, and $C_{10}H_{18} \rightarrow C_{10}H_{17} + H$) along the entire reaction pathway, covering the initial, transition, and final states. As shown in **Figure S52b**, the saturated cycloalkane reactants exhibit negligible Pt-C bonding ($-ICOHP \approx 0$), reflecting weak interaction with Pt surfaces. In contrast, on low-coordinated Pt₁/MAO sites, -ICOHP values exceed 1, indicating that such under-coordinated Pt atoms can still form appreciable adsorption interactions even with saturated hydrocarbons.

At the C-H activation transition state, -ICOHP increases, signifying the onset of Pt-C bond formation as the C-H bond is being cleaved (**Figure S52c**). After C-H bond scission, fully formed Pt-C bonds result in larger -ICOHP values (**Figure S52d**). Across all three stages, -ICOHP values systematically increase as the Pt coordination number decreases, confirming stronger Pt-adsorbate bonding arising from enhanced d- π^* back-donation. Furthermore, for a given catalyst model, the magnitude of -ICOHP follows

the trend decalin > methylcyclohexane > cyclohexane, consistent with the trend observed in the final product adsorption states, thereby reinforcing the reactant-dependent volcano-type activity relationship proposed in this study.

Figure S52. -ICOHP values of Pt-C bond for products adsorption (a), reactants adsorption (b), C-H activation (c), and intermediates adsorption (d) on various Pt catalyst models.

We further examined the -ICOHP values of the optimal active sites for both the activation and final adsorption states across the three reactions. In each case, the most active sites correspond to an intermediate -ICOHP range (highlighted in red), representing a balance between sufficient bond activation and moderate adsorption strength. Notably, the optimal -ICOHP range differs among the elementary steps (reactant activation, intermediate adsorption, and product adsorption), reflecting their

distinct Pt-C bonding requirements. Taking the C-H activation step as a reference, the optimal Pt-C interaction typically falls within the -ICOHP range of 1.4-2.0.

The reviewer's comment has greatly helped us refine our understanding of the optimal -ICOHP ranges. In response, we have added these results and the corresponding discussion in the revised manuscript.

“To further quantify this interaction, crystal orbital Hamilton populations (COHP) analysis of Pt-C bonds was performed for reactants adsorption, C-H activation, intermediates adsorption, and products adsorption (Figure S52). The results show that -ICOHP increases with decreasing Pt coordination, consistent with stronger covalent interactions from enhanced d- π^* back-donation. Optimal catalytic performance across the different reactions is achieved when -ICOHP falls within an intermediate range, representing a balance between sufficient stabilization for C-H bond activation and facile product desorption. Moreover, the optimal -ICOHP range varies among the elementary steps, reflecting their distinct Pt-C bonding requirements. Specifically, the most effective Pt-C interactions for the C-H activation and product adsorption steps are found within -ICOHP ranges of around 1.4-2.0 and 1.2-1.4, respectively.” (Lines 580-590 on Page 33)

3. The continuous activity decay observed for 0.05 wt% Pt/MAO is attributed to single-atom clustering. Given that clustering increases CN, the reported -ICOHP trends would predict adsorption/desorption shifting toward a more balanced regime.

Could clustering instead evolve toward a quasi-steady (or a stable steady) state or even regenerate a more active, clustered catalyst compared to the initial single-atom state?

To clarify this point, I recommend the following:

(i) Conducting time-on-stream experiments combined with intermittent characterization (e.g., HAADF-STEM, EXAFS, CO-DRIFTS) to directly monitor the changes in CN and site population.

(ii) Testing a pre-clustered control sample to determine whether its performance stabilizes or surpasses that of the single-atom starting material.

In addition, the authors claims that MAO resists sintering, but it remains unclear why such stabilization would not apply to the 0.05 wt% sample. Please elaborate on this discrepancy. Finally, discuss whether sintering should indeed be considered a dominant deactivation pathway under LOHC dehydrogenation conditions, which generally occur at temperatures lower than those of classical sintering regimes.

Reply:

We thank the reviewer for this insightful comment. To clarify the deactivation mechanism of the 0.05Pt/MAO catalyst, we conducted additional experiments and revised our discussion accordingly. We performed an accelerated deactivation experiment under a high WHSV of 7 h⁻¹ (Figure S16). The catalytic activity of 0.05Pt/MAO dropped from 59.8% to 52.3% over 5 hours. After a 2-hour H₂ purge at reaction temperature, the activity recovered to 59.4%. This cycle was repeated and yielded consistent regeneration, indicating that the observed deactivation is largely

reversible. This reversibility implies that the primary deactivation mechanism is not irreversible Pt sintering, which is typically not recoverable by simple H₂ treatment. Instead, reversible surface poisoning, likely due to strong adsorption of aromatic products such as toluene and mild coke formation, appears to play the primary role.

Figure S16. Long-term stability of 1Pt/MAO and 0.05Pt/MAO for methylcyclohexane dehydrogenation at 300 °C. Activity loss on 0.05Pt/MAO is reversible upon H₂ purging.

This was further supported by in-situ FTIR, which showed characteristic C=C stretching bands of toluene (1456 and 1608 cm⁻¹) increasing during MCH exposure and gradually decreasing upon H₂ purging (Figures S17a-b). These results confirm that low-coordinated Pt sites in 0.05Pt/MAO are susceptible to product-induced poisoning, but can be regenerated via hydrogen-assisted desorption or hydrogenation (*ACS Catal.*, 2022, 12, 7248-7261). This observation aligns with our mechanistic framework, in which lower CN_{Pt} enhances adsorption of reaction intermediates. Given its high proportion of single atoms (63%), 0.05Pt/MAO offers abundant undercoordinated sites, rendering it prone to reversible deactivation. In contrast, 0.15Pt/MAO and 1Pt/MAO

are dominated by high-coordinated clusters and nanoparticles, exhibiting moderate to weak adsorption strength and thus superior stability under reaction conditions.

Figure S17. In-situ FTIR spectra of 0.05Pt/MAO catalyst during (a) methylcyclohexane dehydrogenation and (b) subsequent H₂ purging. (c) CO-DRIFTS spectra of 0.05Pt/MAO before and after long-term testing.

We further conducted CO-DRIFTS measurements on the 0.05Pt/MAO catalyst before and after long-term testing (Figures S17c). The CO adsorption peak near 2064 cm⁻¹ showed only slight shifts, suggesting minor electronic or structural changes. While limited Pt migration or aggregation cannot be completely excluded, the observed deactivation is largely reversible, and both catalytic and spectroscopic data strongly support that sintering is not the dominant deactivation pathway under our reaction conditions. As the reviewer rightly pointed out, classical sintering typically occurs at

much higher temperatures than those used for LOHC dehydrogenation.

We have carefully clarified and incorporated these points into the revised manuscript as suggested.

“However, 0.05Pt/MAO showed gradual but reversible deactivation, as H₂ purging restored its initial activity (Figure S16), excluding irreversible sintering as the main cause. Instead, deactivation is mainly attributed to surface poisoning caused by strong adsorption of products and slight coke formation on low-coordinated Pt sites, which can be removed by hydrogen-assisted desorption or hydrogenation⁴⁷. In-situ FTIR confirmed this mechanism, showing gradual strengthen of toluene-related C=C stretching bands (1456 and 1608 cm⁻¹) during reaction and their disappearance upon H₂ treatment (Figure S17a-b). Additionally, CO-DRIFTS before and after stability testing showed a slight shift in the CO adsorption peak (Figure S17c), suggesting that partial migration of Pt single atoms to clusters or subsurface sites may also contribute to the observed deactivation.” (Lines 303-313 on Page 17)

“Furthermore, the higher reaction order of hydrogen for 0.05Pt/MAO compared to 3Pt/MAO indicates stronger adsorption of dehydrogenated intermediates on low-coordinated Pt. This behavior arises because low-coordinated Pt atoms stabilize dehydrogenation-derived intermediates more strongly, amplifying the thermodynamic driving force for reverse hydrogenation⁴⁷. This mechanistic understanding aligns with our earlier stability tests, where 0.05Pt/MAO showed reversible deactivation due to strong product adsorption and was fully regenerated upon H₂ purging, which is not

observed for 1Pt/MAO (Figure S16).” (Lines 401-404 on Page 23)

“In-situ FTIR measurements were conducted to monitor toluene adsorption behavior during methylcyclohexane dehydrogenation. The catalyst pre-reduced following the same procedure described above was purged with N₂ at 300 °C for 20 min to remove residual hydrogen species. A background spectrum was first collected before introducing methylcyclohexane into the N₂ carrier gas. Spectra were then recorded at 3 min intervals for 60 min. Subsequently, the reaction gas was switched to H₂ for purging over 60 min, during which spectra were continuously collected at 3 min intervals.” (Lines 250-257 on Page 11-12 of SI)

“For long-term stability evaluations, pure methylcyclohexane (211 μL min⁻¹) was fed over 1.4 g of 0.05Pt/MAO, 2 g of 0.15Pt/MAO, 2 g of 1Pt/MAO, and 2 g of 0.15Pt/Al₂O₃ catalysts.” (Lines 288-291 on Page 13 of SI)

4. The improved R-factor is noted and appreciated, but the reported Pt–O bond length of ~1.75 Å is unusually short compared to the commonly observed 1.9–2.1 Å range for Pt–O coordination. To assess the robustness of this result, the authors should provide (i) the exact fitting model (scatterers, multiple-scattering paths), k- and R-ranges, S₀² and ΔE₀ handling, constraints/tie-lines across shells, and amplitude reduction factors; (ii) confidence intervals, parameter correlations, and residuals; and (iii) tests of alternative physically plausible models (e.g., Pt–O–Al bridges, mixed Pt–O/Pt–Cl or Pt–C environments if relevant).

A concise methodological appendix or an SI table compiling these details would clarify whether the short Pt–O distance originates from model selection, path interference, or reflects a genuine chemical feature. Such clarification is critical for interpreting the local structure with confidence.

Reply:

We thank the reviewer for raising this important point regarding the unusually short Pt-O bond length (~ 1.75 Å) reported in our original EXAFS fitting. Following the reviewer’s suggestions, we have significantly revised our EXAFS analysis to improve both physical plausibility and transparency. First, we have compiled all fitting details into **Table S7**, including: the fitting scatterers, scattering paths (Single Scattering), k- and R-ranges used in the fitting, amplitude reduction factors (S_0^2), energy shift parameters (ΔE_0), applied constraints or tie-lines across coordination shells (none), confidence intervals (\pm values) and residuals (R factor) for each dataset.

Table S7. EXAFS fitting parameters at the Pt L3-edge for various Pt samples

$(S_0^2=0.90)$.

Sample	Shell^a	CN^b	R(Å)^c	σ^2 (Å²)^d	ΔE_0(eV)^e	R factor
Pt foil	Pt-Pt	12	2.76±0.01	0.003±0.001	8.48±0.31	0.001
0.02Pt/MAO	Pt-O	4.07±1.0	1.93±0.04	0.009±0.007	9.26±3.43	0.036
0.05Pt/MAO	Pt-O	3.75±0.8	1.86±0.03	0.004±0.001	7.14±3.9	0.009

	Pt-Pt	0.93±0.2	2.58±0.03	0.004±0.001		
	Pt-O	1.90±0.26	1.97±0.02	0.001±0.001		
0.15Pt/MAO					7.88±2.25	0.041
	Pt-Pt	5.30±0.58	2.71±0.05	0.003±0.001		

^a The EXAFS fitting employed single-scattering paths for both the Pt-O and Pt-Pt contributions, derived from PtO₂ and Pt foil reference structures, respectively.

^bCN: coordination number; ^cR: distance between absorber and backscatter atoms; ^dσ²: Debye-Waller factor to account for both thermal and structural disorders; ^eΔE₀: inner potential correction.

R factor indicates the goodness of the fit. Fitting range: 3 < k (Å⁻¹) < 11 and 1.0 < R (Å) < 3.6. No inter-shell constraints or tie-lines were applied.

Upon closer inspection, we found that the previously reported short Pt-O bond length likely resulted from excessive fitting freedom aimed at minimizing the R factor. To evaluate model robustness, we further tested alternative fitting configurations, including Pt-C and Pt-Cl coordination environments, as suggested by the reviewer (Table R1). However, these alternative models produced either higher R factors and unphysical σ² values (Pt-C) or unrealistic negative ΔE₀ shifts (Pt-Cl), indicating poor physical consistency. Even when the k range was reduced to improve signal-to-noise, these models failed to yield stable or meaningful fits.

Therefore, the Pt-O coordination model based on PtO₂ references was retained as the most physically reliable representation of the local structure in 0.02Pt/MAO. The

updated fitting gives a Pt-O bond length of 1.93 ± 0.04 Å, fully consistent with the typical 1.9-2.0 Å range reported for Pt-O bonding in oxidized Pt species. All updated results and discussions are included in the revised supporting information, which together clarify the origin of the previous discrepancy and enhance confidence in the structural interpretation.

Table R1. Comparison of alternative EXAFS fitting models for 0.02Pt/MAO at the Pt L3-edge ($S_0^2=0.90$).

Model	Shell	CN	R (Å)	σ^2 (Å²)	ΔE_0 (eV)	R factor	k (Å⁻¹)
Pt-O	Pt-O	4.07±1.0	1.93±0.04	0.009±0.007	9.26±3.43	0.036	3-11
Pt-C	Pt-C	4.07±2.4	2.00±0.04	0.018±0.007	6.55±4.12	0.010	3-11
Pt-C	Pt-C	4.07±1.5	2.01±0.08	0.007±0.017	8.59±8.47	0.050	3-7
Pt-Cl	Pt-Cl	4.07±4.5	2.26±0.03	0.029±0.006	-22.46±3.99	0.042	3-11

5. Several of the central trends depend on site fractions of single atoms, clusters, nanoparticle edges and their intrinsic TOFs through simultaneous fitting, under assumptions such as fixed reaction orders across compositions and size-independent edge TOFs. To ensure robustness of these conclusions, the authors should provide (i) parameter identifiability diagnostics (profile likelihoods or Fisher information), (ii) uncertainty propagation to site-specific TOFs/volcano positions (bootstrap or Bayesian intervals), and (iii) sensitivity to the assumed reaction orders and particle-shape model

(see Point 6). Reporting CIs on optimal CN and volcano peak will strengthen the claims.

Reply:

We appreciate the reviewer's insightful comment regarding parameter identifiability and robustness. As our primary objective in the kinetic fitting is to determine the relative fractions of Pt single atoms (f_s), clusters (f_c), and nanoparticles (f_n) across different catalysts, we focused on assessing the identifiability of these parameters. We conducted profile likelihood analysis for f_s , f_c , and f_n in both the 0.05Pt and 0.15Pt/MAO samples. The profile likelihood function was defined as:

$$PL(f) = \exp\left(-\frac{1}{2} \times \frac{RSS(f) - RSS_{min}}{\frac{RSS_{min}}{n - p}}\right)$$

where $RSS(f)$ is the residual sum of squares when parameter f is fixed, RSS_{min} is the minimum residual sum of squares obtained from the global optimization, n is the number of data, and p is the number of fitted parameters. This expression provides a normalized likelihood curve (maximum = 1), which enables us to estimate confidence intervals for each parameter by locating the range over which $PL(f) > \text{threshold}$ (0.15 for 95% CI).

Figure S9. Normalized likelihood curves for fitted site fractions of Pt single atom (f_s), Pt cluster (f_c), and Pt nanoparticle (f_n) in 0.05Pt/MAO (a-b) and 0.15Pt/MAO (c-e) catalysts. Gray shaded regions indicate 95% confidence intervals based on the likelihood threshold criterion.

“To assess the identifiability of the fitted parameters, profile likelihood analysis was performed using the following definition:

$$PL(f) = \exp\left(-\frac{1}{2} \times \frac{RSS(f) - RSS_{min}}{RSS_{min}}\right) \quad (16)$$

where $RSS(f)$ is the residual sum of squares when parameter f_s , f_c or f_n is fixed, RSS_{min} is the minimum residual sum of squares obtained from the global optimization, n is the number of data, and p is the number of fitted parameters. This expression provides a normalized likelihood curve (maximum = 1), which enables us to estimate confidence intervals for each parameter by locating the range over which $PL(f) >$ threshold (0.15 for 95% CI).” (Lines 402-410 on Page 19 of SI)

The resulting PL curves (Figure S9) indicate sharp maxima for all three parameters, confirming good identifiability. Based on the 95% confidence intervals derived from PL curves, we then propagated the parameter uncertainty to calculate confidence bands for key observables such as site-specific TOFs and CN_{Pt-Pt} . These confidence intervals are now explicitly presented as horizontal or vertical error bars in revised Figure 2d. Interestingly, we find that while the uncertainty in parameters has relatively minor influence on the calculated TOFs, it has more pronounced impact on the computed CN values. Nevertheless, the reactant-specific volcano trends remain robust, and our updated volcano plots now explicitly present the optimal CN ranges instead of fixed value estimates, which improves the scientific rigor of the trend analysis. To better reflect parameter uncertainty and enhance clarity, the optimal CN values are now reported using “~” rather than single-point estimates.

Figure 2d. Site-specific TOF as a function of CN_{Pt-Pt} for different dehydrogenation reactions at 280 °C.

“To account for uncertainty, we propagated the 95% confidence intervals obtained from profile likelihood analysis (Figures S9) to the derived TOF and CN_{Pt-Pt} . The

resulting error bars are shown in Figure 2d. While the uncertainty in TOF is minimal, CN values exhibit broader variation. Importantly, the overall volcano-shaped trends and optimal CN ranges remain robust.”

“Using a tunable Pt/MgAl₂O₄ system, we identify optimal CN_{Pt-Pt} values of ~2.5, ~4.7, and ~7.0 for cyclohexane, methylcyclohexane, and decalin, respectively.”

“Figure 2d summarizes the site-specific TOF as a function of the CN_{Pt-Pt} for the dehydrogenation of cyclohexane, methylcyclohexane, and decalin, where the optimal CN_{Pt-Pt} values for the three reactions were identified as ~2.5, ~4.7, and ~7.0, respectively.”

“By engineering a coordination-tunable Pt/MgAl₂O₄ system and employing cyclohexane, methylcyclohexane, and decalin as model substrates, we identify volcano-type activity trends with optimal CN_{Pt-Pt} values of ~2.5, ~4.7, and ~7.0, respectively.”

Regarding the assumption of fixed reaction orders, we note that the reaction orders used in the fitting were experimentally derived from both low-loading (0.05Pt/MAO) and high-loading (3Pt/MAO) catalysts, and their values differ only slightly (0.41-0.56, 0.42-0.47, 2.03-2.06 for cyclohexane, methylcyclohexane, and decalin, respectively). To further assess the sensitivity of our fitting results to these assumptions, we varied the reaction orders by $\pm 50\%$ and recalculated the corresponding entropy contributions and volcano curves. The results show that the impact of these variations on the calculated activation entropy is minimal, with absolute deviations of only 0.69-0.94, 0.47-0.62, and 5.8-5.9 J mol⁻¹ K⁻¹ for cyclohexane, methylcyclohexane, and decalin,

respectively. This analysis is now included and discussed in the revised supporting information with updated Figure S24.

Figure S24. Apparent activation energy (square) and entropy (circle) of Pt/MAO catalysts for the dehydrogenation of cyclohexane, methylcyclohexane and decalin.

“Based on the consistent reaction orders observed in Figure 3c-e between low (0.05Pt/MAO) and high-coordinated Pt (3Pt/MAO), we approximated the orders for intermediate catalysts as follows: for 0.02-0.15Pt/MAO, we used values similar to 0.05Pt/MAO, while for 0.375-3Pt/MAO, we adopted values comparable to 3Pt/MAO. Notably, the measured reaction orders of reactant differ only slightly across these catalysts (0.41 to 0.56 for cyclohexane, 0.42 to 0.4 for methylcyclohexane, and 2.03 to 2.06 for decalin).” (Lines 418-420 on Page 20 of SI)

“To evaluate the robustness of our analysis, we varied the reaction orders by $\pm 50\%$ and recalculated the activation entropy and volcano plots. The resulting changes in $\Delta S_{app}^{\ddagger}$ were minor and had negligible influence on the overall volcano shape or the optimal

CN values (Figure S24), confirming that the fitted trends are insensitive to small deviations in reaction order.” (Lines 429-433 on Page 20 of SI)

We address the sensitivity to the assumed particle-shape model in our response to Comment 6 below.

6. Because site counts directly determine the site-normalized TOFs, the choice of particle-shape model is critical. At present, the main text and SI appear to apply different assumptions (e.g., truncated octahedron vs. cuboctahedron), which may compromise consistency. The author should (i) standardize the particle-shape model (ensure a single geometry is used throughout the manuscript), and explicitly justify the choice (e.g., supported by TEM) as well as (ii) provide a sensitivity analysis by repeating the site-counting procedure with alternative plausible geometries, e.g., truncated octahedron, cuboctahedron, decahedron).

Reply:

We apologize for the inconsistency in terminology. We confirm that a truncated octahedron model was consistently used throughout the manuscript for Pt nanoparticle site counting. This geometry was chosen because (1) it is widely adopted in literature to represent equilibrium shapes of Pt nanoparticles, and (2) it aligns with our HAADF-STEM observations (Figure S6), where most Pt nanoparticles exhibit truncated or semi-truncated octahedral morphologies with well-defined (111) and (100) facets. We have

now corrected all mentions of the Pt nanoparticle shape model in the main text and supporting information to reflect this.

“Figure S7. (a) Number of surface atoms per gram of Pt as a function of Pt particle size based on a truncated octahedral model.”

“In the 0.375Pt/MAO, 1Pt/MAO, and 3Pt/MAO catalysts, Pt was predominantly present as nanoparticles, enabling site-specific TOF analysis using the truncated octahedral model, which is commonly used to represent Pt nanoparticle geometry^{41, 42} and consistent with our HAADF-STEM observations (Figure S6).” (Lines 198-200 on Page 12)

Figure S6. HAADF-STEM images of Pt nanoparticles in 1Pt/MAO catalysts with schematic truncated octahedral models overlaid.

To assess the sensitivity of our results to the assumed particle shape, we have followed the reviewer’s suggestion and repeated the site-counting and TOF normalization using a cuboctahedron model. As shown in Figure S13a, this alternative model still supports the conclusion that edge Pt atoms are the dominant active sites on

nanoparticles. However, due to the lower fraction of edge atoms in the cuboctahedron, the resulting edge TOFs are artificially higher. This shifts the volcano plots (particularly for methylcyclohexane dehydrogenation), making higher CN sites appear more favorable, thus flattening or distorting the volcano shapes compared to those obtained with the truncated octahedron. This highlights the significant influence of the assumed geometry on quantitative trends.

Furthermore, this shape model change can slightly alter the kinetic fitting results. The estimated distribution of Pt single atom (f_a), cluster (f_c) and nanoparticle (f_n) in the 0.15Pt/MAO catalyst changes from 0/0.54/0.46 to 0.04/0.63/0.33, and the corresponding cluster CN increases from ~ 4.7 to ~ 5.2 . All updated results are now included in the revised version, and a detailed discussion of shape sensitivity has been added.

Figure S13. Site-specific TOF of Pt/MAO catalysts estimated using the cuboctahedral particle model: (a) TOF across different Pt loadings; (b) TOF as a function of CN_{Pt-Pt} for different dehydrogenation reactions at 280 °C.

“To assess nanoparticle model sensitivity, we repeated the analysis using a cuboctahedron geometry. Although this still highlights edge Pt as the dominant sites, the lower edge fraction leads to higher apparent TOF of edge Pt and flatter volcano shapes, especially for methylcyclohexane dehydrogenation (Figure S13). This highlights the importance of nanoparticle model assumptions in quantitative analysis. The shape change also affects site distributions: for 0.15Pt/MAO, the fractions of Pt single atom, cluster and nanoparticle shift from 0/0.54/0.46 to 0.04/0.63/0.33, with the cluster CN_{Pt-Pt} increasing from ~ 4.7 to ~ 5.2 . Regardless of the specific model, both geometries represent idealized shapes, and real Pt nanoparticles may deviate from these forms, potentially exposing additional low-coordination or defect sites. Complementary techniques such as CO-DRIFTS⁴⁵, TPD⁴⁶, and EXAFS will be valuable in future studies to validate facet distributions and refine structural models.” (Lines 266-276 on Page 15-16)

7. The author need to clarify whether the RDS barriers correlate most strongly with the product LUMO, the reactant LUMO, or $\Delta LUMO$ along the reaction path. Including a short comparison in the SI would be helpful.

Reply:

We thank the reviewer for this valuable suggestion. In response, we have computed the LUMO energies of the reactants (Figure S49) and analyzed their correlation with the Pt d-band centers at the optimal active sites for cyclohexane,

methylcyclohexane, and decalin dehydrogenation. As shown in Figure S51a, the LUMO of the saturated reactants is consistently high (less occupied) and shows minimal variation among different molecules. In contrast, the LUMO energies of the dehydrogenated intermediates and especially the final products exhibit a more pronounced decrease and greater variation, indicating stronger π^* orbital interactions as dehydrogenation proceeds. Consequently, the product LUMO shows the strongest correlation with the optimal Pt d-band center, suggesting the dominant role of $d-\pi^*$ orbital interactions in governing adsorption strength and thus activity.

Figure S49. LUMO energy comparison of reactants and partially dehydrogenated intermediates.

Figure S51. (a) Correlation of the optimal Pt d-band center with the LUMO energy of reactants, partially dehydrogenated intermediates, and products. (b) Correlation of the optimal Pt d-band center with the Δ LUMO values along different reaction steps.

To further validate this, we followed the reviewer's suggestion and analyzed correlations between the d-band center and various Δ LUMO values along the reaction path (reactant-intermediate, intermediate-product, and reactant-product) (Figure S51b). Among these, Δ LUMO (reactant-product) and Δ LUMO (intermediate-product) show reasonable correlations, but this is largely driven by differences in the product LUMO, since the reactant and intermediate LUMOs are quite similar. From a quantitative perspective, the correlation between the optimal d-band center and the product LUMO yields the lowest relative error (13.2%), slightly outperforming Δ LUMO-based correlations (15.3% and 15.2%, respectively). These results further confirm that the product LUMO is the most sensitive electronic descriptor for interpreting the observed reactant dependent coordination-activity relationship.

We have incorporated these results into the revised manuscript and added the corresponding discussion in the main text.

“Additionally, the correlations of LUMO energies for reactants and intermediates with optimal Pt d-band center were analyzed (Figure S51a). While reactants show similarly high LUMO levels, the dehydrogenated species, particularly the products exhibit significantly lower and more variable LUMO energies, indicating stronger π^* orbital interactions as dehydrogenation proceeds. Although Δ LUMO values were also analyzed (Figure S51b), the trends are predominantly governed by product LUMOs. Thus, product LUMO correlates most strongly with the optimal Pt d-band center, highlighting the dominant role of d- π^* orbital interactions in governing adsorption strength and thus activity.” (Lines 571-579 on Page 32-33)

8. For clarity, the author need to specify explicitly whether the volcano positions are referenced to the Pt-Pt CN of the active atom(s) or to the average particle CN. Consistent notation across the manuscript would help avoid ambiguity.

Reply:

We thank the reviewer for pointing out this ambiguity. Throughout the manuscript, all CN values refer specifically to the Pt-Pt coordination number (CN_{Pt-Pt}) of the active sites, rather than the average particle CN. We have now clearly defined this usage at the first mention of CN_{Pt-Pt} and revised the manuscript to ensure consistent terminology throughout. In cases where cited literature discusses average coordination numbers, we have explicitly labeled these as “average CN_{Pt-Pt} ” to avoid confusion. These clarifications have been uniformly implemented to eliminate ambiguity.

“In this study, we strategically engineered a tunable Pt/MgAl₂O₄ (Pt/MAO) system spanning single atoms, clusters, and nanoparticles, creating a coordination gradient (CN_{Pt-Pt}, defined as the Pt-Pt coordination number of catalytically active sites) to decode the catalytic fingerprint of dehydrogenation.” (Line 94 on Page 6)

“For example, Ma et al. reported that fully exposed Pt ensembles with an average CN_{Pt-Pt} of around 2 achieved optimal catalytic performance for cyclohexane dehydrogenation²³.” (Line 252 on Page 15)

“Nevertheless, Pt/Al₂O₃ catalysts still exhibited reactant-dependent volcano-type trends: decalin dehydrogenation peaked at 0.15Pt/Al₂O₃ (nanoparticle-dominated), methylcyclohexane at 0.05PtPt/Al₂O₃, closely resembling the 0.15Pt/MAO, while cyclohexane reached its optimum between 0.02 and 0.05Pt/Al₂O₃, aligning with the ~2.5 CN_{Pt-Pt} optimum identified on MAO (Figure S15b).” (Lines 288-289 on Page 16)

“To validate the mechanistic changes associated with CN_{Pt-Pt}, 0.05Pt/MAO and 3Pt/MAO were selected as representative catalysts for low- and high- coordinated Pt, respectively, and their reaction orders were measured (Figure S25-27).” (Line 382 on Page 22)

9. Add a schematic showing which structural motif(s) (SA, small cluster edge, NP step) dominate in each loading.

Reply:

We thank the reviewer for the helpful suggestion. A schematic (Figure 2c) illustrating the dominant Pt structural motifs across different loadings was already included in the original manuscript. We have now further optimized this figure and the corresponding caption to more clearly depict the structural evolution from isolated single atoms to clusters and eventually to nanoparticles as the Pt loading increases. The corresponding structural fractions are quantitatively derived and discussed in detail in the revised manuscript.

Figure 2c. Schematic illustration of the dominant Pt structural motifs across different Pt loadings, as determined from global kinetic fitting.

“Collectively, data fitting identified four representative active-site ensembles: atomically dispersed Pt ($CN_{Pt-Pt}=0$), small clusters ($CN_{Pt-Pt}\approx 2.5$), medium clusters ($CN_{Pt-Pt}\approx 4.7$), and nanoparticles (CN_{Pt-Pt} of edge Pt atom=7) (Figure 2c).” (Line 244 on Page 14)

Reviewer #3 (Remarks to the Author):

The authors have thoroughly addressed most of reviewer comments and provided substantial revisions throughout the manuscript. The inclusion of new experiments, expanded analyses, and clarifications substantially strengthen the work, making the study more robust and compelling. Relative to my previous assessment, the manuscript now represents a significant improvement.

Previously, my review was divided into two main categories:

1) Catalyst Characterization, with subsections:

1-1) Support Effect, 1-2) Inhomogeneity of Pt Sites, 1-3) CN vs. Facet Distribution

2) Performance Evaluation, with subsections:

2-1) LOHC Selection Rationale, 2-2) Pt Speciation and TOF Calculation, 2-3)

Long-Term Stability Testing

Reply:

We sincerely thank the reviewer for the thoughtful feedback and constructive suggestions provided during the review process. We greatly appreciate your recognition of the improvements made in our revised manuscript. Your comments have significantly enhanced the scientific rigor and clarity of the work.

Several points, however, still require further attention before publication:

(1) Support Effects and Volcano Relationship (Comment 1-1):

The comparative study with Pt/Al₂O₃ convincingly demonstrates that MgAl₂O₄ stabilizes low-CN Pt species to a greater extent, supporting claims of enhanced dispersion and stability. However, the assertion of "support-independent" behavior may not be fully validated, as the low-CN regime cannot be accessed on Al₂O₃. Phrasing such as "largely support-independent" or "consistent across supports within the experimentally accessible CN range" would be more precise. Additionally, given the pronounced differences in thermal stability, basicity, and metal-support interaction between MgAl₂O₄ and Al₂O₃, these factors should be acknowledged as potentially influencing both Pt properties and the observed volcano-type reactivity trends. It would be valuable to note that further comparison with additional supports could reinforce the conclusion, though this is not essential for the current study.

Reply:

We thank the reviewer for this careful and constructive comment. We agree that while MgAl₂O₄ enables access to a broader range of Pt coordination environments, particularly in the low-CN regime, the comparative data with Al₂O₃ do not fully justify a support-independent conclusion. In response, we have revised the manuscript to adopt more precise phrasing.

Additionally, we now explicitly acknowledge that support-specific properties such as thermal stability, surface basicity, and metal-support interaction strength may influence both Pt dispersion and the observed activity trends. This has been added to

the revised discussion. We also note that exploring a broader range of supports in future studies would be valuable to further assess the generality of the observed structure-activity relationship.

“These results indicate that the reactant-dependent volcano relationship between CN_{Pt-Pt} and activity is largely support-independent, but the ability of MAO to stabilize low-coordinated Pt clusters makes it uniquely suitable for systematically probing this relationship. Moreover, the pronounced differences in thermal stability, basicity, and metal-support interactions between MAO and Al_2O_3 may also influence Pt speciation and reactivity, and comparisons with additional supports in future work could help generalize these findings.” (Lines 289-296 on Page 16-17)

(2) *CN vs. Facet Distribution (Comment 1-3):*

The authors now clearly describe the methodology for quantifying facet distributions using a truncated octahedron model and site-specific TOF normalization (Figure S6). This effectively identifies edge Pt atoms as probable primary active sites, while average CN is an appropriate descriptor in the single-atom and small-cluster regime. The supporting information addresses earlier concerns regarding preparation, calculation details, and figure interpretation. Including a brief acknowledgment that the truncated octahedron model is an idealization—and that real nanoparticles may expose additional sites—would further strengthen the discussion. Since the numerical facet distribution is central to correlating CN with TOF, the authors should also

mention complementary experimental methods, such as temperature-programmed reduction or desorption, for future quantitative facet analysis.

Reply:

We thank the reviewer for these thoughtful suggestions. In the revised manuscript, we now explicitly acknowledge that the truncated octahedron model represents an idealized geometry and that real Pt nanoparticles may deviate from this shape, potentially exposing additional low-coordination or defect-related sites. We also included a parallel analysis using a cuboctahedron geometry to assess the sensitivity of the model assumptions on TOF calculations, with the results and discussion now provided in **Figure S13** and the main text. Nonetheless, we adopted the truncated octahedron model as the primary framework because (1) it is widely used in the literature to represent the equilibrium shape of Pt nanoparticles, and (2) it aligns well with our HAADF-STEM observations (**Figure S6**), which show faceted particles consistent with this geometry.

We have also added a brief discussion on complementary experimental techniques, such as CO-DRIFTS, TPD and EXAFS, which can be employed in future studies to provide direct insight into facet distributions and validate model assumptions.

“To assess nanoparticle model sensitivity, we repeated the analysis using a cuboctahedron geometry. Although edge Pt remains the dominant active site, the lower edge fraction leads to higher apparent TOF of edge Pt and flatter volcano shapes, especially for methylcyclohexane dehydrogenation (**Figure S13**). This highlights the

importance of nanoparticle model assumptions in quantitative analysis. The shape change also affects site distributions: for 0.15Pt/MAO, the fractions of Pt single atom, cluster and nanoparticle shift from 0/0.54/0.46 to 0.04/0.63/0.33, with the cluster CN_{Pt} increasing from ~ 4.7 to ~ 5.2 . Regardless of the specific model, both geometries represent idealized shapes, and real Pt nanoparticles may deviate from these forms, potentially exposing additional low-coordination or defect sites. Complementary techniques such as CO-DRIFTS⁴⁵, TPD⁴⁶, and EXAFS will be valuable in future studies to validate facet distributions and refine structural models.” (Lines 266-276 on Page

15-16)

(3) Single-Atom Migration (Comment 2-3):

The authors attribute slight deactivation of 0.05Pt/MAO to migration of Pt single atoms onto clusters. While this is plausible, the mechanistic discussion would be improved by either briefly explaining how this migration was inferred from experimental results, or citing studies that have observed similar behavior for Pt on oxide supports. Such clarification would aid readers unfamiliar with single-atom instability and reinforce the proposed mechanism of deactivation at low metal loading. Notably, Pt single atoms abundant in 0.05Pt/MAO may penetrate into the MgAl₂O₄ support via strong metal-support interaction, leading to minimal catalytic activity and limiting the applicability of the proposed CN–TOF relationship.

Reply:

We thank the reviewer for this insightful comment. We agree that the slight deactivation observed for 0.05Pt/MAO may stem from multiple mechanisms related to single-atom instability. In the revised manuscript, we have incorporated new experimental evidence to clarify the deactivation mechanism. We conducted an accelerated deactivation test under a high WHSV of 7 h⁻¹. Within 5 hours, the catalytic activity dropped from 59.8% to 52.3%. Following a 2-hour H₂ purge at reaction temperature, the activity recovered to 59.4% (Figures S16). Repeating this cycle yielded consistent results, suggesting that the deactivation is largely reversible.

Figure S16. Long-term stability of 1Pt/MAO and 0.05Pt/MAO for methylcyclohexane dehydrogenation at 300 °C. Activity loss on 0.05Pt/MAO is reversible upon H₂ purging.

This reversibility implies that the primary deactivation mechanism is not irreversible Pt sintering, which is typically not recoverable by simple H₂ treatment. Instead, it indicates reversible surface poisoning, likely due to strong adsorption of aromatic products such as toluene and mild coke formation. This was further supported by in-situ FTIR, which showed characteristic C=C stretching bands of toluene (1456

and 1608 cm^{-1}) increasing during MCH exposure and gradually decreasing upon H_2 purging (Figures S17a-b). This behavior confirms that the low-coordinated Pt sites on 0.05Pt/MAO are prone to product-induced poisoning, but can be regenerated via hydrogen-assisted desorption or hydrogenation. These findings are consistent with our mechanistic proposal that lower $\text{CN}_{\text{Pt-Pt}}$ shows enhanced adsorption for reaction species.

Figure S17. In-situ FTIR spectra of 0.05Pt/MAO catalyst during (a) methylcyclohexane dehydrogenation and (b) subsequent H_2 purging. (c) CO-DRIFTS spectra of 0.05Pt/MAO before and after long-term testing.

We further conducted CO-DRIFTS measurements on the 0.05Pt/MAO catalyst before and after long-term testing (Figures S17c). The CO adsorption peak near 2064 cm^{-1} showed only slight shifts, suggesting minor electronic or structural changes. While we cannot rule out the potential contributions of atom migration or embedding

into the MgAl₂O₄ lattice, our combined catalytic and spectroscopic results strongly support that product poisoning is the dominant deactivation pathway under our conditions. We have clarified and discussed this in the revised text as suggested.

“However, 0.05Pt/MAO showed gradual but reversible deactivation, as H₂ purging restored its initial activity (Figure S16), excluding irreversible sintering as the main cause. Instead, deactivation is mainly attributed to surface poisoning caused by strong adsorption of products and slight coke formation on low-coordinated Pt sites, which can be removed by hydrogen-assisted desorption or hydrogenation⁴⁷. In-situ FTIR confirmed this mechanism, showing gradual strengthen of toluene-related C=C stretching bands (1456 and 1608 cm⁻¹) during reaction and their disappearance upon H₂ treatment (Figure S17a-b). Additionally, CO-DRIFTS before and after stability testing showed a slight shift in the CO adsorption peak (Figure S17c), suggesting that partial migration of Pt single atoms to clusters or subsurface sites may also contribute to the observed deactivation.” (Lines 303-313 on Page 17)

“In-situ FTIR measurements were conducted to monitor toluene adsorption behavior during methylcyclohexane dehydrogenation. The catalyst pre-reduced following the same procedure described above was purged with N₂ at 300 °C for 20 min to remove residual hydrogen species. A background spectrum was first collected before introducing methylcyclohexane into the N₂ carrier gas. Spectra were then recorded at 3 min intervals for 60 min. Subsequently, the reaction gas was switched to H₂ for purging over 60 min, during which spectra were continuously collected at 3 min intervals.” (Lines 250-257 on Page 11-12 of SI)

“For long-term stability evaluations, pure methylcyclohexane ($211 \mu\text{L min}^{-1}$) was fed over 1.4 g of 0.05Pt/MAO, 2 g of 0.15Pt/MAO, 2 g of 1Pt/MAO, and 2 g of 0.15Pt/Al₂O₃ catalysts.” (Lines 288-291 on Page 13 of SI)

Additional comments (not previously covered as reviewer #3):

(1) Error Bars on Coordination Number (CN) Data:

Figures 1d, 3a, and S40 report coordination numbers derived from EXAFS fitting, but currently lack error bars. Since uncertainties are already documented in Table S7, these should also be reflected in the graphical data to better represent data variability and reliability.

Reply:

We appreciate the reviewer’s suggestion. Based on the uncertainty ranges reported in Table S7 and the uncertainty propagation from kinetic fitting, we have now calculated the error margins for the coordination numbers of the clusters. These error bars have been added to Figures 2d, 3a, and S45 in the revised manuscript to more accurately reflect the variability and reliability of the data, thereby improving the overall robustness and transparency of our analysis.

Figure 2d. Site-specific TOF as a function of CN_{Pt-Pt} for different dehydrogenation reactions at 280 °C.

Figure 3a. Apparent activation energy (square) and entropy (circle) as a function of CN_{Pt-Pt} for the dehydrogenation of cyclohexane, methylcyclohexane, and decalin.

Figure S45b. Correlations of CN_{Pt-Pt} with Pt 5d-electron vacancy from EXAFs analysis.

(2) Limitations of TGA Alone for Coke Resistance and Pt Sintering Claims:

TGA analysis, by itself, cannot fully support claims of coke resistance or suppression of Pt sintering. TGA primarily measures bulk carbon oxidation with limited measurement sensitivity and may not detect smaller coke deposits directly at Pt active sites, which are critical for catalyst deactivation. Likewise, TGA does not provide information on Pt agglomeration or sintering. Given the manuscript's emphasis on stability and performance, further post-reaction analyses—such as STEM-HAADF for Pt dispersion and size, and Raman spectroscopy or XPS for coke characterization—would provide more robust evidence for the proposed stability mechanism.

Reply:

We thank the reviewer for raising this important point. In response, we conducted additional characterizations to provide more robust evidence for the proposed stability

mechanisms. First, we performed Raman spectroscopy on the spent 0.15Pt/MAO catalysts to assess possible carbon deposition. As shown in Figure S20, no distinct D and G bands indicative of coke formation were observed, indicating the absence of coke formation. This result corroborates the excellent coke resistance of the MAO-supported Pt catalyst during long-term operation. Second, we carried out HAADF-STEM imaging on the 0.15Pt/MAO catalyst after 100 hours of continuous reaction. The images (Figure S18) reveal that Pt remains predominantly as sub-nanometer clusters, with only a small fraction of nanoparticles present. The measured average particle size is 0.80 ± 0.16 nm, nearly identical to the fresh catalyst. This demonstrates that no significant Pt sintering occurred during the long-term test and supports our conclusion that the defect-engineered MgAl_2O_4 stabilizes Pt species via oxygen vacancy-mediated strong metal-support interaction. These complementary results collectively strengthen the evidence for both coke resistance and thermal stability of the Pt/MAO catalyst under operating conditions.

Figure S20. Raman spectroscopy of 0.15Pt/MAO catalyst before and after long-term

testing.

Figure S18. HAADF-STEM images and particle size distribution of spent 0.15Pt/MAO catalyst after 100 h dehydrogenation.

“In sharp contrast, 0.15Pt/Al₂O₃ suffered rapid deactivation, with conversion dropping from ~100% to ~80% within 15 h. The robust stability of Pt/MAO can be attributed to two key factors: oxygen vacancy-mediated strong metal-support interactions in the defect-engineered MAO, which effectively suppress Pt species sintering during prolonged reactions (Figure S18), and the excellent coke resistance, as evidenced by TG and Raman analysis of the spent catalyst revealing negligible carbon deposition after 100 h (Figure S19-20).” (Lines 315-320 on Page 18)

“Raman spectra were collected using a LabRAM HR Evolution spectrometer with a 325 nm laser excitation source.” (Lines 242-243 on Page 11 of SI)

Response to the comments and suggestions of the reviewers

We sincerely thank the reviewers for carefully reviewing our manuscript and their valuable comments, which certainly help improve our manuscript. We also appreciate the opportunity that editor has given us, to address the comments and revise the manuscript. The changes in the revised manuscript have been highlighted in yellow for your review. The point-by-point responses are presented below.

Reviewer #2 (Remarks to the Author):

The manuscript has been substantially improved and is now nearly ready for publication. The revisions have addressed the key scientific concerns raised previously, and the work is presented in a coherent and technically sound manner. Only a minor issue remains to be corrected.

- In Figure S17a–c, the wavenumber axis of the in situ FTIR and CO-DRIFTS spectra is plotted with lower wavenumber values on the left. It is conventional to display vibrational spectra with higher wavenumber values on the left for easier comparison and readability. Please reverse the axis orientation accordingly.

Reply:

We sincerely thank the reviewer for the positive evaluation and for pointing out this valuable detail. We have corrected the orientation of the wavenumber axis in **Figure S17a-c** to follow the conventional representation, with higher wavenumber values displayed on the left for clarity and consistency with standard vibrational spectroscopy

conventions. The revised figure has been updated accordingly in the Supporting Information.

Figure S17. In-situ FTIR spectra of 0.05Pt/MAO catalyst during (a) methylcyclohexane dehydrogenation and (b) subsequent H₂ purging. (c) CO-DRIFTS spectra of 0.05Pt/MAO before and after long-term testing.

Reviewer #3 (Remarks to the Author):

The authors have addressed most of the questions I raised. However, relying solely on the truncated octahedron model for facet analysis remains a weakness. Ultimately, precise CO-drift, TPD, and ultra-high-resolution TEM/STEM analyses are needed. Nevertheless, the paper has now reached a level suitable for publication in Nature Communications.

Reply:

We sincerely thank the reviewer for the constructive feedback and for the positive recommendation for publication. We fully agree that more precise facet identification requires complementary techniques such as CO-DRIFTS, TPD, and high-resolution TEM/STEM. In our future work, we will build upon the current study by incorporating these advanced characterizations to gain deeper insights into the structure-activity relationships.